# A multi-scale model for hair follicles reveals heterogeneous domains driving rapid spatiotemporal hair growth patterning

Qixuan Wang[1,2†], Ji Won Oh[3,4,5,6,7†], Hye-Lim Lee[3,4], Anukriti Dhar[3,4], Tao Peng[1], Raul Ramos[3,4], Christian Fernando Guerrero-Juarez[3,4], Xiaojie Wang[3,4], Ran Zhao[3,4,8], Xiaoling Cao[3,4,9], Jonathan Le[3,4], Melisa A Fuentes[3,4], Shelby C Jocoy[3,4], Antoni R Rossi[3,4], Brian Vu[3,4], Kim Pham[3,4], Xiaoyang Wang[3,4], Nanda Maya Mali[5,6], Jung Min Park[5,6], June-Hyug Choi[5,6], Hyunsu Lee[10], Julien M D Legrand[11], Eve Kandyba[12], Jung Chul Kim[7], Moonkyu Kim[7], John Foley[13], Zhengquan Yu[8], Krzysztof Kobielak[3,4,14], Bogi Andersen[4,15], Kiarash Khosrotehrani[11], Qing Nie[1,2,3*], Maksim V Plikus[2,3,4*]

[1]Department of Mathematics, University of California, Irvine, United States; [2]Center for Complex Biological Systems, University of California, Irvine, United States; [3]Department of Developmental and Cell Biology, University of California, Irvine, United States; [4]Sue and Bill Gross Stem Cell Research Center, University of California, Irvine, United States; [5]Department of Anatomy, School of Medicine, Kyungpook National University, Daegu, Korea; [6]Biomedical Research Institute, Kyungpook National University Hospital, Daegu, Korea; [7]Hair Transplantation Center, Kyungpook National University Hospital, Daegu, Korea; [8]Beijing Advanced Innovation Center for Food Nutrition and Human Health and State Key Laboratories for Agrobiotechnology, College of Biological Sciences, China Agricultural University, Beijing, China; [9]Department of Burn Surgery, The First Affiliated Hospital, Sun Yat-sen University, Guangzhou, China; [10]Department of Anatomy, School of Medicine, Keimyung University, Daegu, Korea; [11]UQ Diamantina Institute, Experimental Dermatology Group, Translational Research Institute, The University of Queensland, Brisbane, Australia; [12]Department of Pathology, Eli and Edythe Broad CIRM Center for Regenerative Medicine and Stem Cell Research, University of Southern California, Los Angeles, United States; [13]Department of Dermatology, Medical Sciences Program, Indiana University School of Medicine, Bloomington, United States; [14]Centre of New Technologies, CeNT, University of Warsaw, Warsaw, Poland; [15]Departments of Medicine and Biological Chemistry, University of California, Irvine, United States

*For correspondence: qnie@ math.uci.edu (QN); plikus@uci.edu (MVP)

†These authors contributed equally to this work

Competing interests: The authors declare that no competing interests exist.

**Abstract** The control principles behind robust cyclic regeneration of hair follicles (HFs) remain unclear. Using multi-scale modeling, we show that coupling inhibitors and activators with physical growth of HFs is sufficient to drive periodicity and excitability of hair regeneration. Model simulations and experimental data reveal that mouse skin behaves as a heterogeneous regenerative field, composed of anatomical domains where HFs have distinct cycling dynamics. Interactions between fast-cycling chin and ventral HFs and slow-cycling dorsal HFs produce bilaterally symmetric patterns. Ear skin behaves as a hyper-refractory domain with HFs in extended rest phase. Such hyper-refractivity relates to high levels of BMP ligands and WNT antagonists, in

part expressed by ear-specific cartilage and muscle. Hair growth stops at the boundaries with hyper-refractory ears and anatomically discontinuous eyelids, generating wave-breaking effects. We posit that similar mechanisms for coupled regeneration with dominant activator, hyper-refractory, and wave-breaker regions can operate in other actively renewing organs.

## Introduction

Featuring prominent growth cycles, the hair follicle (HF) is a model system of choice for studying tissue regeneration. At the level of cellular activities, the hair growth cycle consists of three consecutive phases: anagen, phase of active proliferation; catagen, apoptotic involution phase; and telogen, relative quiescence phase (*Al-Nuaimi et al., 2010*; *Paus and Foitzik, 2004*; *Schneider et al., 2009*; *Stenn and Paus, 2001*). Cyclic regeneration is sustained by the bulge stem cells, located at the base of the permanent HF portion (*Cotsarelis et al., 1990*). During anagen initiation, signals from the niche, including the dermal papilla (DP), stimulate bulge stem cells and adjacent hair germ (HG) progenitors to proliferate (*Enshell-Seijffers et al., 2010*; *Greco et al., 2009*; *Legrand et al., 2016*). Activated progenitors generate all lower HF structures, including the outer root sheath (ORS) and hair matrix. During catagen, a widespread apoptotic program remodels the HF back toward a telogen state (*Botchkarev et al., 2001b*; *Fessing et al., 2006*; *Foitzik et al., 2000*; *Lindner et al., 1997*; *Mesa et al., 2015*). Conceptually, since the bulge produces downward migrating progeny (*Hsu et al., 2011*), it effectively serves as a progenitor source, while the matrix functions as a sink, and the ORS as a channel for progenitors transiting between them.

The signaling mechanisms that time these coordinated cellular activities during hair regeneration remain incompletely understood (*Al-Nuaimi et al., 2014*; *Bernard, 2012*; *Lin et al., 2009*; *Paus et al., 1999*). The putative 'hair cycle clock' is thought to be composed of one or several activator/inhibitor pairs acting to time key cycle phase transitions at set thresholds of their activities. Accordingly, cycle pace will depend on the speed at which activators and inhibitors reach their respective thresholds (*Chen et al., 2015*). Importantly, HFs exist as large populations and at least in the dorsal skin they interact to coordinate growth cycles (*Hodgson et al., 2014*; *Plikus et al., 2011*, *Plikus and Chuong, 2008a*, *Plikus et al., 2008b*). Such coordination implies that at least some of the activators and inhibitors should be present between HFs, in the so-called skin macro-environment. Previous work on dorsal skin indicates that BMP and WNT pathways constitute important components of the hair cycle clock. Indeed, defects in either of these pathways can dramatically change hair cycle progression (*Botchkarev et al., 2001a*; *Botchkarev and Sharov, 2004*; *Choi et al., 2013*; *Enshell-Seijffers et al., 2010*; *Kandyba and Kobielak, 2014*; *Kandyba et al., 2013*; *Kobielak et al., 2003*, *2007*; *Sharov et al., 2005*, *2006*), and ligands and antagonists for both pathways mediate macro-environmental coordination between HFs (*Chen et al., 2014*; *Plikus et al., 2011*; *Plikus and Chuong, 2014*). Additionally, FGF, PDGF, TGFβ, TNFα and other pathways can modulate hair cycle (*Chen et al., 2015*; *Festa et al., 2011*; *Higgins et al., 2014*; *Ito et al., 2003*; *Kimura-Ueki et al., 2012*; *Leishman et al., 2013*; *Oshimori and Fuchs, 2012*; *Plikus, 2012*; *Rivera-Gonzalez et al., 2016*). Importantly, the combined signaling activities for the above pathways partition the hair cycle in the dorsal skin into four functional phases, each with its distinct activator/inhibitor profile: propagating anagen (*P*), autonomous anagen (*A*), refractory telogen (*R*) and competent telogen (*C*) (*Hodgson et al., 2014*; *Plikus and Chuong, 2014*; *Plikus et al., 2008b*). Interactions between HFs enable hair regeneration across dorsal skin to self-organize into dynamic patterns. Critical for this self-organization are the following HF-to-HF interactions: *P*-phase HFs can induce neighboring *C*-phase HFs to enter anagen via diffusible activators, leading to hair growth coupling, while *A*-phase anagen or *R*-phase telogen HFs cannot couple due to high levels of inhibitors (*Murray et al., 2012*; *Plikus et al., 2011*; *Plikus and Chuong, 2014*; *Plikus et al., 2008b*). It remains unknown, however, whether this self-organization mechanism and its underlying WNT/BMP signaling activities is a general feature of all body skin or a special case for dorsal skin only.

Integrative understanding of large-scale hair regeneration requires a systems biology approach. Previous modeling on HFs include cellular automaton models (*Halloy et al., 2000*; *Plikus et al., 2011*), feedback-control model (*Al-Nuaimi et al., 2012*) and the FitzHugh-Nagumo (FHN) excitable medium model (*Murray et al., 2012*). Here, we present a unified three-dimensional and stochastic

**eLife digest** Skin includes hundreds of thousands of hair follicles that cycle through different stages of activity. Each follicle grows hair, sometimes (in the case of long hairs like human head hair and horse tail hairs) for several years, before losing it. The follicle then goes through a resting stage before starting to grow another hair. To achieve high hair density, the follicles need to coordinate their hair-making activities. If they all worked independently from one another, bald patches would inevitably form that would compromise how effectively the skin works.

Groups of cells can communicate using a variety of chemical signals. It was not known whether cells in hair follicles from different regions of the skin rely on the same signals to communicate, and whether follicles in neighboring regions are able to 'understand' one another.

Through a combination of mathematical modeling and experimental results from mice, Wang, Oh et al. now show that hair follicles across the body use a common signaling system. This system consists of a pair of signals: 'activators' that stimulate hair growth, and 'inhibitors' that prevent it. The balance between these two signals affects the pattern of hair growth. For example, higher levels of activators allow fur to grow thickly on the belly of the mouse, likely to protect against heat loss and injuries from the ground. By contract, higher levels of inhibitors make the hairs on the ear sparse, which may prevent them from interfering with hearing.

There is little evidence that hair follicles on the scalp communicate in adult humans. Learning to activate and control communication between these follicles could provide a way to treat male pattern baldness and similar conditions. Understanding how hair follicles communicate may also help researchers to develop ways of regenerating other fast-renewing organs, such as the gut and bone marrow.

modeling framework for the HF that captures: (i) activator/inhibitor signaling dynamics in a single HF, (ii) cyclic growth of a single HF, and (iii) coupling between multiple HFs through diffusive signals. Using this model, we reveal that skin as a whole behaves as a heterogeneous regenerative field, where: (a) dominant hair cycle waves start in the ventrum, (b) propagate dorsally in a bilateral pattern, (c) stop at the boundary with hyper-refractory ear skin, and (d) break at non-propagating anatomical landmarks, such as eyelids and ears. We also show that WNT and BMP serve as a universal activator/inhibitor signaling pair, whose varying activities underlie distinct hair regeneration dynamics in all anatomical locations studied. These results provide new understanding of how the entire skin of the animal manages all of its hair regeneration.

## Results

### A multi-scale model recapitulates a single growing HF, as well as HF-to-HF communication

First, in modeling the geometry of a single HF, we considered four key expression sites for activator/ inhibitor ligands, antagonists, and receptors along the HF axis: bulge, HG, matrix, and DP (*Figure 1A*). During the cycle, the bulge (assigned as Region I) remains relatively static, whereas the DP moves up and down along the HF axis. Also dynamic are the HG and matrix. The former only exists during telogen, while the latter only exists during anagen. The HG grows down to make matrix during anagen onset, whereas during catagen, the matrix collapses, and a new HG reforms. Simplistically, cyclic HG→matrix→HG dynamics are coordinated with the DP; thus, in the model we identify them jointly as Region II. Next, we considered that both regions produce signaling factors. Although a biological simplification, we assumed that Region I does so at a rather constant rate, while Region II shows distinct temporal dynamics (*Appendix 2—table 4*). We also assumed that Region II is essential for sending hair cycle-promoting signal(s), while Region I is the primary signal target. In short, we hypothesized that the essential temporal molecular dynamics in the HF operate as follows: Region II generates a signaling ligand (L) gradient; Region I detects it and transmits it into ligand-bound receptors (LR) that then, through a series of intermediate signaling steps not captured in the model directly (such as activities of the downstream signaling pathways and involvement of additional cell

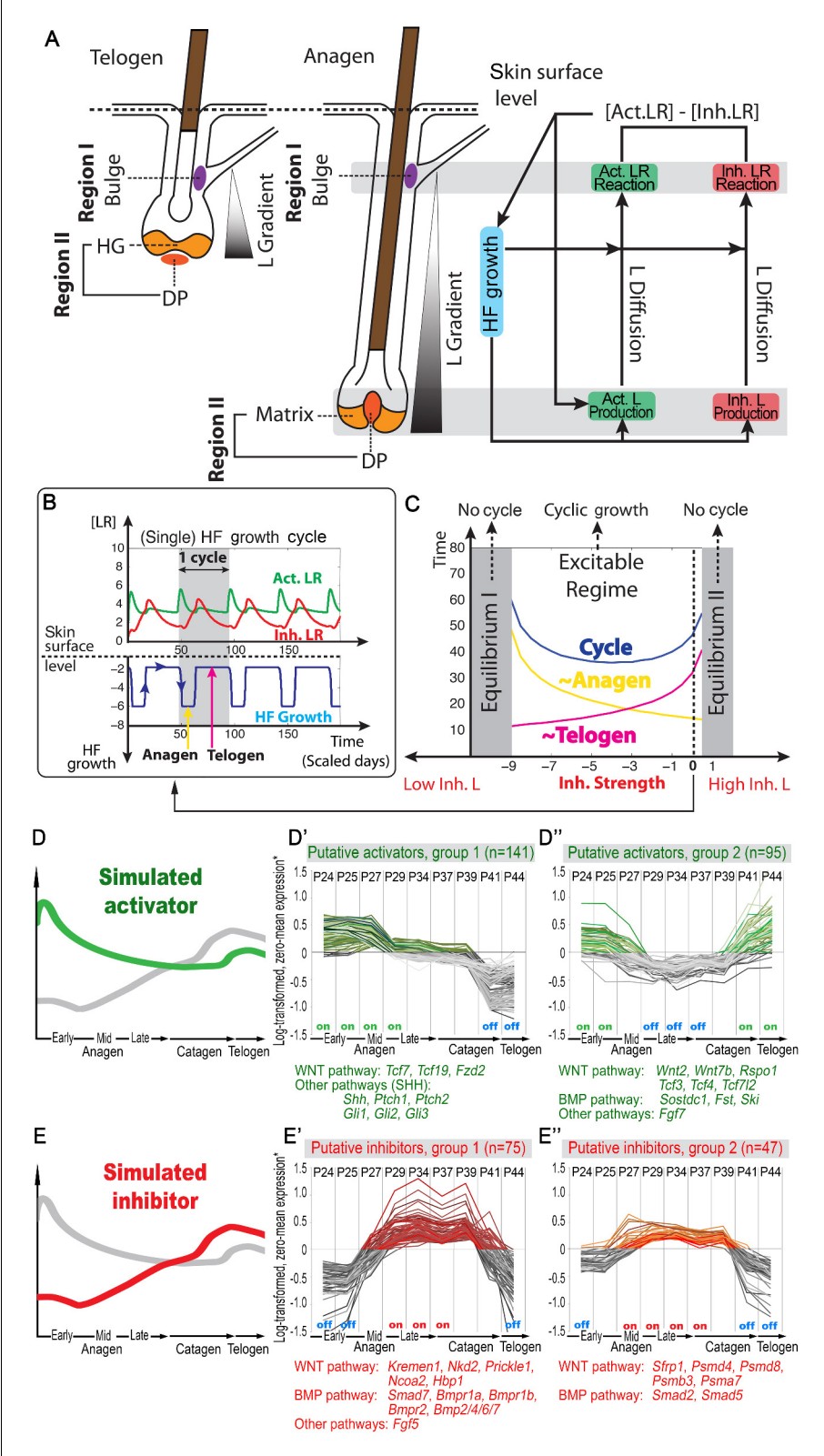

**Figure 1.** Model recapitulates hair cycling and its associated activator and inhibitor signaling dynamics. (**A**) Schematic depiction of HF growth dynamics during telogen and anagen. Telogen and anagen HFs are shown on the left and in the center, respectively. In both hair cycle phases, Region I (purple) represents bulge and Region II (orange) represents DP with HG during telogen phase, and DP with matrix during anagen phase. On the right, schematic drawing of diffusive activator (Act. L. in green) and inhibitor (Inh. L. in red) interactions with their corresponding receptors (Act. R and Inh. R,

*Figure 1 continued on next page*

*Figure 1 continued*

not depicted) that form ligand-bond-receptors (Act. LR and Inh. LR) and their coupling with physical growth of the HF (blue) is shown. (**B**) Typical noise-free dynamics of the activator (green) and inhibitor (red) and cyclic HF growth (blue) are shown. X-axis is time in simulated days. Y-axis for activator and inhibitor shows simulated signaling levels, and for HF growth – simulated length of the HF. Grey area demarcates one modeled hair growth cycle. (**C**) The duration of ~*anagen* and ~*telogen* phases as the function of inhibitor signaling strengths. X-axis shows modeled inhibitor levels with '0' being an arbitrary baseline levels. Y-axis shows time in simulated days. Upon stronger inhibitory signaling (high Inh. L level) ~*anagen* shortens (yellow) and ~*telogen* lengthens (purple). The entire cycle (blue) becomes longer either with stronger or weaker inhibitory signaling. When inhibitory signaling becomes either very strong or very weak, the excitability of the system breaks down and HFs equilibrate in one state (grey regions). Also see *Appendix 2—tables 1*, *2* and *4*. (**D–E''**) A total of 236 putative activator genes (green) and 122 putative inhibitor genes (red) available from a whole skin microarray dataset were identified to recapitulate temporal dynamics of the simulated activator (**D**) and inhibitor (**E**), respectively. Multiple WNT pathway members are in the putative activator gene set (**D'**, **D''**), while BMP pathway members are among the putative inhibitor genes (**E'**, **E''**). See gene list in Dataset 1. For all genes log-transformed, zero-mean expression profile values were calculated using colorimetric ratio-scale algorithm as reported in (*Lin et al., 2009*).

populations), regulates cyclic HF growth (*Figure 1A*). The molecular signaling events, either activating or inhibitory, can be summarized as:

$$\frac{\partial}{\partial t}[L] = \text{Diffusion} + \text{Production} + \text{Reaction of L and R} \qquad (1)$$

$$\frac{\partial}{\partial t}[LR] = \text{Reaction of L and R} + \text{Degradation} + \text{Extra Source} \qquad (2)$$

where L, R and LR stand for ligands, receptors, and ligand-bound receptors, respectively. In the dynamics of LR (*Equation 2*), the 'Extra Source' describes stochastic signaling effects due to noise, and potential signaling contributions from Region I (Appendix 2-Governing equations for activators and inhibitors). As *Equations 1 and 2* show, ligand-receptor interactions in the model take place only for the same signaling pathway, and no direct pathway cross-talk is set to occur. This, again, is a biological simplification. Recently, evidence for pathway interactions have emerged (*Kandyba et al., 2013*), and its effect is explored in Appendix 2-Possible interactions between the activator and inhibitor pathways do not qualitatively alter the HF dynamics.

Our model integrates key signaling features of the hair growth cycle: strong activator signals enhance HF growth, while strong inhibitor signals prevent it. We modeled HF growth through the spatial average of LR concentration differences between the levels of activator and inhibitor in Region I (*Equation 7* in Appendix 2-Modeling HF phases by concentration difference). We assumed the hair cycle has two critical 'checkpoints': (i) the event in late competent telogen, when production of activator starts to increase (*Chen et al., 2014*; *Greco et al., 2009*; *Oshimori and Fuchs, 2012*; *Plikus et al., 2008b*), and (ii) the event of anagen termination, when the HF starts to involute. Thus, our model recognizes two phases determined by these checkpoints: ~*anagen*, starting from the moment of activator amplification until anagen termination, and ~*telogen*, lasting until the next activator amplification event. In the context of the conventional hair growth cycle, ~*anagen* incorporates the late portion of competent telogen and the entire anagen, while ~*telogen* includes catagen, refractory telogen and the remainder of competent telogen (*Plikus et al., 2011*; *Plikus and Chuong, 2014*; *Plikus et al., 2008b*) (Appendix 2-Modeling HF phases by concentration difference; *Appendix 2—figure 2*).

Model simulations produce several emergent behaviors. The cycle becomes autonomous – that is, it displays stable periodicity and excitability emerges naturally without a built-in 'clock' (*Figure 1B*). Cycling is maintained within a range of parameter values, allowing testing for various intrinsic and extrinsic signaling scenarios (*Figure 1C*). Associated with these dynamics are periodic changes in the system's geometry – the signaling source in Region II moves cyclically. Simulations indicate that the moving HF geometry in the model is critical, greatly contributing to the regulation of the cycle. In a single HF model, activator/inhibitor diffusion occurs only along the HF axis. When a HF population is modeled, hair-to-hair communication emerges naturally as ligand diffusion from neighbors supplements intrinsic HF ligand levels. As such, hair cycle pace depends on interactive signaling between neighboring HFs – a feature that we explore below.

## HF cycling emerges from the growth-mediated coupling of activator and inhibitor

Our model predicts that HF cycling occurs only within a certain range of signal strengths, that is the excitable regime (*Figure 1C*, white region). Within this regime, activator and inhibitor are predicted to inversely modulate duration of both ~*telogen* and ~*anagen* phases. At certain, either too high or too low signal strengths, the excitability is predicted to break down and the HF is expected to enter a non-cycling state of equilibrium (*Figure 1C*, grey regions). For example, when inhibitor levels are very high, the HF is predicted to equilibrate in an extended telogen (*Appendix 2—figure 5A*), while extended anagen is predicted for the opposite signaling condition (*Appendix 2—figure 5B*).

Next, we used bioinformatic and experimental approaches to validate the model's key prediction that the same activator or inhibitor pathway can inversely modulate telogen and anagen phase duration. Considering the established roles for BMP and WNT as respective inhibitor and activator pathways regulating telogen duration in the dorsal skin, we explored if they can also regulate anagen duration in the same skin region in a model-predicted fashion. First, we found that model-predicted temporal dynamics for inhibitor and activator during ~*anagen* (*Figure 1D and 1E*) match the actual anagen expression dynamics for multiple BMP and WNT pathway members established on a highly temporally resolved whole-tissue dorsal skin microarray dataset (*Lin et al., 2009*) (*Figure 1D–E''*; Appendix 1-Identifying model predicted hair cycle activators and inhibitors). We also show that perturbing BMP (for details see Appendix 1-Validating model-predicted roles for BMP signaling in hair cycle control) and WNT in transgenic mice (for details see Appendix 1-Validating model-predicted roles for WNT signaling in hair cycle control) alters dorsal anagen phase duration and leads to hair length defects in a way that is consistent with the model's predictions. Overall, this data shows that our model generates biologically meaningful outcomes and that its predictive power is robust.

## Model reveals skin is a heterogeneous regenerative field

Next, we set out to explore novel aspects of hair regeneration at the population level. For this purpose, we modeled a linear array of HFs (i.e. two-dimensional organization; *Appendix 2—figure 4A*) and a grid of HFs (i.e. three-dimensional organization; *Appendix 2—figure 4B*). In both cases, the diffusion of activators and inhibitors accompanying each HF during growth naturally led to HF coupling (Appendix 1-Validating model-predicted roles for BMP signaling in hair cycle control) and emergence of several known features of collective hair growth behavior, including spontaneous anagen initiation and anagen wave spreading (*Appendix 2—figures 11, 12*). We then focused on the phenomenon of bilaterally symmetric hair growth that is prominent in young mice (*Plikus et al., 2008b*) yet remains unexplained. Conventionally, first anagen in the dorsal skin of newborn mice is considered synchronous. On the other hand, adult mice display fully asynchronous and asymmetric dorsal hair growth patterns (*Chen et al., 2014*; *Plikus and Chuong, 2008a*; *Plikus et al., 2009*). This, however, is preceded by prominent bilateral symmetry, which often persists into the fourth hair cycle (*Plikus and Chuong, 2008a*). We now show that in the three-dimensional model where all HFs are assumed to be identical, full asynchrony evolves within just one cycle, and bilateral symmetry cannot be achieved (Appendix 2-Dorsal and ventral HF patterns; *Appendix 2—figures 11, 12*; *Appendix 2—video 1*). Therefore, we hypothesized that first anagen is inherently asynchronous as a result of spatially patterned HF development. Indeed, spatial distribution of early anagen HFs in the dorsal skin of newborn mice (*Figure 2A–D*) reveals head-to-tail and subtle lateral-to-medial asynchronies. We modeled the impact of these asynchronies on hair growth pattern evolution. Simulations reproduced head-to-tail asynchrony (Appendix 2-Dorsal and ventral HF patterns; *Appendix 2—figures 14, 15*; *Appendix 2—video 3*); however, it persisted for at least 10 cycles, which is far more than the 3–4 cycles observable in mice. Moreover, prominent bilateral symmetry failed to form.

We note that the above and previous simulations (*Murray et al., 2012*; *Plikus et al., 2011*) were performed on homogenous HF populations, where all HFs are assumed to be identical. We then considered that novel patterns might develop upon interaction of two or more HF populations, whose activator/inhibitor signaling levels are inherently different. In principle, dorsal skin HFs can interact with HFs from other body regions, such as ventral skin, where hair cycle dynamics are potentially distinct. Because all skin is continuous and forms an approximation of a cylinder, we modeled it as an unrolled sheet, where two *Ventral* sub-domains flank a rectangular *Dorsal* domain (*Figure 2E*).

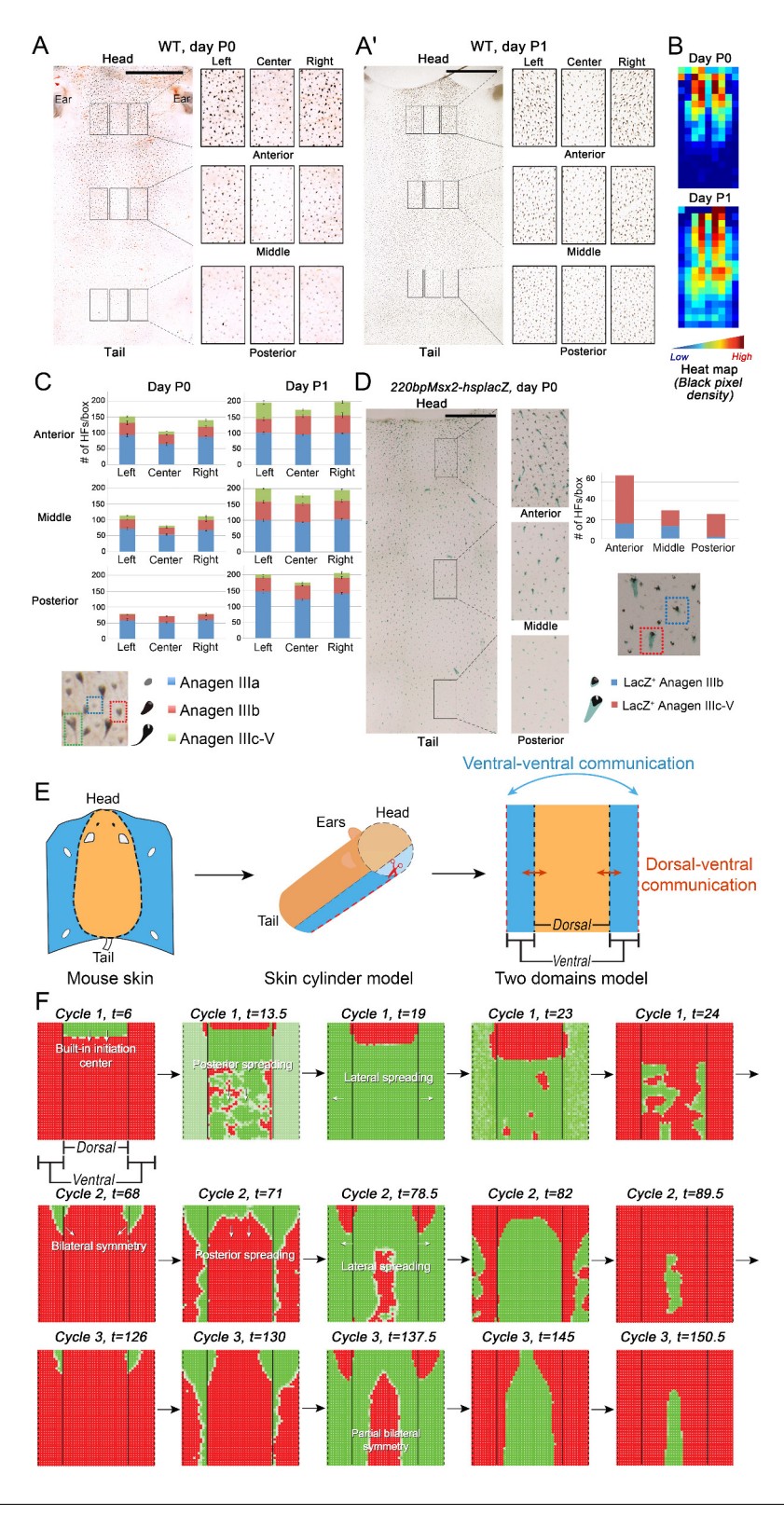

**Figure 2.** Spatiotemporal patterning of early hair cycles. (A–D) Analysis of the whole mount dorsal skin samples from P0 (n = 3) (A) and P1 WT mice (n = 3) (A') reveals subtle head-to-tail and lateral-to-medial hair cycle asynchronies. Asynchronies were inferred from examining the size of pigmented HFs. Larger HFs result from earlier anagen onset. (B) Heatmaps of skin samples from A and A' built based on black pixel density (reflecting pigmented anagen HFs). (C) Quantification of anagen HFs at different phases confirms head-to-tail pattern asynchrony. Morphological definition of anagen phases

*Figure 2 continued on next page*

*Figure 2 continued*

used for this analysis is provided at the bottom on the panel. (D) Analysis of the whole mount dorsal skin samples from P0 *220bpMsx2-hsplacZ* mice, where lacZ reporter activates in anagen HFs starting from phase IIIb, confirms head-to-tail asynchrony. (E, F) Modeling rapid hair growth pattern evolution in the context of two heterogeneous domains. (E) Schematic depiction of the modeling conditions with central *Dorsal* domain flanked by two lateral *Ventral* sub-domains with coupling between *Dorsal* and *Ventral* HFs. (F) Compared to *Dorsal* domain HFs, *Ventral* domain HFs were assigned with higher levels of total available activator and inhibitor receptors, allowing shorter ~*anagen* and ~*telogen* duration. Furthermore, hair cycle asynchrony was introduced into *Dorsal* domain to model the initial head-to-tail asynchrony. In simulations, interactions between HFs across domain boundaries result in bilateral symmetry during the second cycle (simulated time t68-78.5). Also, initial asynchrony breaks down in the cycle 3 (t130), and partial bilateral symmetry maintains into the late cycles (see *Appendix 2—video 4*). Scale bars: A, A', D – 5 mm. Images on A, A' and D are composites.

For initial modeling conditions (Appendix 2-Dorsal and ventral HF patterns), we considered that: (i) the first cycle on the dorsal skin has built-in head-to-tail asynchrony, and that (ii) ventral HFs develop with a 3- to 4-day delay relative to dorsal HFs (*Appendix 1—figures 7–9*). Because ventral HFs are known to produce distinctly shorter hairs (*Candille et al., 2004*), in the model we assumed that they have faster cycle dynamics compared to dorsal HFs (Appendix 2-Changes in the total amount of activator and inhibitor receptors results in different sensitivity of ~anagen and ~telogen lengths to signaling changes, Appendix 2-Dorsal and ventral HF patterns; *Appendix 2—figures 8*, *9*). Indeed, in this configuration, our model readily reproduced patterns with aspects of bilateral symmetry already in the second cycle as the result of dominant waves spreading from the *Ventral* to the *Dorsal* domain (*Figure 2F*, t68-78.5). Importantly, after the second cycle, the effect of the initial built-in head-to-tail asynchrony started to disappear. Instead, the interaction between *Ventral* and *Dorsal* HFs continued to produce prominent bilateral symmetry in the third (*Figures 2F*, t130-145) and later cycles (*Appendix 2—video 4*). Taken together, the model predicts that rapid hair growth pattern evolution requires interaction of two or more skin domains with distinct hair cycle parameters.

## Ventral-dorsal interactions produce bilaterally symmetric hair growth patterns

Next, we imaged *Flash* mice, whose luciferase reporter produces skin-specific WNT activity signal and allows to sensitively and non-invasively determine the location and percentage of anagen HFs across the entire body (*Hodgson et al., 2014*) (*Figure 3A–C*). Luminescence levels were measured both dorsally and ventrally and mice were followed up until day P119, encompassing up to five hair growth cycles. Combined analysis from multiple mice reveals prominent phase advancement in ventral over dorsal anagen, specifically during the second, third, and fourth hair cycles (*Figure 3B*, blue area). Additionally, the spatial luminescence signal mapping reveals distinct ventral-to-dorsal anagen propagation with features of bilateral symmetry during second (*Figure 3C*; *Appendix 1—figure 6*) and third cycles (*Figure 3C'*), supporting the patterning mechanism predicted by the model (*Figure 2F*). We also mapped body-wide hair growth patterns on the basis of anagen HF pigmentation between days P0-P55 (*Figure 3D–G*; *Appendix 1—figures 7–12*). This analysis confirms ventral over dorsal anagen phase advancement starting from the second cycle and provides the following additional insights:

(i) Ventrally, anagen phase is the shortest in the 'chin domain', ending around P10. It is longer in the 'ventral domain' proper, ending in the genital area around P14 and in the chest area around P17 (*Figure 3D*).

(ii) Dorsally, anagen is the shortest in the 'cranial domain', ending around P14. In the 'dorsal domain' proper it ends as a head-to-tail wave between P15-P20 (*Figure 3D*; *Appendix 1—figures 10*, *11*).

(iii) First ventral telogen is shorter than dorsal telogen. Second anagen initiates in the chin and ventral domains already between P21-24 and then spreads toward ventral-dorsal boundaries in form of two bilaterally symmetric waves (*Figure 3E*; *Appendix 1—figure 12*). Second anagen also ends faster in the ventral skin, maintaining ventral-dorsal asynchrony and bilateral symmetry (*Figure 3F*).

(iv) Third anagen initiates the fastest in the chin domain, as early as P42 (*Figures 3G* and *4H*).

When transplanted onto the back of pigmented SCID mice, chin skin grafts (n = 8) showed faster cycling compared to dorsal skin grafts (n = 8). While first post-transplantation anagen started with

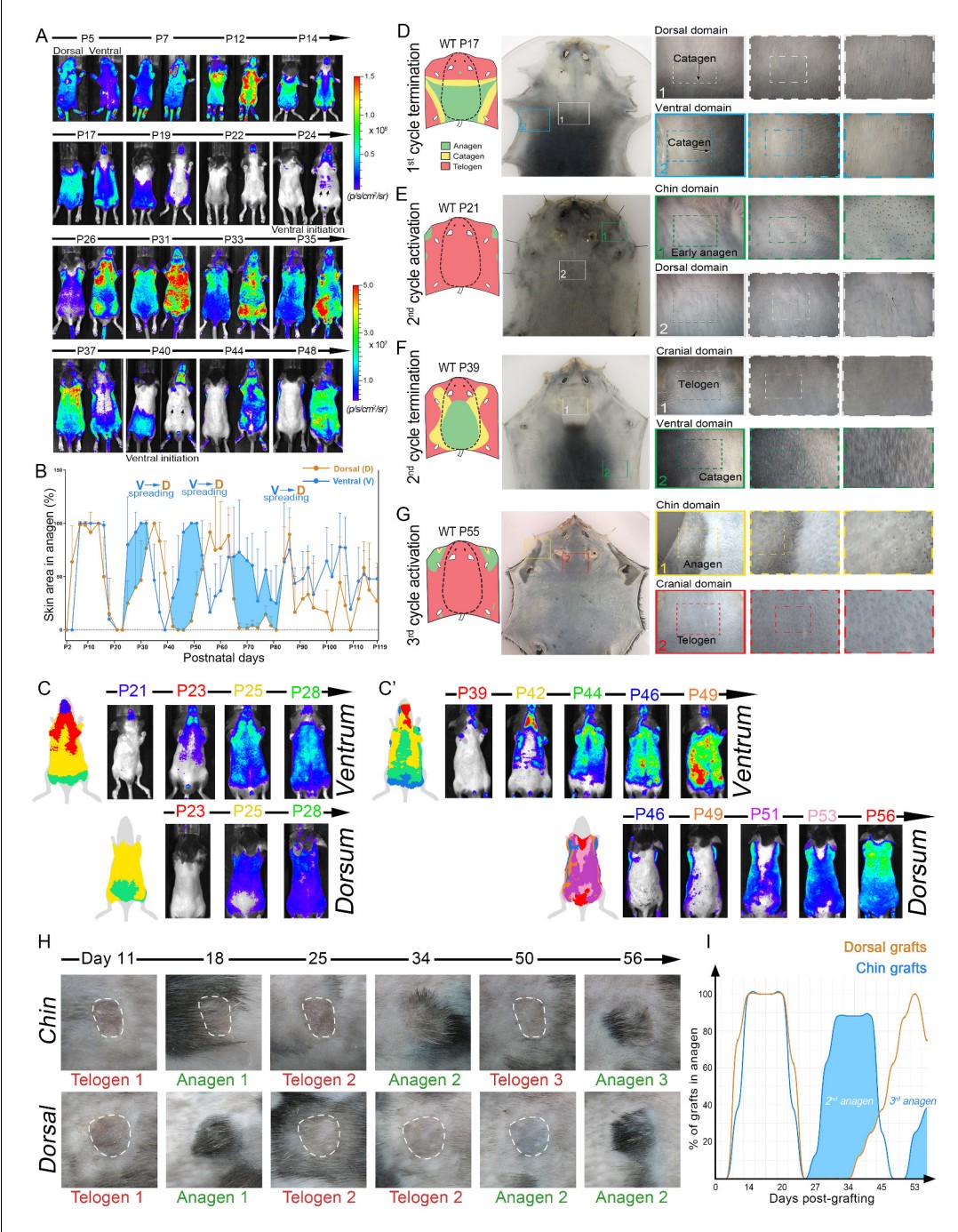

**Figure 3.** Dorsal-ventral HF interactions produce bilateral symmetry. (**A**) Time-lapse bioluminescence in dorsal and ventral skin of the representative *Flash* mouse between days P5-P48. Bioluminescent signal is color-coded according to the colorimetric scale on the right. (**B**) Combined temporal dynamics (from 6 *Flash* mice) of the bioluminescent signal-based anagen measurements over four hair cycles (days P5-P119). Dorsal skin dynamics are in brown and ventral skin dynamics are in blue. Prominent temporal advancement of ventral over dorsal anagen initiation can be seen during second, third and fourth cycles (light blue areas). This advancement is accompanied by dominant ventral-to-dorsal anagen wave spreading. (**C, C'**) Mapping of *Flash*-based anagen reveals ventral-to-dorsal hair growth wave propagation and bilateral pattern symmetry. New anagen areas for each time point are color-coded. Second anagen initiation is shown on panel **C**, and third anagen initiation on panel **C'**. Also see *Appendix 1—figure 6*. (**D–G**) Hair growth distribution patterns on P17 (**D**), P21 (**E**), P39 (**F**) and P55 (**G**). Three mice were analyzed at each time point. Inverted whole mount skin samples from representative mice are shown. Schematic pattern maps are provided with color-coded anagen (green), catagen (yellow) and telogen (red) regions. Also see *Appendix 1—figures 7–12*. (**H, I**) HF cycling dynamics in chin skin grafts remain faster compared to dorsal skin grafts. After transplantation, first anagen initiated similarly in both chin and dorsal skin grafts, however, second anagen initiated significantly faster in chin grafts. Representative chin and

*Figure 3 continued on next page*

*Figure 3 continued*

dorsal skin grafts are shown on (**H**). Combined temporal dynamics of skin grafts in anagen and telogen are shown on (**I**). Dorsal graft dynamics are in brown and chin graft dynamics are in blue. Temporal advancement of chin over dorsal second anagen initiation is highlighted with light blue color. Also see *Appendix 1—figure 13*.

similar timing in both chin and dorsal grafts, consecutive anagen started significantly faster in chin grafts (*Figure 3H and I*; *Appendix 1—figure 13*). Furthermore, in many instances, grafts induced anagen in the surrounding dorsal host skin. Taken together, these data support that dominant ventral-to-dorsal hair wave spreading drives rapid hair growth pattern evolution and bilateral symmetry. Underlying this behavior are faster hair growth cycle dynamics in chin and ventral HFs, a property that is partially maintained upon skin grafting.

Next, we asked if faster hair cycle dynamics in chin and ventral domains correlate with distinct molecular dynamics in putative activators and inhibitors. We performed RNA-seq profiling of whole skin from chin, ventral and dorsal domains at six hair cycle time points: first (*aka* competent) telogen, early anagen, mid-anagen, late anagen, catagen and early second (*aka* refractory) telogen. Analysis revealed non-overlapping transcriptomic trajectories of the hair cycle between the three domains (*Figure 4A–B''*) and domain-specific expression patterns for multiple putative activator and inhibitor genes at all hair cycle time points (*Appendix 1—figures 14–19*; Dataset 2). We then asked if refractory properties of early telogen differ between the domains. Differential gene expression analysis (*Figure 4C–D*) revealed enrichment in chin and ventral domains for gene ontologies related to macrophage function and lipid storage, and enrichment in chin domain for muscle-related genes (*Figure 4E*). Consistently, chin skin shows more contractile cells around HFs, and chin and ventral skin have thicker dermal adipose tissue and substantially more CD11b$^+$;F4/80$^+$ macrophages as compared to dorsal skin (*Appendix 1—figures 20*, *21*). Furthermore, dorsal early telogen skin shows gene expression changes consistent with higher refractivity – it is enriched for several BMP ligands, and depleted for BMP antagonists and WNT ligands (*Figure 4F*). Consistently, in *Axin2-lacZ* WNT reporter mice, many more HFs with WNT-active DPs are seen in chin and ventral as compared to dorsal skin at P36 (*Figure 4G*; *Appendix 1—figure 22A*). WNT activity increases in dorsal skin to the levels of ventral skin only by P42 (*Figure 4H*; *Appendix 1—figure 22B*). Furthermore, in P42 *BRE-gal* BMP reporter mice, many more HFs with BMP-active bulges are seen in dorsal as compared to chin and ventral skin (*Figure 4I*; *Appendix 1—figure 22C*). In *Krt14-Wnt7a* mice, spontaneous anagen initiation sites in dorsal skin overrun ventral-to-dorsal wave dominance (*Figure 4J*; *Appendix 1—figure 23*). In contrast, in *Krt14-Bmp4* mice, ventral-dorsal hair growth waves stall and asymmetric anagen patches form instead (*Figure 4K*). Together, this data confirms that lower refractivity and the underlying differences in BMP and WNT activities form the bases for ventral-dorsal hair growth dominance.

## Ear pinna behaves as a hyper-refractory skin domain

Our model also predicts conditions when hair cycling stops and HFs equilibrate in an extended telogen, such as due to high levels of inhibitors (Appendix 2-Hyper-refractory domain; *Appendix 2—figure 5A*, *16*). We profiled mouse skin for the existence of such behavior and found ears to match such prediction. In the ear skin, HF morphogenesis begins between days P2-P4, and HFs remain in anagen until about P15 (*Figure 5A*). After first anagen, and for at least three months, they remain in an extended telogen, while at the same time dorsal HFs have already reached their third cycle (*Figure 5B and C*). Seldom, solitary anagen HFs can be found, but no coordinated hair growth waves, characteristic to other skin regions, are observed (*Figure 5B*, day P95). Moreover, anagen waves spreading from cranial skin could not propagate into ear skin (*Figure 7E*). These observations are consistent with the possibility that ear skin is hyper-refractory. Next, we examined ear HFs' responses to several potent anagen inducers: cyclosporin A (*Maurer et al., 1997*; *Paus et al., 1989*), smoothened agonist (SAG) (*Paladini et al., 2005*) and hair plucking (*Chen et al., 2015*). We show that while dorsal telogen HFs readily respond to cyclosporin A (*Appendix 1—figure 27B*), ear HFs remain quiescent even 3 weeks after treatment (*Figure 5F*). Anagen can be induced in response to SAG; however, this response occurs late, after 3 weeks, and remains restricted to the medial side of the ear (*Figure 5E*). This is contrasted by rapid SAG-induced anagen in dorsal skin (*Appendix 1—*

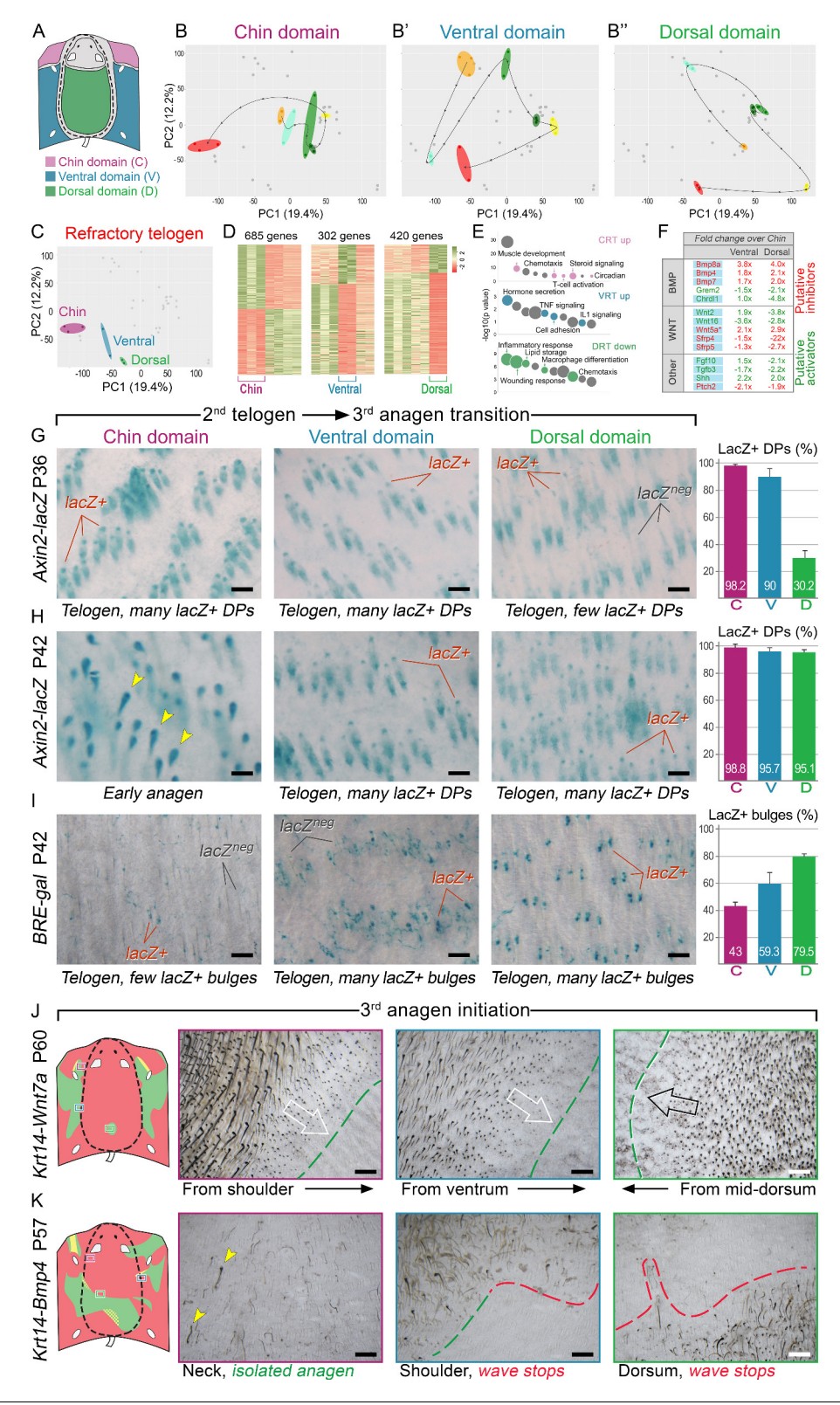

**Figure 4.** BMP and WNT signaling differences underlie regionally specific telogen phase duration. (A–B'') PCA analysis reveals largely non-overlapping transcriptomic trajectories across six hair cycle stages in chin (B), ventral (B') and dorsal domains (B''). Combined, deconstructed PCA plots are shown with all data points marked as grey dots and domain-specific data points outlined and color-coded. Color-coding is based on the hair cycle timeline from ***Appendix 1—figure 14A***; transcriptomic trajectories are drawn with dark lines. (C) Deconstructed PCA plot for refractory (early second) telogen is

*Figure 4 continued on next page*

*Figure 4 continued*

shown with domain-specific data points highlighted and color-coded based on the schematic drawing on A. (D–F) Analysis of refractory telogen data identified 1407 differentially expressed genes across the three domains (D), with each domain showing enrichment for distinct gene ontologies (E). Multiple putative hair cycle activator and inhibitor genes show domain-specific differential expression (F). Putative activators are in green and putative inhibitors are in red. For each gene, relative fold changes for ventral over chin and dorsal over chin expression levels are indicated. Genes that show cyclic expression patterns are highlighted with blue. See additional expression data analysis on *Appendix 1—figures 14–19* and in Dataset 2. Asterisk marks non-canonical WNT ligand. (G, H) Analysis of *Axin2-lacZ* skin during second telogen reveals faster activation of WNT signaling in chin and ventral HFs over dorsal HFs. At P36 majority of HFs in chin and ventral skin have WNT-active DPs. In dorsal skin, the number of HFs with WNT-active DPs is low at P36, but increases by P42. (I) Analysis of P42 *BRE-gal* skin shows that many more dorsal HFs have BMP-active bulges as compared to chin and ventral HFs. Also see *Appendix 1—figure 22*. (J) Overexpression of Wnt7a results in disruption of ventral-to-dorsal hair growth wave dominance and spontaneous anagen appears in the dorsal domain at P60. (K) Overexpression of Bmp4 results in stalled ventral-to-dorsal hair growth wave spreading and patchy, asymmetric hair growth at P57. Scale bars: G-I – 200 um, J-K – 500 um.

figure 27A). Plucking induces anagen along the medial side of the ear; however, there is no anagen wave spreading into the unplucked region, a feature common in dorsal skin (*Chen et al., 2015*) (*Figure 5D*; *Appendix 1—figures 25A-B*, *26*). Furthermore, whole ear plucking experiments reveal very sparse anagen activation along the lateral side (*Appendix 1—figure 25C*). These data demonstrate that physiologically, adult ear HFs equilibrate in a hyper-refractory telogen state, yet in principle remain capable of regenerative cycling in response to selective external stimuli.

To understand how ear HF hyper-refractivity relates to activator and inhibitor signaling levels, we compared on RNA-seq refractory telogen dorsal skin with telogen ear skin and, additionally, cartilage/muscle complex, a structure unique to ears. We show that, transcriptionally, these three tissue types are distinct (*Figure 6A*), containing large number of differentially expressed genes (*Figure 6B*; Dataset 3) enriched for distinct gene ontologies (*Figure 6C*). Analysis of the signaling pathways implicated in the hair cycle control revealed a number of differentially expressed WNT and BMP pathway ligands and antagonists (*Figure 6D*). Compared to dorsal skin, ear skin is enriched for transcripts for several WNT antagonists, including *Dkkl1*, *Dkk2* and *Sfrp2*, as well as collagen *Col17a1*, implicated in HF stem cell maintenance (*Matsumura et al., 2016*). Cartilage/muscle complex is prominently enriched for *Bmp5*, and multiple WNT antagonists, including *Frzb, Sfrp2, Sfrp5* and *Wif1*. Additionally, it showed upregulated expression of other known hair cycle inhibitors *Fgf18* (*Kimura-Ueki et al., 2012*; *Leishman et al., 2013*) and *Ctgf* (*Liu and Leask, 2013*).

We validated WNT and BMP changes from RNA-seq by studying relevant pathway reporters and measuring changes in ear hair cycling in mutant mouse models. Using *Axin2-lacZ* reporter mice, we show isolated sites of WNT activity in ear skin dermis, and a lack of activity in telogen HFs as well as in the cartilage and muscle (*Figure 6G and G'*). Using *BRE-gal* reporter mice, we show high levels of BMP activity in telogen ear HFs (in the bulge), as well as in the cartilage and muscle (*Figure 6F and F'*). Overexpression of the BMP antagonist Noggin in *Krt14-Noggin* mice partially rescued the hyper-refractory state – substantially more spontaneous anagen HFs can be found in *Krt14-Noggin* ears as compared to wild-type control (*Figure 6H and H'*; *Appendix 1—figure 28B, D*). Wnt7a overexpression in *Krt14-Wnt7a* mice also reactivated anagen in ear skin, albeit to a lesser extent compared to Noggin overexpression (*Figure 6H''*; *Appendix 1—figure 28C, D*). Together, these results support that hyper-refractivity of ear HFs depends on higher levels of BMP ligands and WNT antagonists, in part produced by the cartilage/muscle complex (*Figure 6E*).

## Hair growth waves distort around hyper-refractory and hairless skin regions

Lastly, our model predicts that hair growth waves can form distorted patterns around non-propagating skin regions, such as hyper-refractory hair-bearing skin or hairless skin (*Figure 7A and B*; Appendix 2-Hyper-refractory domain and Wave breaker; *Appendix 2—figure 17*; *Appendix 2—video 5*). We considered that pattern distortion could occur in the cranial skin at the boundaries with hyper-refractory ears and eyelids – naturally occurring physical breaks in the skin. Indeed, we observe that hair growth waves prominently break around the eyelids and ears – anagen waves propagate faster through the hair-bearing skin around the eyelids and ears, and then distort into the spaces in front

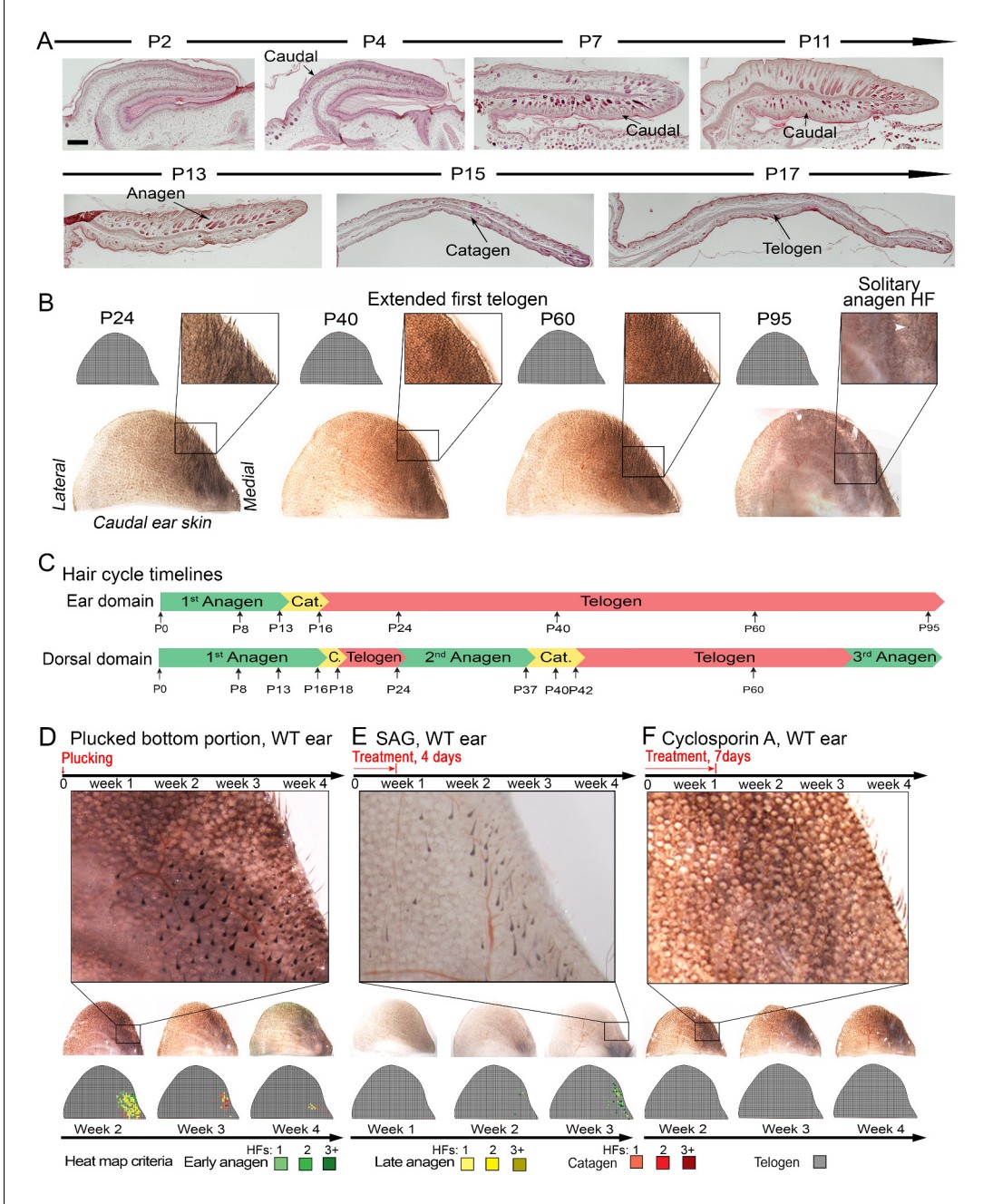

**Figure 5.** Ear skin shows hyper-refractory properties with telogen arrested HFs. (A–C) Morphogenesis and physiological cycling of ear HFs. (A) Analysis of ear tissue histology shows that developing HFs first appear on day P4, and progress toward mature anagen by P7. They enter catagen around P15 and first telogen by P17 (based on three mice for each time point). (B, C) Whole mount ear skin analyses show that ear HFs fail to enter second coordinated anagen and, instead, remain in an extended telogen. Seldom, isolated anagen HFs can be found (see P95 sample on B). Data are based on three mice for each time point. (D) HFs along medial side of the ear re-enter anagen after plucking (also see *Appendix 1—figure 25*). Experiment is based on five mice for each time point analyzed. Representative ear skin image and accompanying heatmap is shown. Heatmap criteria are shown at the bottom. (E) Unlike in dorsal skin (see *Appendix 1—figure 27A*), ear HFs poorly respond to topical SAG treatment. Anagen induction is limited to the medial edge of the ear. (F) Unlike in dorsal skin (see *Appendix 1—figure 27B*), ear HFs fail to re-enter anagen in response to topical cyclosporin A treatment. Experiments on E and F are based on three mice for each time point analyzed. Representative ear skin images and accompanying heatmaps are shown. Scale bars: A – 100 um.

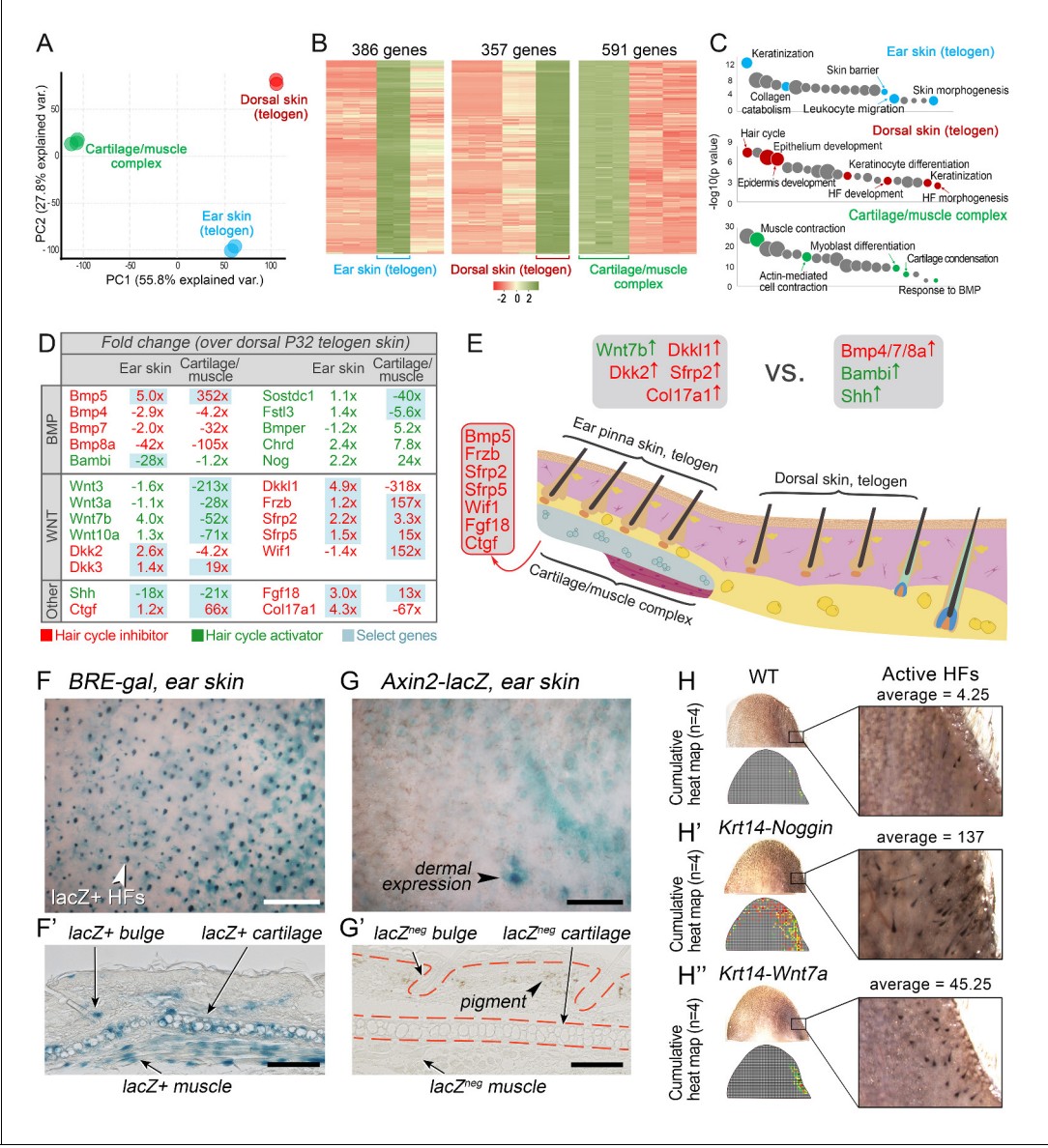

**Figure 6.** WNT and BMP signalings modulate ear HF hyper-refractory state. (A–C) Transcriptomes of first telogen ear skin, first telogen dorsal skin and ear cartilage/muscle complex are distinct, as revealed by PCA analysis (A). They contain 1334 differentially expressed genes (B), spanning distinct gene ontologies (C). (D, E) These tissues show differential expression of multiple ligands and antagonists for several major signaling pathways, prominently WNT and BMP. Putative activators are in green and putative inhibitors are in red. For each gene, relative fold changes for ear skin over dorsal skin and cartilage/muscle complex over dorsal skin expression levels are indicated. Select genes are highlighted. (F, F') *BRE-gal* reporter reveals high BMP activity in telogen ear HFs and in the adjacent cartilage/muscle complex (n = 8). (G, G') *Axin2-lacZ* reporter reveals near absence of WNT activity in ear HFs and cartilage/muscle complex. Seldom, sites of dermal reporter activity can be found (n = 8). (H–H'') Compared to wild type mice (n = 4) (H), ears of *Krt14-Noggin* (n = 4) (H') and *Krt14-Wnt7a* mice (n = 4) (H'') show prominent increases in spontaneous anagen frequency. Cumulative heatmaps from four individual ear samples are shown. Also see *Appendix 1—figure 28*. Scale bars: F, G – 500 um; F', G' – 100 um.

of these anatomical structures (*Figure 7C and E*). Similar patterns are also observed for the ventral-to-dorsal hair growth wave around the limbs (*Figure 7D*). We conclude that distortions of hair growth waves around anatomical structures with temporary or permanent non-propagating properties contribute to rapid body-wide hair growth pattern evolution.

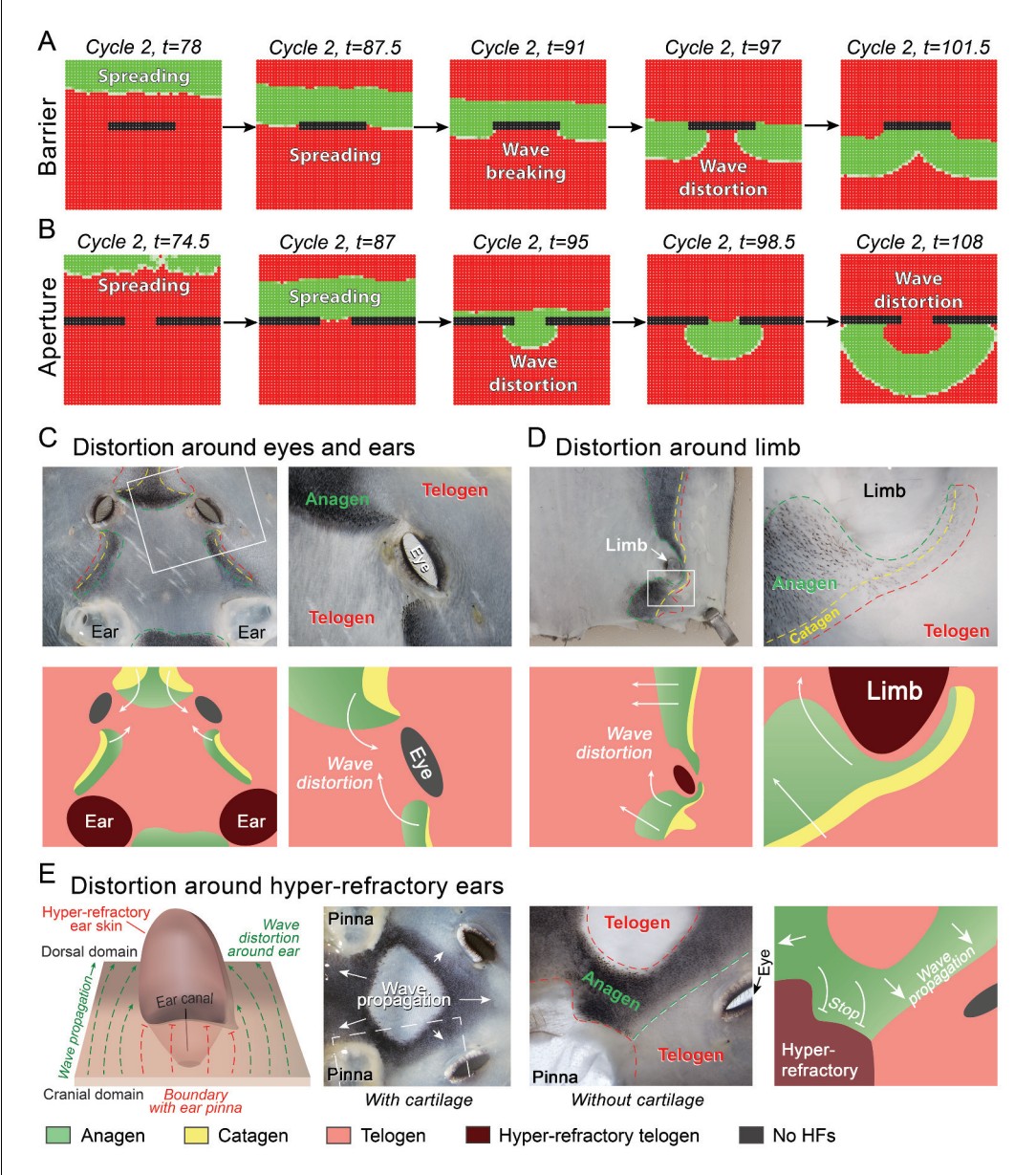

**Figure 7.** Hair growth waves distort at the non-propagating boundaries. (**A, B**) Introduction of a non-propagating barrier (**A**) or an aperture (**B**) into the model produces simulations with distorted anagen spreading wave front (green). (**C, E**) Distortions in the geometry of hair growth waves are commonly seen in the head region at the boundaries with the hyper-refractory ears and eyelids, the physical breaks in the skin. Seldom, similar distorted patterns can be seen around limb skin (**D**). Hair growth patterns on C-E are accompanied by color-coded schematic drawings. Colors are defined at the bottom. Hair growth distortion patterns shown were documented in ten mice each.

## Discussion

### Growth-regulated parallel signaling makes the HF an excitable medium

Previous mathematical models have recapitulated cycling of a single HF (*Al-Nuaimi et al., 2012*; *Halloy et al., 2000*) or in HF populations in two dimensions (*Murray et al., 2012*; *Plikus et al., 2011*). Here, we developed a multiscale model where coupling of activator and inhibitor signals with the movements of a HF in a three-dimensional space simulates cyclic growth and communication between neighboring HFs. In a single HF regime, our model faithfully predicts the effects that changes in WNT and BMP signaling can exert on the length of the anagen phase of the hair cycle.

Similar to the FHN generic excitable media model (*Murray et al., 2012*), our model also recapitulates several known population-level features of the HF system such as spontaneous hair growth initiation and hair wave spreading. Importantly, however, only our model allows incorporation of differential HF growth in space, a feature required for simulating heterogeneous skin properties such as interactions between skin domains with different hair cycle frequencies or the hair wave distortion effect. Thus, while the multiscale nature and non-linearity make our model more difficult to derive analytical results, its heterogeneous domain feature allows studying complex skin-wide hair growth dynamics (see Appendix 2-Comparison with FitzHugh-Nagumo (FHN) model).

## HF morphogenesis across mouse skin is spatially asynchronous

Hair growth in newborn mice is commonly thought to occur simultaneously across the entire skin. In fact, we show that the first cycle is already distinctly patterned: at birth, anagen HFs in dorsal skin have head-to-tail and lateral-to-medial asynchronies, while first anagen entry by ventral HFs is delayed by approximately 3 days and proceeds as a concentric lateral-to-midline wave. Similarly delayed by 6 days are ear HFs. First anagen naturally follows the process of HF morphogenesis, which is known to be temporally asynchronous, and to occur, at least in the dorsal skin, in three successive waves (reviewed in *Clavel et al., 2012*). Pattern-wise, development of HFs relies on reaction-diffusion (*Sick et al., 2006*) and on space-filling expansion-induction mechanisms (*Cheng et al., 2014*). Importantly, models for both mechanisms assume spatially synchronous HF morphogenesis. Our findings of spatial asynchrony of the first anagen indicate spatial asynchrony of HF morphogenesis. Future studies will be required to understand the modeling and signaling aspects of such phenomenon.

## Hair cycle patterns evolve from the interaction of heterogeneous skin domains

Our data reveal prominent regional differences in hair cycle dynamics and show that interaction between HFs across domain boundaries drives rapid evolution of complex hair growth patterns. Specifically, we show that during early postnatal cycles, chin and ventral domains become the dominant sources of skin-wide anagen waves. Such dominant behavior of chin and ventral domains is accompanied by distinct activity dynamics for WNT and BMP, putative hair cycle activators and inhibitors, respectively. Transgenic mouse studies further confirm the functional importance of differential WNT and BMP activities in setting distinct hair growth pace across discrete anatomical skin regions. Admittedly, an in-depth follow-up study will be necessary to identify and verify the major site-specific cellular sources for WNT and BMP ligands and antagonists.

We also show that ear skin behaves as a hyper-refractory domain, where telogen HFs are resistant to anagen-inducing stimuli and cannot participate in hair growth wave propagation. We reveal that such hyper-refractivity relates to high levels of BMP ligands and WNT antagonists, in part produced by the cartilage/muscle complex, a structure unique to the ear skin. Thus, novel behaviors can be produced by the cooption of signals from new tissue modules, rather than by the modification of preexisting ones. This finding parallels the modulatory effects of non-HF cell types on the dorsal skin hair cycle, including adipose progenitors (*Festa et al., 2011*; *Rivera-Gonzalez et al., 2016*), mature adipocytes (*Plikus et al., 2008b*), and resident macrophages (*Castellana et al., 2014*; *Chen et al., 2015*). Finally, we show that anatomically defined structures that cannot propagate hair growth waves, namely ears and eyelids, can generate a 'wave-breaker' effect. Similar distortion effects are likely to occur around other anatomical structures, such as the tail and genitals, and around skin defects, such as scars, and can jointly contribute to rapid hair growth pattern evolution.

Taken together, our study reveals that the skin as a whole functions as a complex regenerative landscape with regions of fast, slow, and very slow hair renewal (*Appendix 1—figure 29*). We show that this behavior produces a fur coat with variable hair density, which likely serves an adaptive role, such as in thermoregulation. Mechanistically, we show that the WNT/BMP activator/inhibitor signaling pair modulates hair regeneration in all skin regions studied. This suggests that the WNT/BMP 'molecular language' for hair growth is general, rather than a special case for a specific body site. Its generality allows for hair-to-hair communications to arise across anatomic domain boundaries, which, in turn, enables novel hair growth dynamics not obvious from prior work – fast cycling skin regions (such as chin skin) function as a kind of hair growth pacemaker. Furthermore, our findings on ear hair

cycle expand the repertoire of tissues with signaling macro-environment function to include any closely-positioned anatomic structures with signaling properties, such as cartilage.

We posit that some of the newly found hair regeneration features can have analogs in other organs. For instance, dominant anatomically defined pacemakers are common in the electrically coupled muscle-based tissues, including heart and stomach, where they generate directional contractile rhythmicity. Other actively regenerating organs, such as the intestines and bone marrow, can likely contain anatomic regions of faster and slower regeneration and, conceivably, they can be coupled to work in coordination. Knowledge learned from the skin system in the current study can guide the search for regenerative landscapes in these and other organs. Because coordination principles observed in the skin may be universal, the likelihood of them operating in other organs is substantial despite prominent anatomical differences.

## Materials and methods

### Computational modeling

The modeling framework is based on a hybrid approach, with individual HFs modeled as an expanding or contracting one-dimensional line and with the diffusive molecules described in reaction-diffusion equations (Appendix 2, *Equations 1–4*). The latter are solved using a finite difference scheme with the standard central difference approximation on the diffusion (see Appendix 2-1-dimensional (1D) HF model to Numerical methods in Appendix 2).

### Experimental mouse models

*Krt14-Noggin* (*Plikus et al., 2004*, *2005*), *Krt14-Bmp4* (*Guha et al., 2004*), *Krt14-Cre;Wnt7b^{fl/fl}* (*Kandyba and Kobielak, 2014*), *Krt14-Wnt7a* (*Plikus et al., 2011*), *Krt5-rtTA;tetO-Dkk1* (*Chu et al., 2004*), *220bpMsx2-hsplacZ* reporter (*Brugger et al., 2004*), *BRE(Bmp response element)-gal* BMP reporter (*Javier et al., 2012*), *Axin2-lacZ* (*Lustig et al., 2002*), and *Flash* WNT reporter mice (*Hodgson et al., 2014*) were used. For Dkk1 induction, P30 *Krt5-rtTA;tetO-Dkk1* mice were placed on 2 mg/ml Doxycycline-containing water *ab libitum*, and skin was collected at P44 for histology and at P50 for hair length measurements.

### Skin grafting

5 × 5 mm skin grafts from chin and dorsal domains of P21 C57BL/6J male mice were transplanted onto the dorsum of gender-matched pigmented P50 SCID recipients. At the time of grafting, donor skin was in first telogen and recipient skin was in second telogen.

### Hair plucking

In dorsal skin, club hairs were plucked from 5 × 5 mm areas. In the ear pinna, plucking was done on the caudal skin. For quantitative plucking, approximately 500 club hairs we plucked along the medial ear side.

### Topical drug treatment

Cyclosporin A: for the dorsal skin, 100 ul of Cyclosporin A solution (1, 5, and 10 mg/ml) was applied topically once a day for 7 days. For the ear pinna, caudal skin was treated with 100 ul of 10 mg/ml of Cyclosporin A once a day for 7 days. Smoothened agonist (SAG): for the dorsal skin, 120 uM of SAG in DMSO/acetone was applied topically once a day for 4 days as described (*Paladini et al., 2005*). For the ear pinna, caudal skin was treated with 25 ul of SAG solution once a day for 4 days.

### Hair length measurements and club hair counting

Guard, awl, auchene and zigzag club hair types were photographed, traced and calibrated using Adobe Illustrator software. See *Appendix 1—table 1*. Club hair density was evaluated on whole-mount telogen skin samples that were pre-treated with 1 mg/mL Collagenase/Dispase and counter-stained with hematoxylin.

## Histology and immunohistochemistry

Histology was performed on 4% PFA-fixed sections. For *BRE-gal* and *Axin2-lacZ* specimens, whole mount lacZ staining was performed first followed by histology. The primary antibodies used were rabbit anti-keratin Krt5 (1:250, Abcam, UK), rabbit anti-perilipin (1:750; Cell Signaling), rabbit anti-α SMA (1:200; Abcam). Actin was detected with phalloidin (Alexa Fluor 488; Molecular Probes).

## Whole mount in vivo bioluminescence imaging

Whole body imaging of *Flash* mice was performed as previously described (*Hodgson et al., 2014*). Briefly, mice were injected with 150 mg/kg of firefly D-luciferin substrate and imaged with the Xenogen IVIS Spectrum system.

## FACS and analysis

Second telogen skin from C57BL/6J male mice was treated with Dispase to separate epidermis from dermis. Epidermis was digested with Accutase and dermis with Collagenase. Epidermal and dermal cell suspensions were combined and stained with anti-CD11b (eBioscience) and anti-F4/80 antibodies (eBioscience). Due to small tissue size, chin skin cells from three mice were combined for each experiment. FACS data were analyzed using FlowJo.

## RNA-sequencing and analyses

Total RNA was isolated using the RNeasy Mini Kit (Qiagen). RNA samples with RIN >8.0 were considered for cDNA library preparation. Full-length cDNA library amplification and tagmentation was performed as previously described (*Picelli et al., 2014*). Libraries were multiplexed and sequenced as paired-end on an Illumina Next-Seq500 platform. Paired-end reads were aligned to the mouse genome (mm10/gencode.vM8) and quantified using the RNA-seq by Expectation-Maximization algorithm (RSEM) with standard parameters (version 1.2.25) (*Li and Dewey, 2011*). Samples were batch-effect corrected. EdgeR (version 3.14.0) was employed to identify differentially expressed genes (DEGs) across samples of interest. FPKM values were taken as inputs for PCA analysis and DEG analyses. Data is available at GEO: GSE85039.

## Acknowledgements

MVP is supported by the NIH NIAMS grants R01-AR067273, R01-AR069653, Edward Mallinckrodt Jr. Foundation grant and Pew Charitable Trust grant. QN is supported by NSF grants: DMS 1161621, and DMS 1562176 and NIH grants: P50-GM076516, R01-GM107264, and R01-NS095355. JWO is supported by the National Research Foundation of Korea (NRF) grant funded by the Korea government (MSIP) (2016R1C1B1015211 and 2014R1A5A2010008). KK (Kobielak) is supported by NIH grants R01-AR061552 and National Science Centre, Poland, OPUS Grant #2015/19/B/NZ3/02948. KK (Khosrotehrani) is supported by the NHMRC Career Development Fellowship 1023371. HLL is supported by NIH NCI T32 training grant (T32-CA009054). CFGJ is supported by NSF-GRFP (DGE-1321846) and MBRS-IMSD training grant (GM055246). ZY is supported by the programs of the Major Project for Cultivation Technology of New Varieties of Genetically Modified Organisms of the Ministry of Agriculture (2014ZX08008001, 2013ZX08008-001); Beijing Nature Foundation Grant (5162018); State Key Laboratory Open Project Grant (2015SKLB6-16). BA is supported by the NIH NIAMS grants R01-AR056439. Authors thank Dr. John A. Kessler for *Krt14-Bmp4* mice, Dr. Sarah E. Millar for *Krt5-rtTA;tetO-Dkk1* mice, Hoang Ha, Buu Le Dao, Andy Lau, Kathleen Nguyen, Hyeon Sung Lee and Manda Nguyen for technical assistance.

## Additional information

### Funding

| Funder | Grant reference number | Author |
|---|---|---|
| National Research Foundation of Korea | 2016R1C1B1015211 | Ji Won Oh |
| National Research Foundation | 2014R1A5A2010008 | Ji Won Oh |

of Korea

| | | |
|---|---|---|
| National Cancer Institute | T32-CA009054 | Hye-Lim Lee |
| National Science Foundation | DGE-1321846 | Christian Fernando Guerrero-Juarez |
| National Institute of General Medical Sciences | GM055246 | Christian Fernando Guerrero-Juarez |
| National Institute of Arthritis and Musculoskeletal and Skin Diseases | R01-AR061552 | Krzysztof Kobielak |
| National Institute of Arthritis and Musculoskeletal and Skin Diseases | R01-AR056439 | Bogi Andersen |
| National Health and Medical Research Council | 1023371 | Kiarash Khosrotehrani |
| National Science Foundation | DMS 1161621 | Qing Nie |
| National Science Foundation | DMS 1562176 | Qing Nie |
| National Institute of General Medical Sciences | P50-GM076516 | Qing Nie |
| National Institute of General Medical Sciences | R01-GM107264 | Qing Nie |
| National Institute of Neurological Disorders and Stroke | R01-NS095355 | Qing Nie |
| National Institute of Arthritis and Musculoskeletal and Skin Diseases | R01-AR067273 | Maksim V Plikus |
| Pew Charitable Trusts | 00029641 | Maksim V Plikus |
| NIH NIAMS | R01-AR069653 | Maksim V Plikus |

The funders had no role in study design, data collection and interpretation, or the decision to submit the work for publication.

### Author contributions

QW, Conceptualization, Data curation, Software, Formal analysis, Validation, Investigation, Methodology, Writing—original draft, Writing—review and editing; JWO, Conceptualization, Data curation, Validation, Investigation, Methodology, Writing—original draft, Writing—review and editing; H-LL, AD, RR, CFG-J, XW, RZ, XC, JL, XW, Investigation, Methodology; TP, Data curation, Software, Validation, Methodology; MAF, SCJ, ARR, BV, NMM, JMP, J-HC, HL, EK, Investigation; KP, Visualization; JMDL, Conceptualization, Investigation, Methodology; JCK, MK, JF, KKo, Writing—original draft; ZY, BA, Writing—original draft, Writing—review and editing; KKh, Conceptualization, Investigation, Methodology, Writing—original draft, Project administration, Writing—review and editing; QN, Conceptualization, Resources, Data curation, Software, Formal analysis, Supervision, Funding acquisition, Validation, Writing—original draft, Project administration, Writing—review and editing; MVP, Conceptualization, Supervision, Funding acquisition, Investigation, Writing—original draft, Project administration, Writing—review and editing

### Author ORCIDs

Qixuan Wang, http://orcid.org/0000-0003-2673-921X
Ji Won Oh, http://orcid.org/0000-0001-5742-5120
Eve Kandyba, http://orcid.org/0000-0002-9219-5284
Zhengquan Yu, http://orcid.org/0000-0001-8696-2013
Bogi Andersen, http://orcid.org/0000-0001-7181-2768
Maksim V Plikus, http://orcid.org/0000-0002-8845-2559

### Ethics

Animal experimentation: This study was performed in strict accordance with the recommendations in the Guide for the Care and Use of Laboratory Animals of the National Institutes of Health. All of

the animals were handled according to approved institutional animal care and use committee (IACUC) protocols (#2012-3054 and #2013-3081) of the University of California, Irvine.

## Additional files

### Supplementary files

• Supplementary file 1. Dataset 1: Putative activator genes (tabs #1, #2) and putative inhibitor genes (tabs #3, #4) available from a whole skin microarray dataset.

• Supplementary file 2. Dataset 2: Putative activator and inhibitor genes displaying domain-specific expression patterns at all hair cycle time points on whole skin RNA-seq profiling. Genes are grouped into individual tabs based on (i) their activator or inhibitor expression profile and (ii) their specificity to one or several skin domains.

• Supplementary file 3. Dataset 3: Differentially expressed genes specific to refractory telogen dorsal skin, telogen ear skin and cartilage/muscle complex.

### Major datasets

The following dataset was generated:

| Author(s) | Year | Dataset title | Dataset URL | Database, license, and accessibility information |
|---|---|---|---|---|
| Plikus MV, Nie Q, Oh JW, Wang Q, Lee H, Peng T | 2016 | A multi-scale model for hair follicle reveals heterogeneous skin domains drive rapid spatiotemporal hair growth patterning | https://www.ncbi.nlm.nih.gov/geo/query/acc.cgi?acc=GSE85039 | Publicly available at the NCBI Gene Expression Omnibus (accession no: GSE85039) |

The following previously published dataset was used:

| Author(s) | Year | Dataset title | Dataset URL | Database, license, and accessibility information |
|---|---|---|---|---|
| Lin KK, Kumar V, Geyfman M, Chudova D, Ihler AT, Smyth P, Paus R, Takahashi JS, Andersen B | 2009 | Expression profiling of mouse dorsal skin during hair follicle cycling | https://www.ncbi.nlm.nih.gov/geo/query/acc.cgi?acc=GSE11186 | Publicly available at the NCBI Gene Expression Omnibus (accession no: GSE11186) |

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

## Appendix 1

# Experimental: A multi-scale model for hair follicles reveals heterogeneous domains driving rapid spatiotemporal hair growth patterning

## Identifying model predicted hair cycle activators and inhibitors

To find out which signaling pathway activities fit the temporal dynamics of activator and inhibitor predicted by the model (*Figure 1D and E*), we examined a highly temporally resolved whole-skin microarray dataset, which includes nine consecutive time points: five for anagen, three for catagen, and one for telogen (*Lin et al., 2009*). We identified two sets of 236 and 122 genes whose temporal dynamics recapitulate those of the simulated activator (*Figure 1D' and D''*) and inhibitor signals (*Figure 1E' and E''*; Dataset S1), respectively. Focusing on major signaling pathways, we show that multiple members of WNT and BMP are represented in each gene set. The putative activator set includes WNT ligands (*Wnt2, Wnt9a, Wnt7b*), soluble WNT activator *r-Spondin1*, multiple WNT-specific Tcf transcription factors, and soluble BMP antagonists, *Follistatin* and *Sostdc1*. The putative inhibitor set includes multiple BMP ligands (*Bmp2, Bmp4, Bmp6, Bmp7*), BMP receptors (*Bmpr1a, Bmpr1b* and *Bmpr2*), BMP-specific transcription factors (*Smad2, Smad5* and *Smad7*), and soluble (*Wif1, Sfrp1*), transmembrane (*Kremen1*), and intracellular WNT antagonists (*Nkd2, Prickle1*).

## Validating model-predicted roles for BMP signaling in hair cycle control

For BMP signaling, we examined *Krt14-Bmp4* (*Guha et al., 2004*) and *Krt14-Noggin* mice (*Plikus et al., 2004*, *2005*) overexpressing BMP ligand and soluble antagonist, respectively. Consistent with the notion of BMP acting as anagen inhibitor, *Krt14-Bmp4* mice show shortened pelage, prominently on their ears, tail, and paws, and partial baldness on the trunk (*Appendix 1—figure 1A*, *2*). Fully grown dorsal hairs in *Krt14-Bmp4* are significantly shorter compared to control across all hair types by 35–47% (p<0.05) (*Appendix 1—figure 1B*). This is accompanied by the shortened anagen duration as established by histology on day P15 (*Appendix 1—figure 3B*). In contrast, *Krt14-Noggin* mice show a general increase in hairiness, prominent on their ears, tail, and paws (*Appendix 1—figure 1A*, *2*), and significantly longer than WT dorsal hairs: 12% longer for guard (p<0.01) and 7% longer for auchene type (p<0.01) (*Appendix 1—figure 1B*). In parallel, we observe longer anagen phase duration as revealed by histology on day P19 (*Appendix 1—figure 3A*). Hair length changes in *Krt14-Noggin* mice were not statistically significant for the zigzag and awl types.

## Validating model-predicted roles for WNT signaling in hair cycle control

For WNT signaling, we examined *Krt14-Cre;Wnt7b^{fl/fl}* (*Kandyba and Kobielak, 2014*) and *Krt5-rtTA;tetO-Dkk1* mice (*Choi et al., 2013*), with constitutive skin-specific deletion of the WNT ligand Wnt7b and conditional overexpression of the soluble WNT antagonist Dkk1, respectively. *Wnt7b* follows model-predicted activator expression pattern (*Figure 1D''*) and consistent with the previous report (*Kandyba and Kobielak, 2014*), Wnt7b-deficient mice display short first anagen (*Appendix 1—figure 5A*) and short pelage (*Appendix 1—figure 4A', B'*). We now show that, intriguingly, during the following hair growth cycle, *Krt14-Cre; Wnt7b^{fl/fl}* mice compensate for the initial defect (*Appendix 1—figure 4A*), and hair length increases between cycles by an average of 22% (p<0.05) (*Appendix 1—figure 4B, B'*). In *Krt5-rtTA;tetO-Dkk1* mice, induction of Dkk1 overexpression during telogen prominently

blocks HFs in an extended telogen (*Choi et al., 2013*) – a scenario equivalent to 'Equilibrium II' in our model (*Figure 1C*). Induction of Dkk1 overexpression during anagen leads to early anagen termination (*Appendix 1—figure 5B*) and shortened hairs (*Appendix 1—figure 4A'', B''*). Compared to control, club hairs in *Krt5-rtTA;tetO-Dkk1* mice across all four types shorten by 22–25% ($p < 0.05$). Taken together, our modeling and experiments confirm that WNT and BMP serve as a core hair cycle activator/inhibitor pair, affecting both telogen and anagen phase timing.

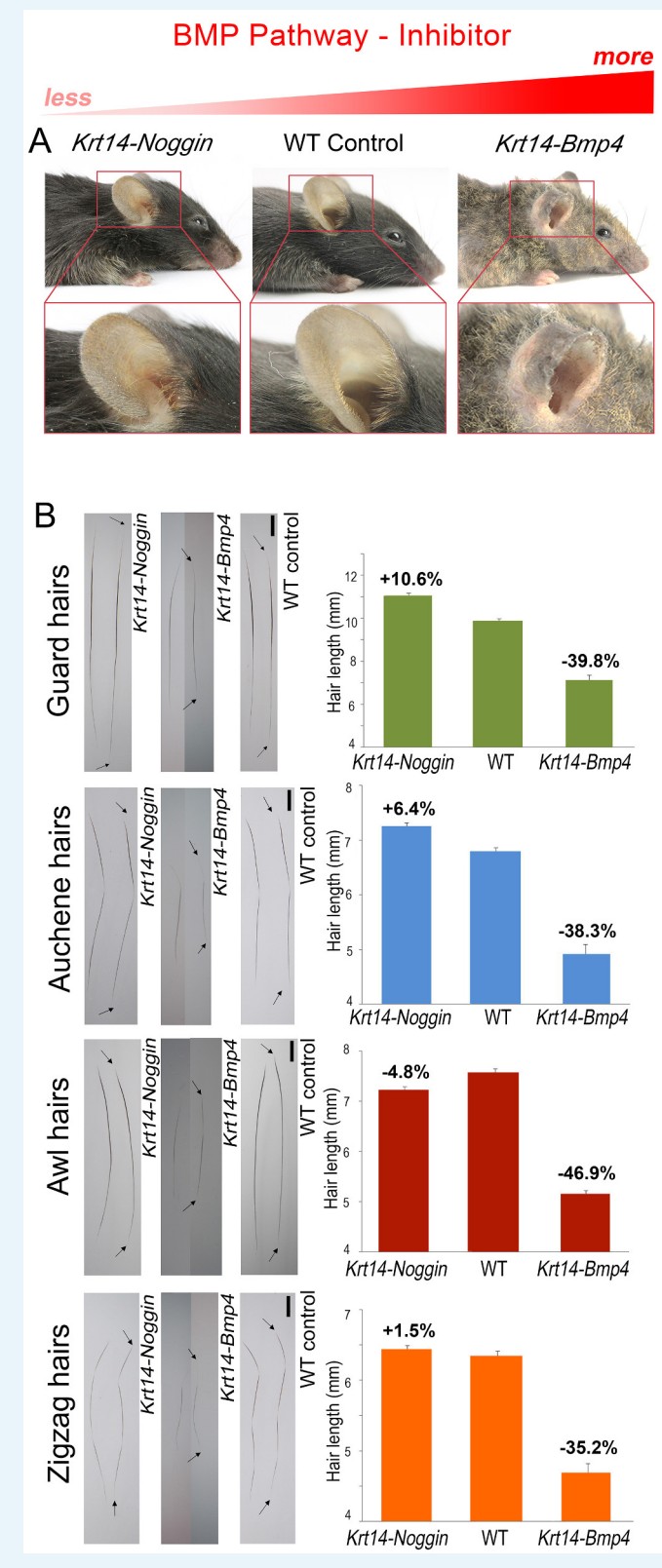

**Appendix 1—figure 1.** BMP changes affect hair length. Grossly, *Krt14-Noggin* mice display longer than normal pelage (**A**). In contrast, *Krt14-Bmp4* mice display generalized pelage shortening. Also see *Appendix 1—figure 2*. (**B**) Compared to control, hairs are longer in *Krt14-Noggin* and shorter in *Krt14-Bmp4* mutants. *Krt14-Noggin* hairs lengthen by up

to ~12%, while *Krt14-Bmp4* hairs shorten by 35–46%. Arrows mark hair ends. Scale bars: B – 1 mm. Images on B are composites.

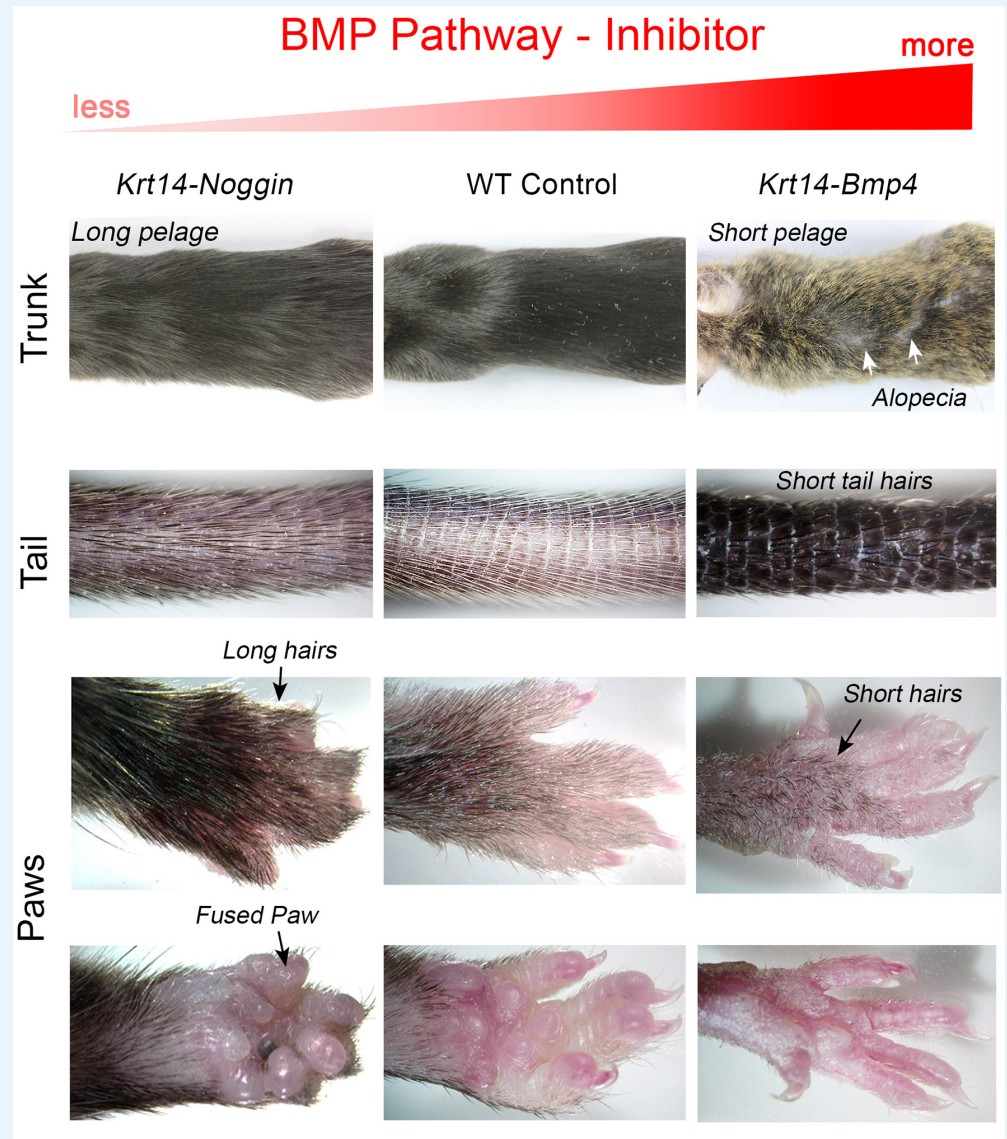

**Appendix 1—figure 2.** Changes in BMP signaling result in morphological pelage length defects. Compared to control, *Krt14-Noggin* mice display generalized increase in pelage length, prominently on the tail and paws. Note fused paws, short digits, lack of claws and polydactyly, phenotypes that were reported previously. *Krt14-Bmp4* mice display visibly short pelage, patches of prominent thinning on the trunk, as well as short hairs on the tail and paws. Also note other paw defects, longer digits and long, curved claws.

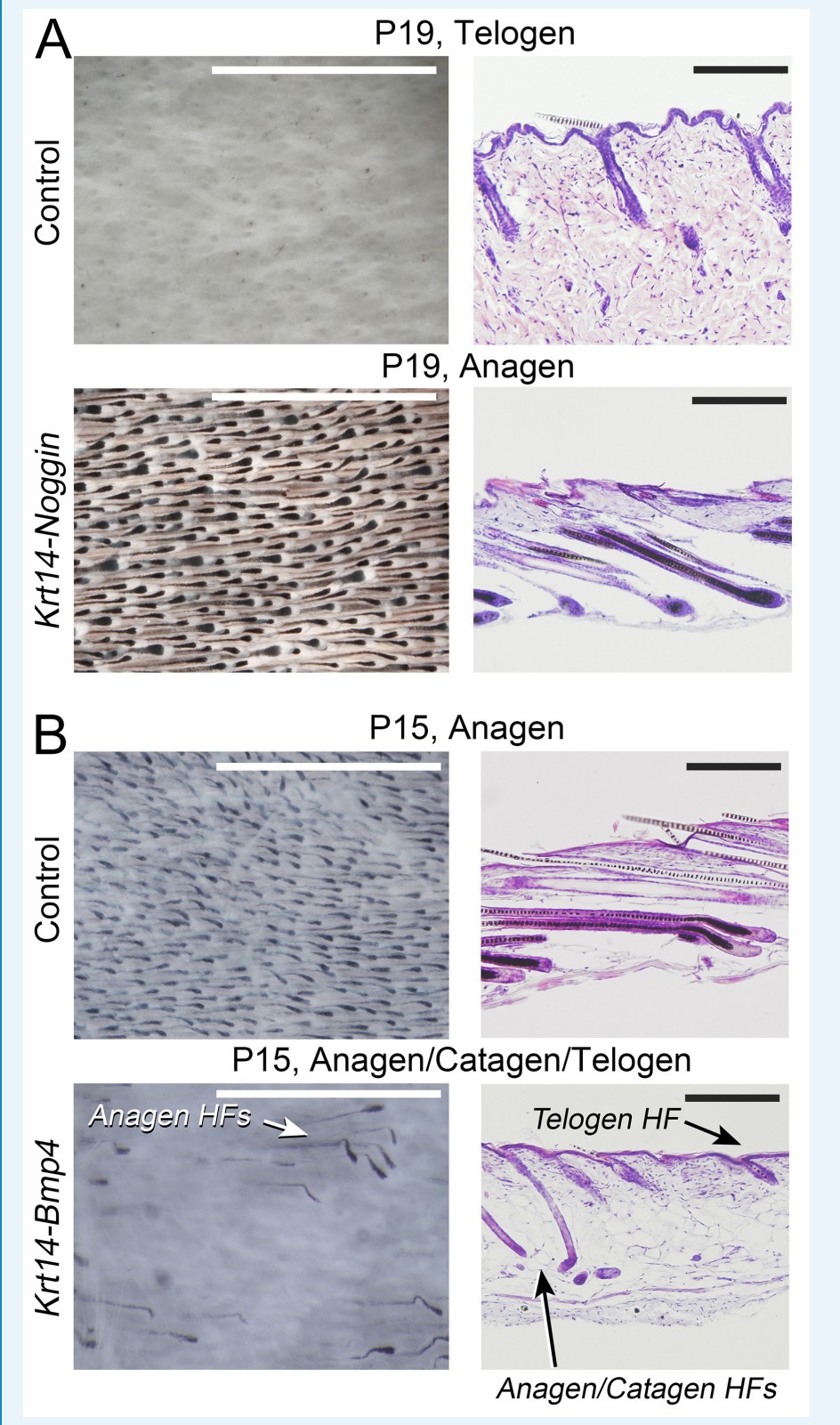

**Appendix 1—figure 3.** BMP modulates anagen phase duration. Repression and activation of inhibitory BMP leads to increased and decreased anagen duration, respectively. Histologically, day P19 *Krt14-Noggin* mice display delayed (**A**), while day P15 *Krt14-Bmp4*

mice show premature anagen termination (**B**). Scale bars: A, B – 1 mm (whole mount) and 200 um (histology).

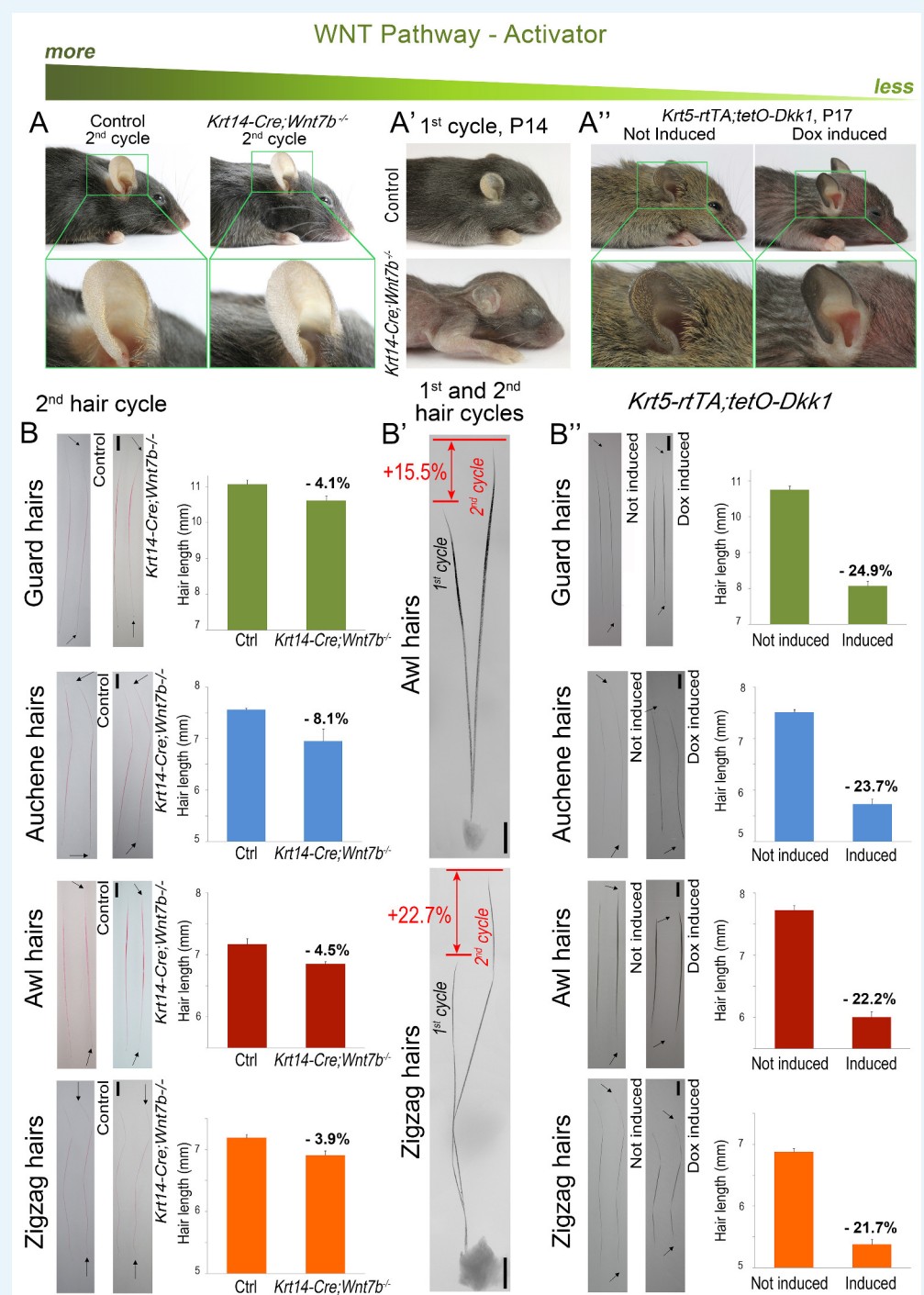

**Appendix 1—figure 4.** Downregulation of WNT leads to hair length shortening. *Krt14-Cre; Wnt7b*[fl/fl] mice display prominent pelage shortening during the first cycle (**A'**), which then becomes largely restored during the second cycle (**A**). Indeed, second cycle hairs in the same mutant HFs are ~15–23% longer than first cycle hairs (**B'**). Note that hairs on **B**) were dyed red. In *Krt5-rtTA;tetO-Dkk1* mice, induction of Dkk1 results in prominent pelage

shortening (**A''**) and individual hair shorten by 22–25% as compared to non-induced control (**B''**). Scale bars: B-B'' – 1 mm. Images on B-B'' are composites.

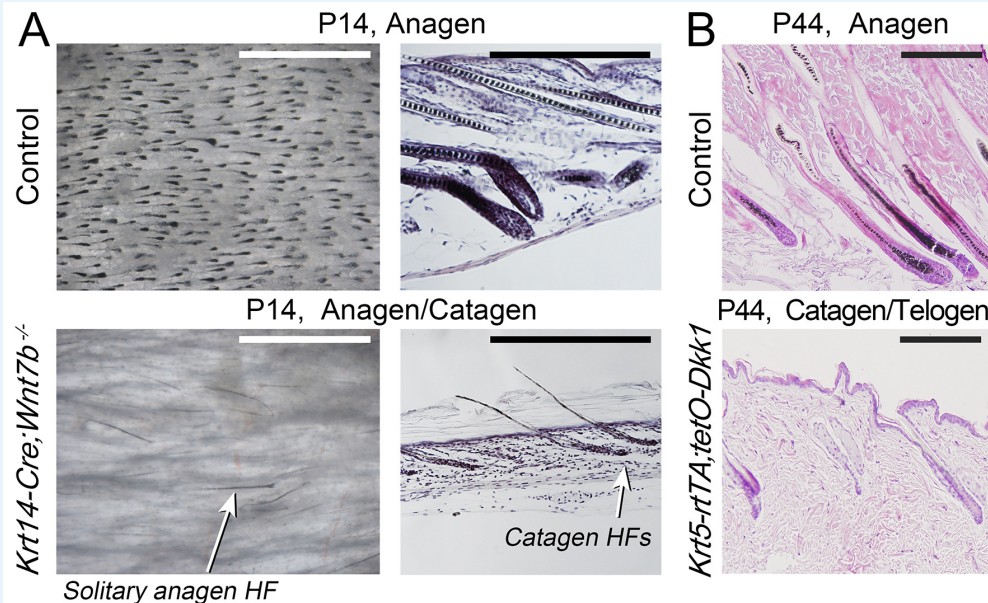

**Appendix 1—figure 5.** Downregulation of WNT leads to decreased anagen phase duration. Histologically, day P14 *Krt14-Cre;Wnt7b$^{fl/fl}$* mice (**A**) and induced day P44 *Krt5-rtTA;tetO-Dkk1* mice (**B**) display premature anagen termination. Scale bars: A – 1 mm (whole mount), A, B – 200 um (histology).

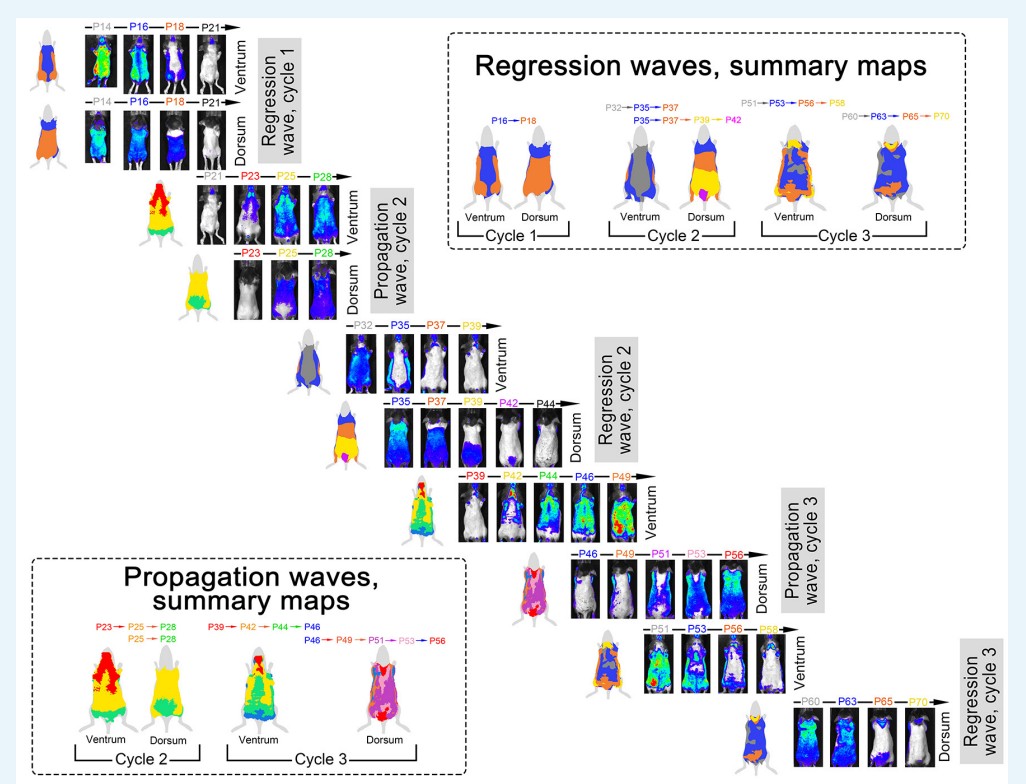

**Appendix 1—figure 6.** Spatial luminescence signal mapping across hair cycles. Spatial maps of the luminescence signal in individual *Flash* mouse across three consecutive hair cycles are shown. Luminescence signal is represented as a 'heatmap' ranging from low (blue) to strong (red). For each time point, both ventral and dorsal views of the animal are provided. Images are grouped into two types of sets: (i) 'propagation waves', which show telogen-to-anagen wave spreading and (ii) 'regression waves', which show anagen-to-catagen-to-telogen wave spreading. During each hair cycle, both propagation and regression waves demonstrate notable temporal phase advancement in the ventral as compared to dorsal skin. Also see main *Figure 3*.

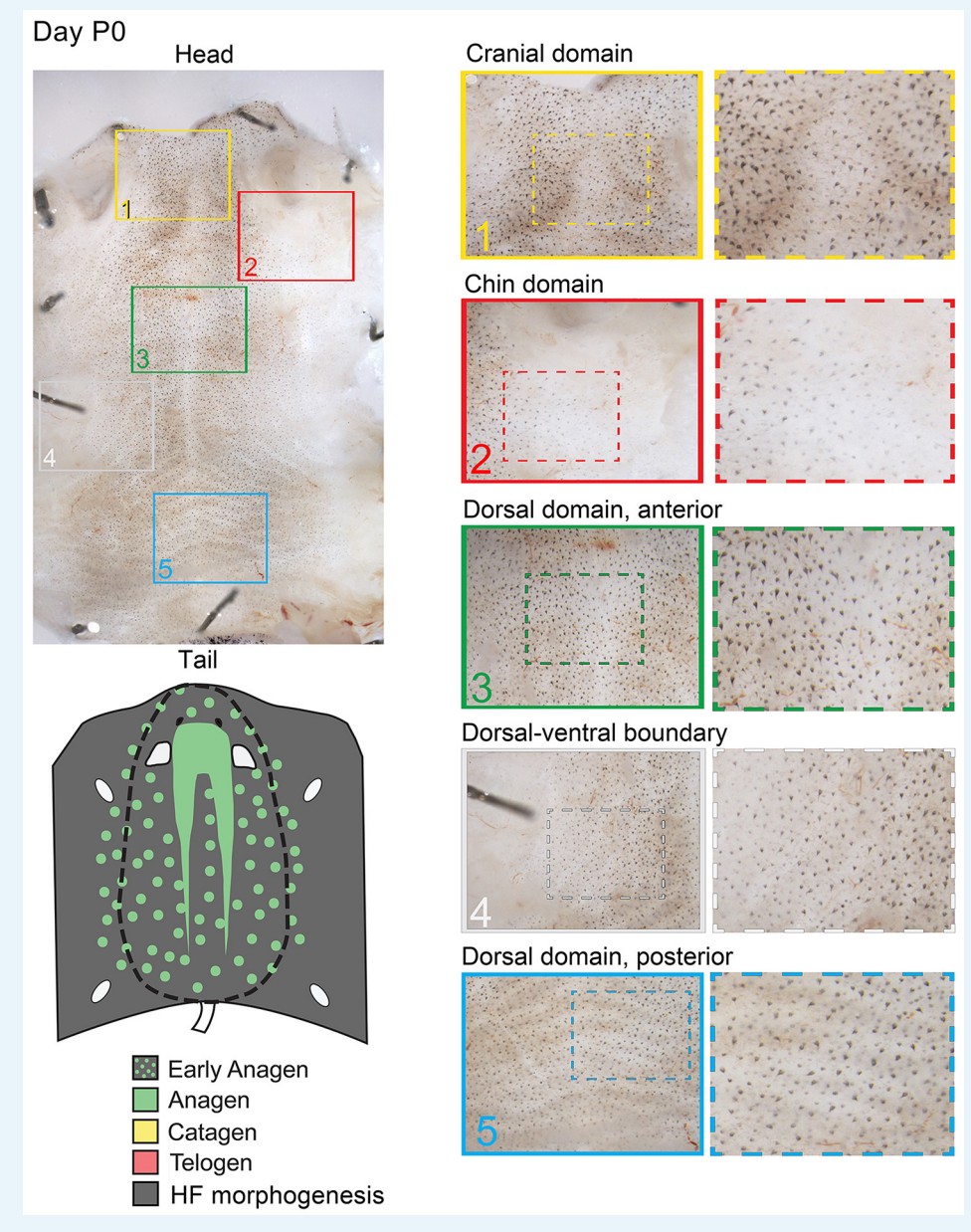

**Appendix 1—figure 7.** Distribution of the hair cycle stages in P0 mouse skin. Day P0 skin displays prominent spatial distribution of anagen HFs (based on macroscopic evaluation of HF pigmentation). Prominent patterning features at this stage include: (**a**) head-to-tail anagen phase advancement in the dorsal skin (compare insert 1 with insert 5) and (**b**) dorsal-to-ventral anagen phase advancement reflective of the temporary delayed HF morphogenesis in the ventral skin (compare insert 2 and 4 with inserts 1 and 3). See main *Figure 2A–2D* for further details.

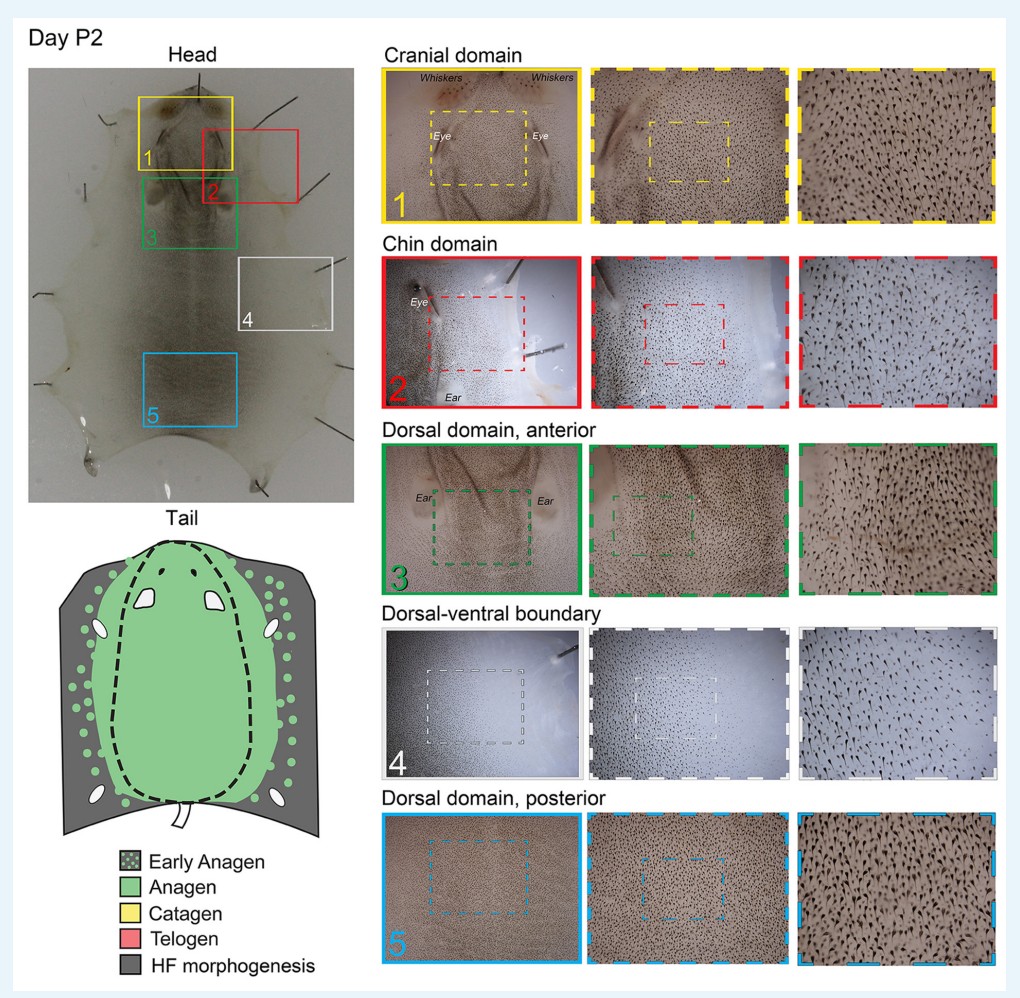

**Appendix 1—figure 8.** Distribution of the hair cycle stages in P2 mouse skin. On day P2, initial head-to-tail hair cycle asynchrony in the dorsal skin becomes less prominent, while dorsal-to-ventral hair cycle asynchrony is maintained (inserts 2 and 4).

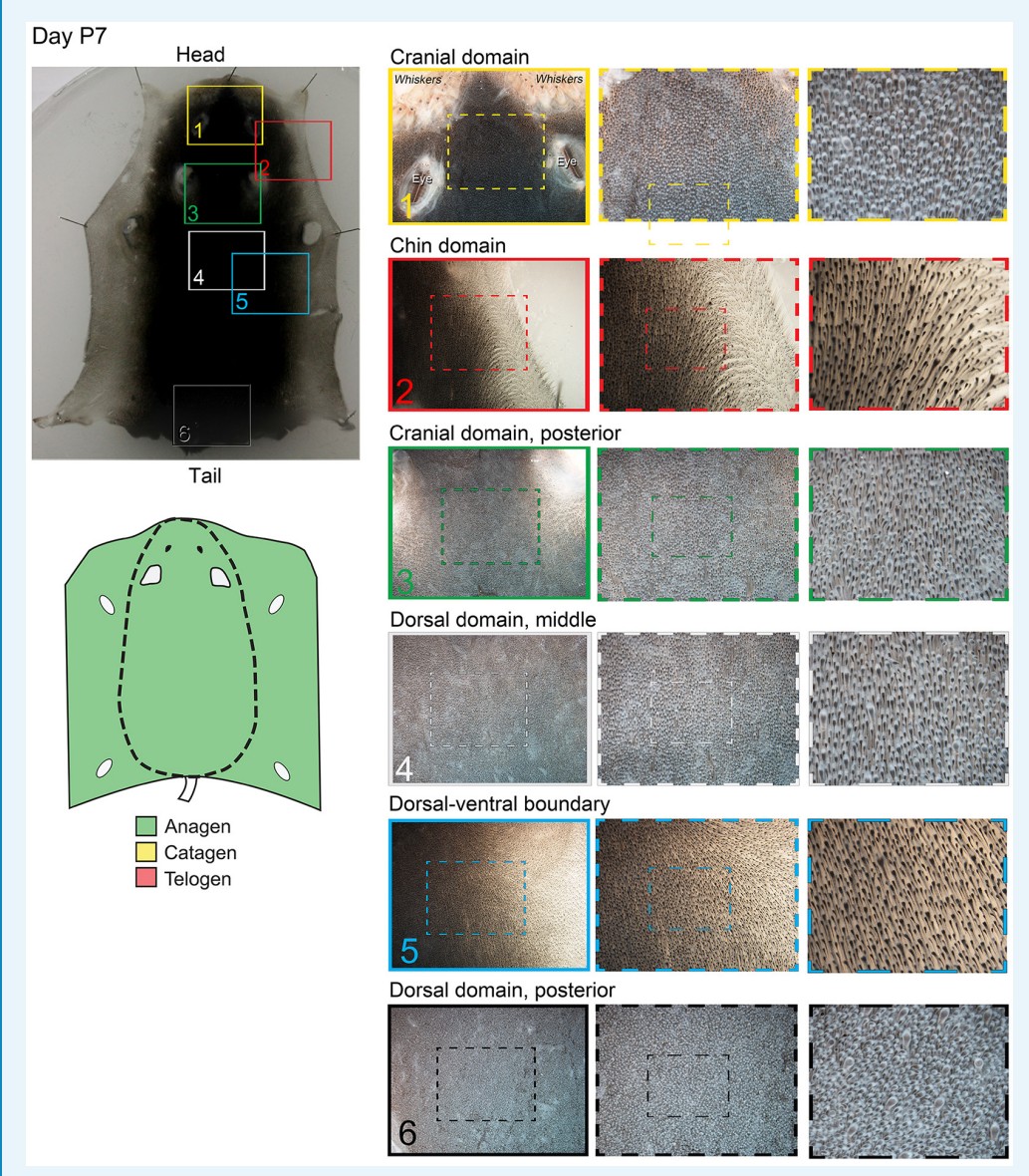

**Appendix 1—figure 9.** Distribution of the hair cycle stages in P7 mouse skin. By day P7, HFs in both dorsal and ventral skin are in mature anagen.

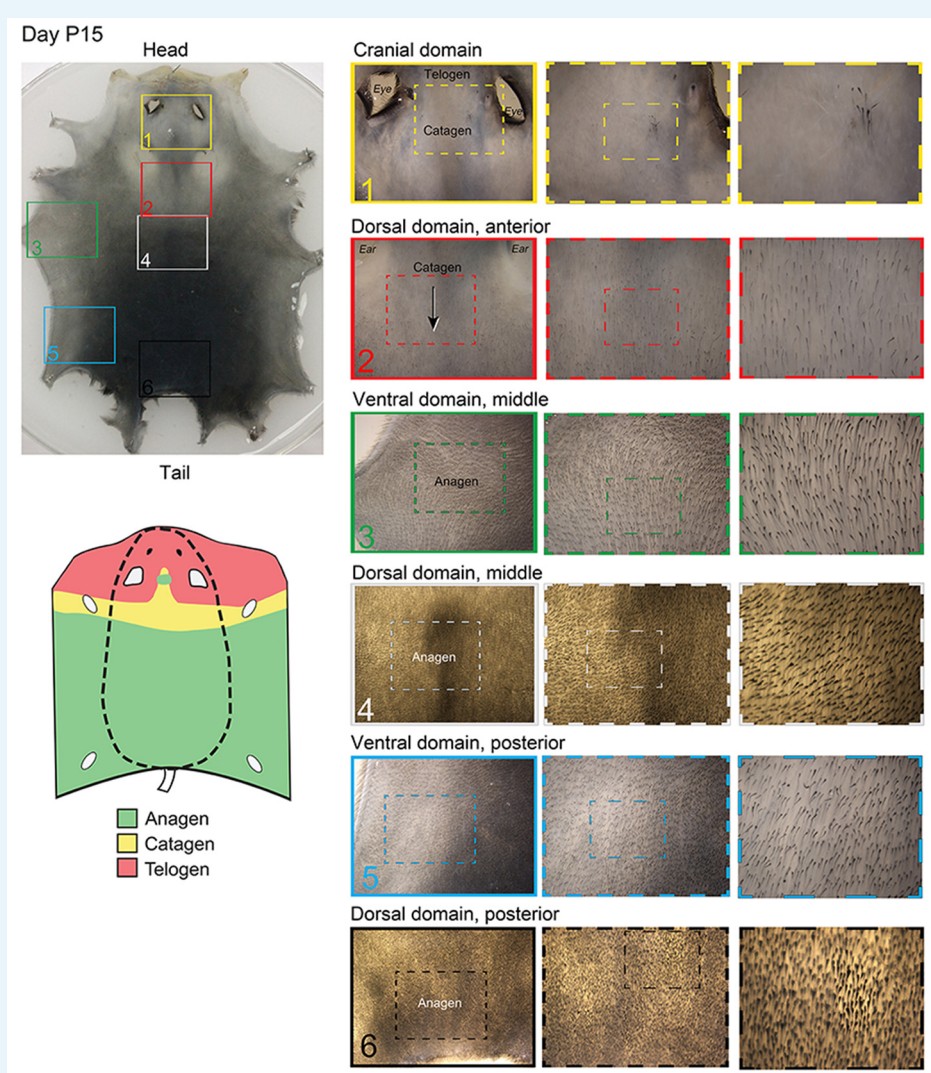

**Appendix 1—figure 10.** Distribution of hair cycle stages in P15 mouse skin. By day P15, HFs in the chin domain and most of the cranial domain undergo catagen-to-telogen transition, whereas HFs in the dorsal and ventral domains are in anagen. At this time, mouse-to-mouse pattern variability starts to become prominent, with more or less (as compared to the example pattern shown here) of the chin and cranial domains having transitioned to telogen.

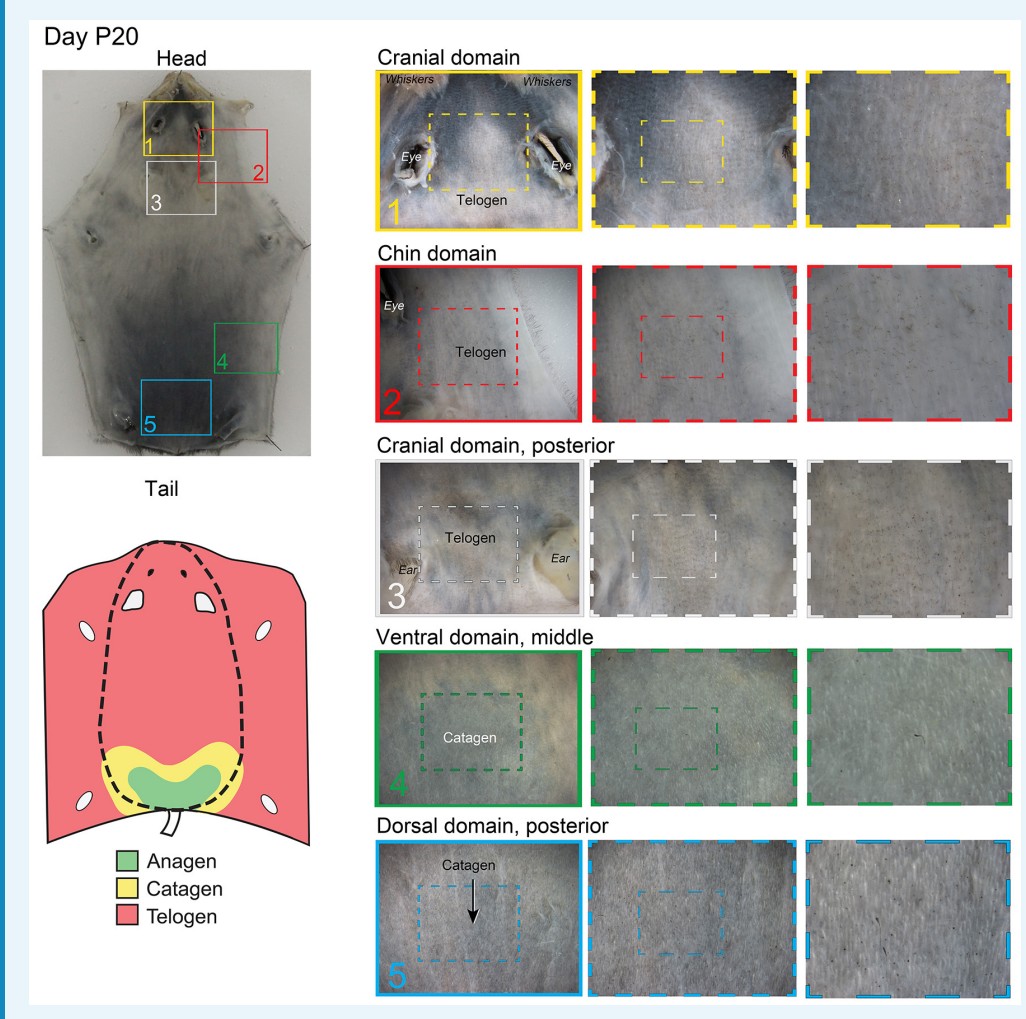

**Appendix 1—figure 11.** Distribution of the hair cycle stages in P20 mouse skin. By day P20, all but the most posterior portion of the dorsal domain HFs (insert 5) progressed into the first telogen.

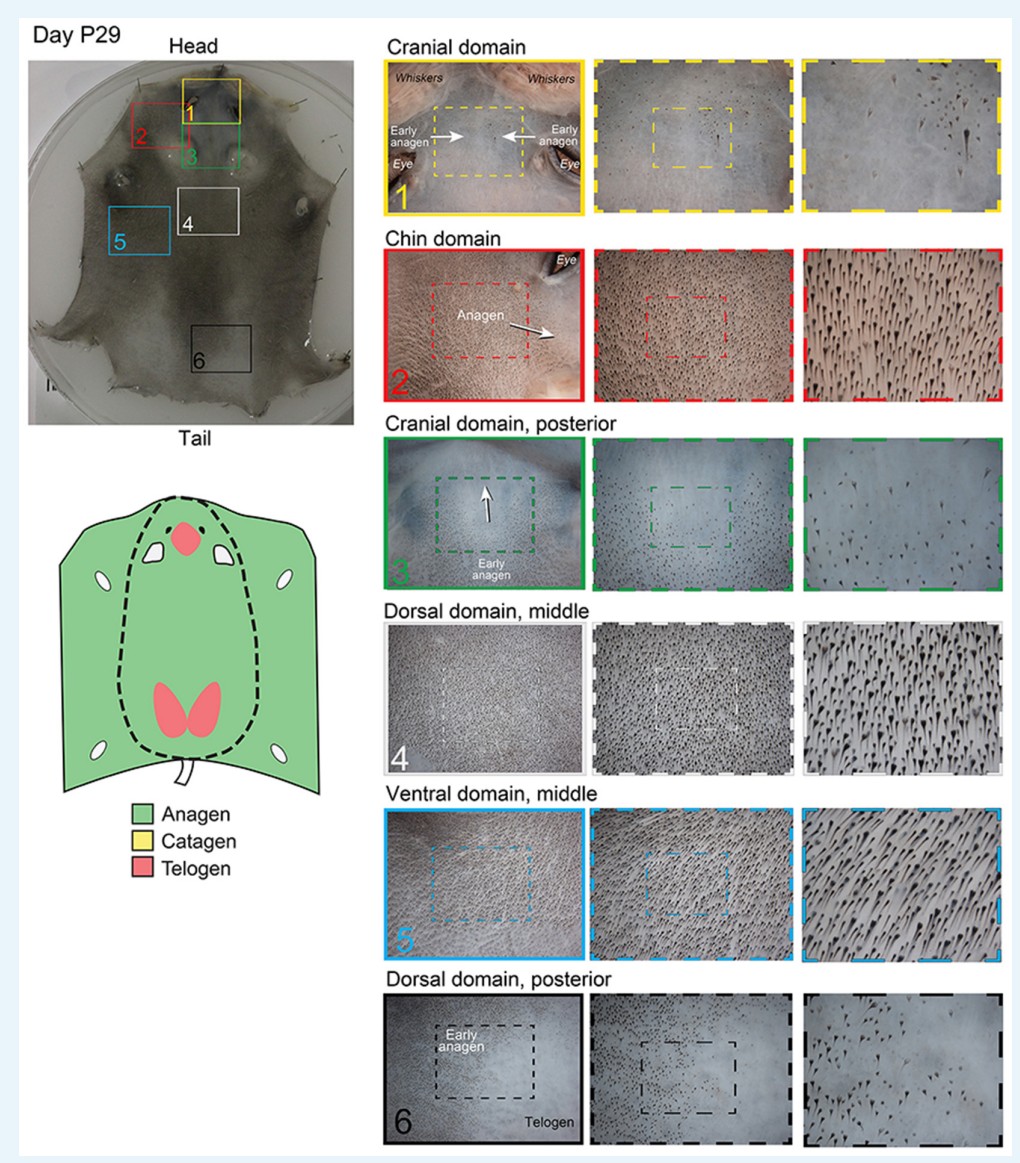

**Appendix 1—figure 12.** Distribution of the hair cycle stages in P29 mouse skin. By day P29, all ventral and most of the dorsal skin is in second anagen. One prominent exception is the cranial domain, situated in the space between eyes and ears, which at this time is still in the first telogen (insert 1). In some cases (as in the skin sample shown here), small regions of skin in the most posterior portion of the dorsal domain are also in the first telogen (insert 6).

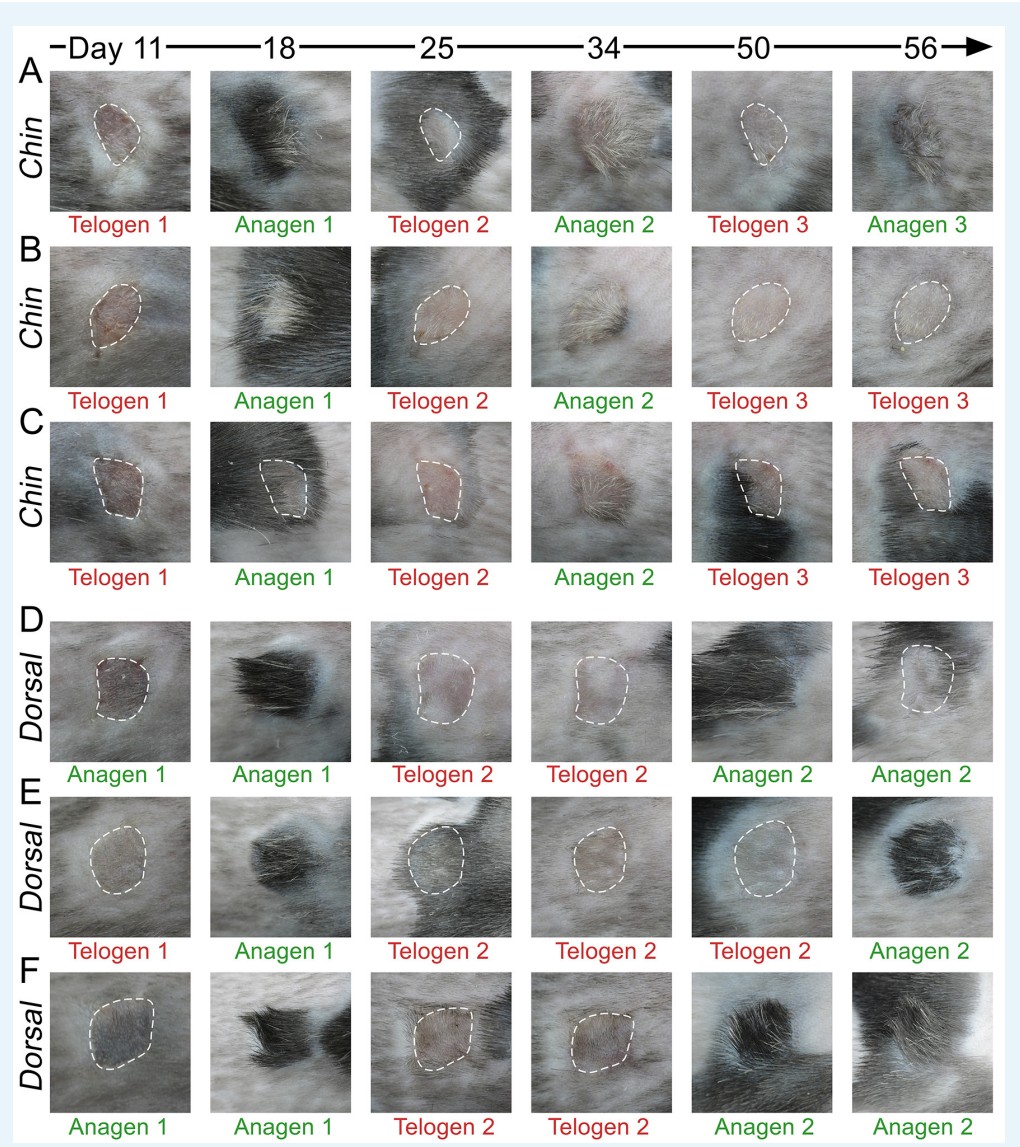

**Appendix 1—figure 13.** Hair growth cycle dynamics in skin grafts. Upon grafting onto the dorsum of pigmented SCID host mice, WT chin skin grafts (**A–C**) cycle faster compared to WT dorsal skin grafts (**D–F**). While first post-transplantation anagen starts similarly in both graft types, second anagen starts substantially faster in chin grafts. Grafts in telogen are outlined.

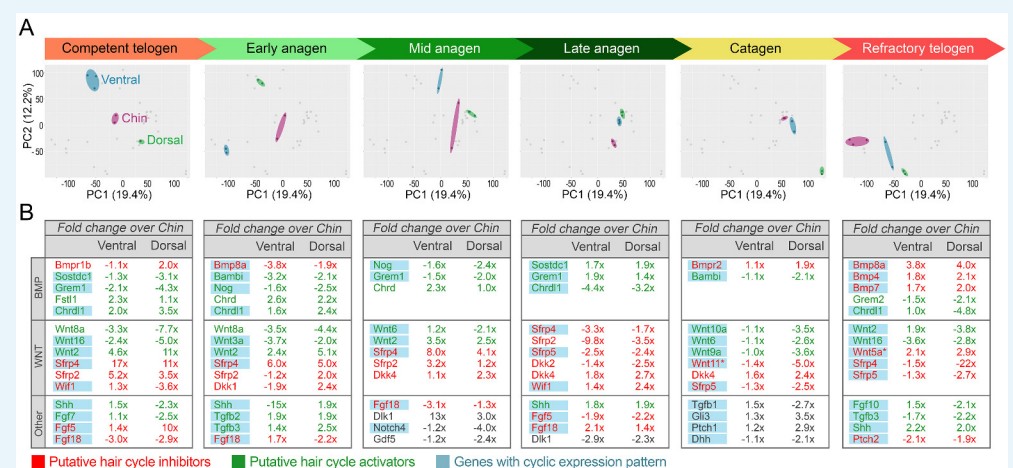

**Appendix 1—figure 14.** Domain-specific transcriptomic dynamics during hair cycle. (**A**) PCA analysis reveals largely non-overlapping transcriptomic trajectories across six hair cycle stages in chin (purple), ventral (blue) and dorsal domains (green). Deconstructed PCA plots are shown with all data points marked as grey dots and domain-specific data points highlighted for each of the six hair cycle stages. (**B**) Multiple putative hair cycle activator and inhibitor genes show domain-specific differential expression at each hair cycle stage. Putative activators are in green and putative inhibitors are in red. For each gene, relative fold changes for ventral over chin and dorsal over chin expression levels are indicated. Genes that show cyclic expression patterns are highlighted in blue.

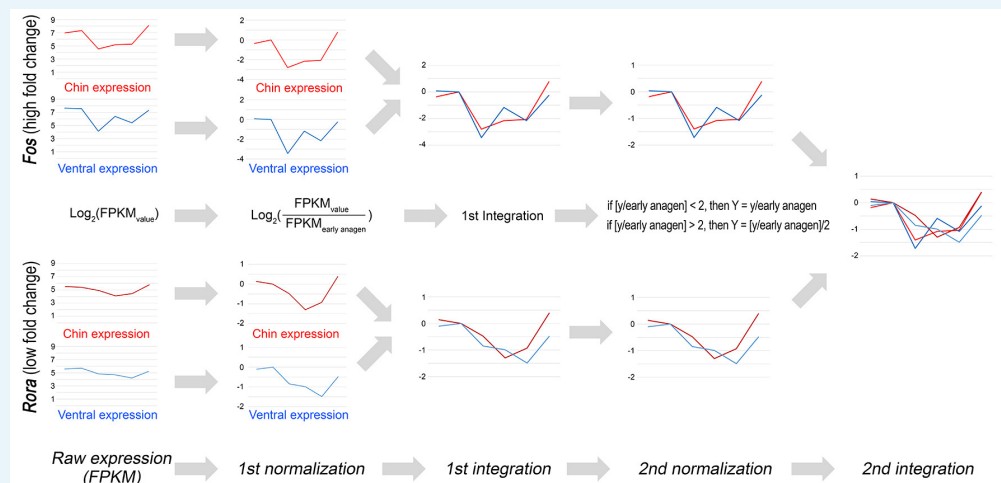

**Appendix 1—figure 15.** Double-normalized fold change calculation algorithm for RNA-seq data. Double-normalized fold change on *Appendix 1—figures 17–19* was calculated as follows: if [y/EA] <2, then Y = y/EA if [y/EA] >2, then Y = [y/EA]/2, where EA is gene expression value for early anagen.

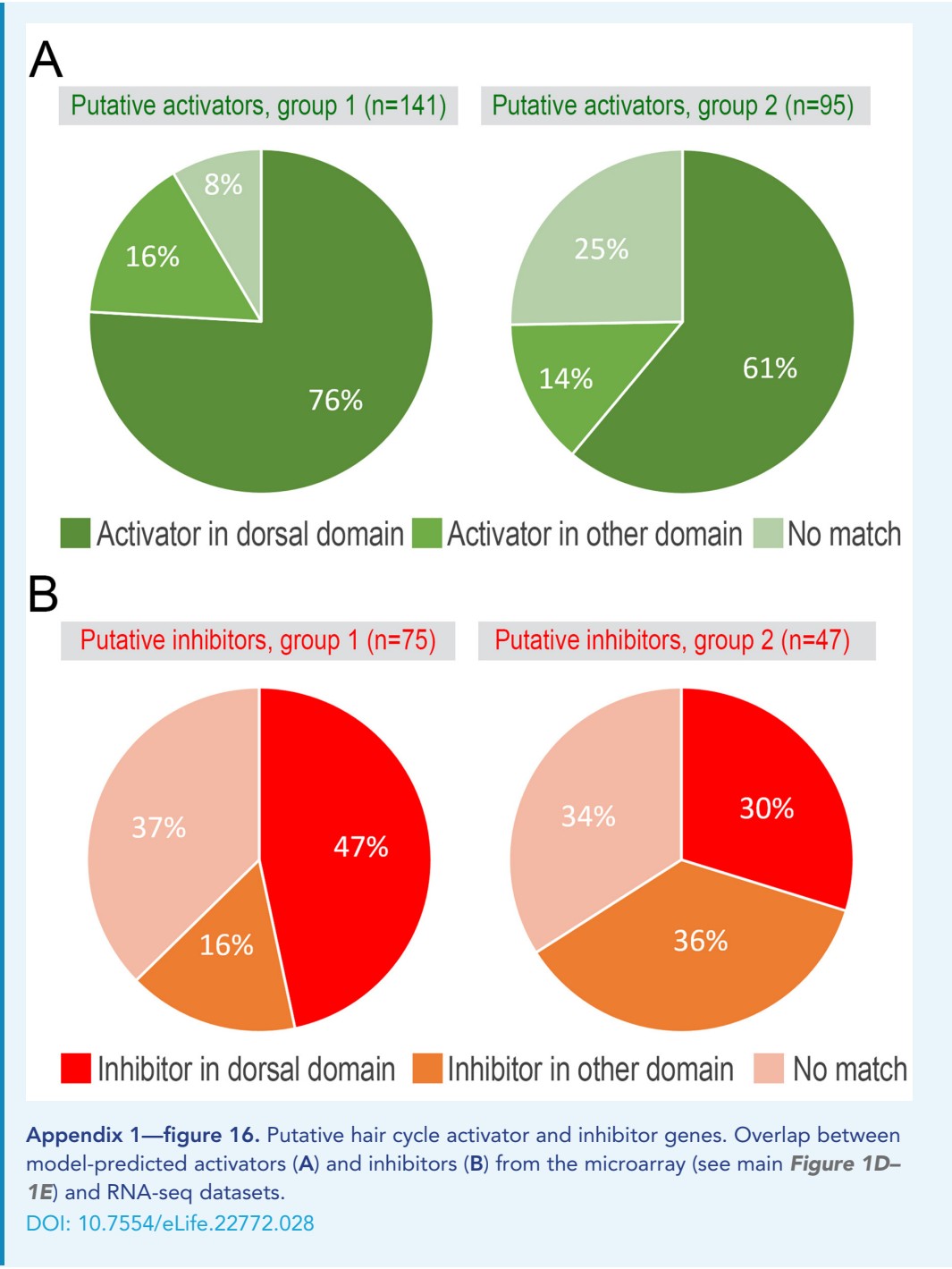

**Appendix 1—figure 16.** Putative hair cycle activator and inhibitor genes. Overlap between model-predicted activators (**A**) and inhibitors (**B**) from the microarray (see main *Figure 1D–1E*) and RNA-seq datasets.

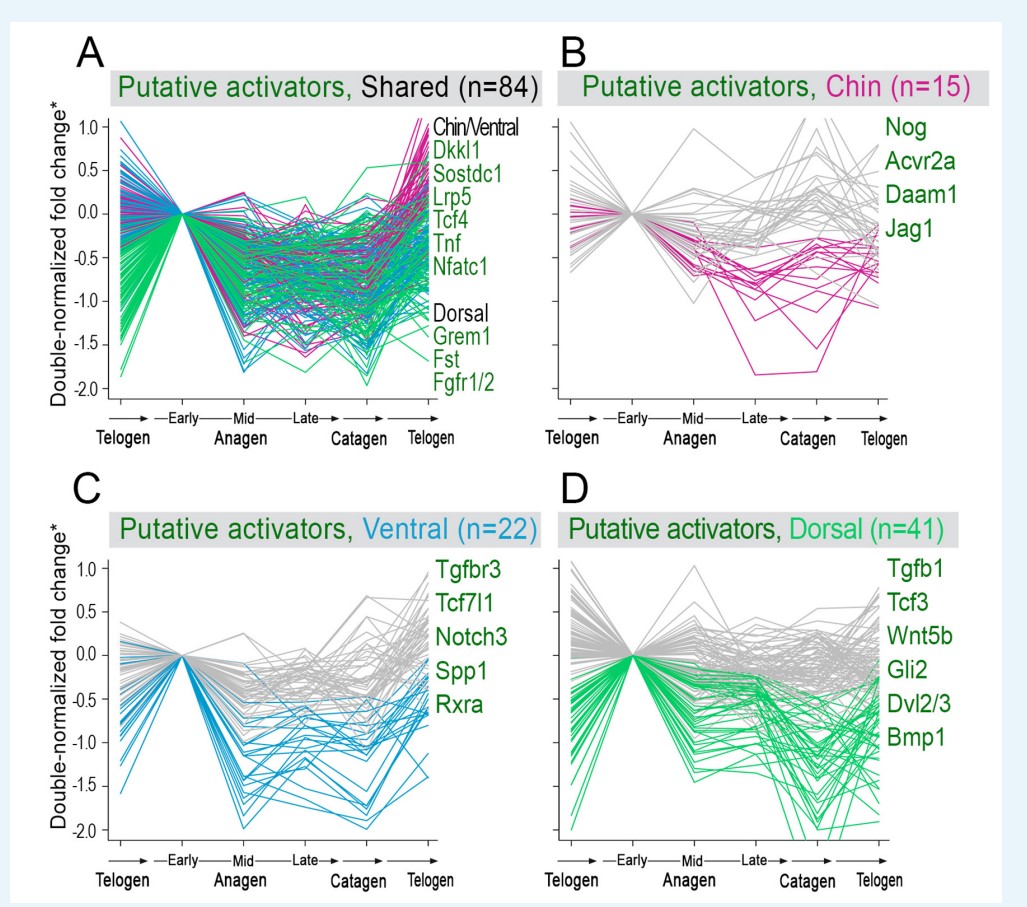

**Appendix 1—figure 17.** Model-predicted hair cycle activators. Multiple putative activator show cyclic temporal expression patterns. Some activator genes show model-predicted cyclic expression patterns in all three domains (**A**), while other activator genes show cyclic patterns only in one domain (**B–D**). Select genes are listed. For **A**), activator genes with distinctly faster expression dynamics in chin/ventral and dorsal domains are listed. (*) Relative fold change was calculated as shown on *Appendix 1—figure 15*.

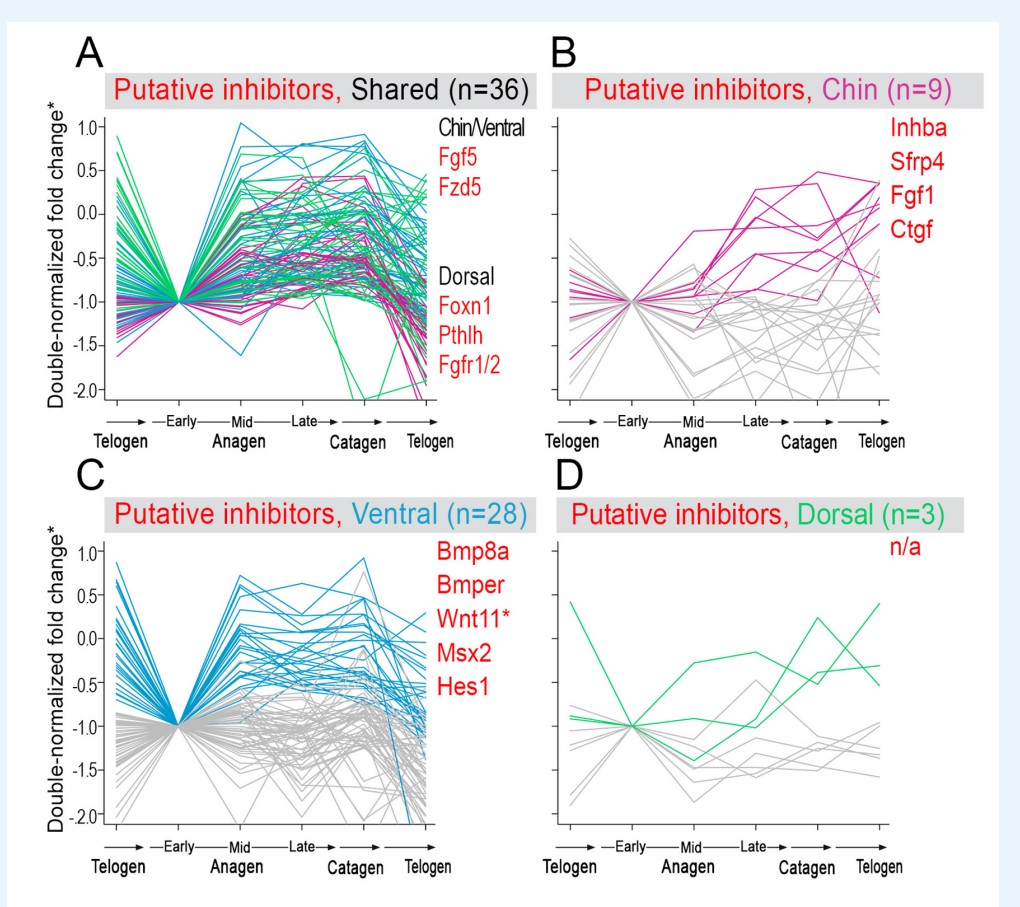

**Appendix 1—figure 18.** Model-predicted hair cycle inhibitors. Multiple putative inhibitors show cyclic temporal expression patterns. Some inhibitor genes show model-predicted cyclic expression patterns in all three domains (**A**), while other inhibitor genes show cyclic patterns only in one domain (**B–D**). Select genes are listed. For **A**), inhibitor genes with distinctly faster expression dynamics in chin/ventral and dorsal domains are listed. (\*) Relative fold change was calculated as shown on **Appendix 1—figure 15**.

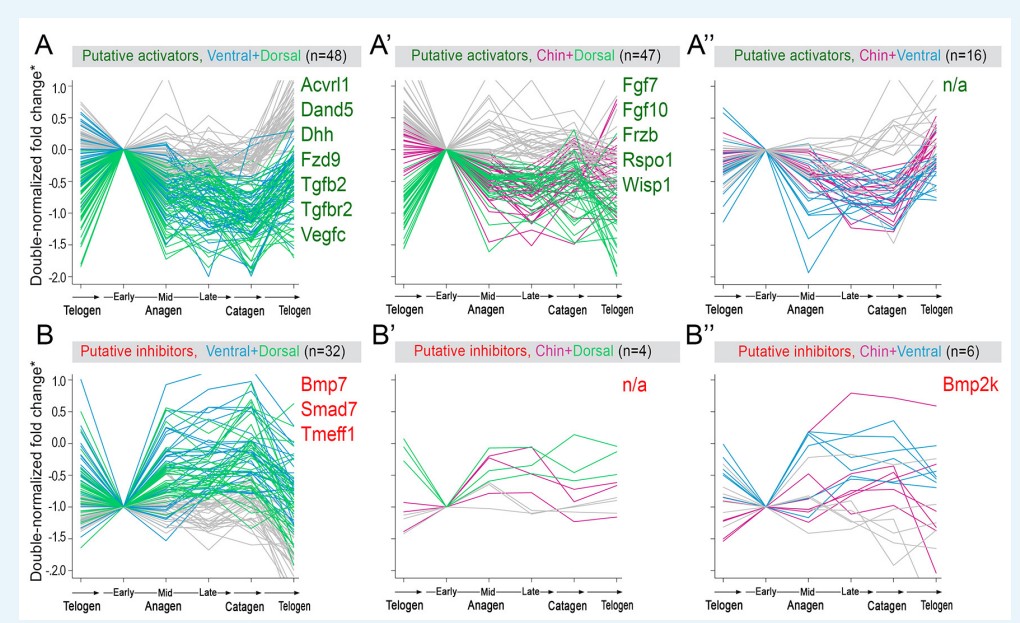

**Appendix 1—figure 19.** Additional activators and inhibitors. Additional domain-specific activators (**A–A''**) and inhibitors (**B–B''**) identified on RNA-seq. (*) Relative fold change was calculated as shown on *Appendix 1—figure 15*.

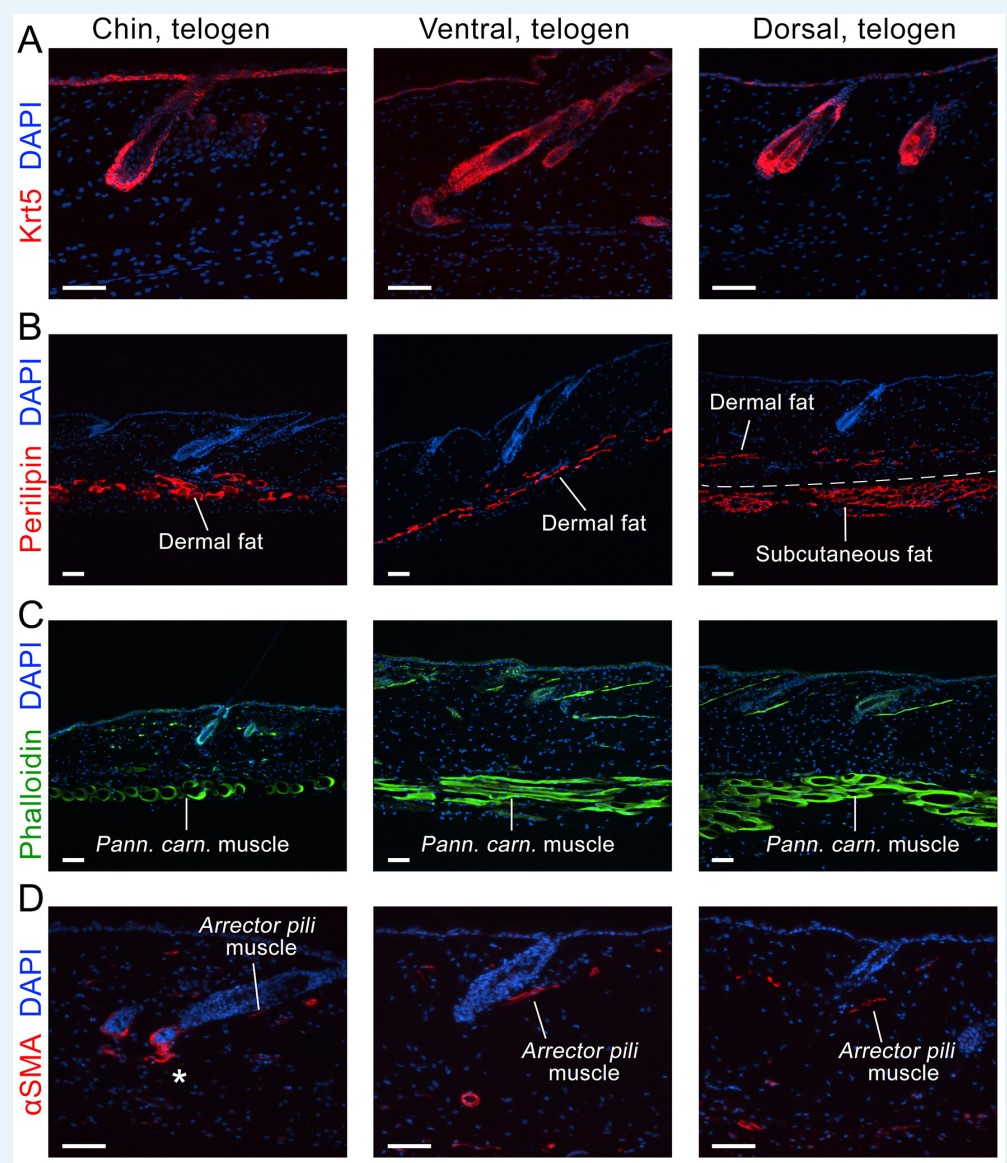

**Appendix 1—figure 20.** Domain-specific expression of select markers. Immunostaining on early telogen skin reveals thicker perilipin-expressing dermal fat layer in chin and ventral domains. Despite having a prominent subcutaneous fat layer, dorsal skin has thin dermal fat layer (**B**). Phalloidin staining shows distinctly thicker *panniculus carnosus* muscle in dorsal and ventral domains as compared to chin domain (**C**). αSMA staining shows more contractile cells around telogen HFs in chin domain (**D**). Scale bars: A-D – 50 um.

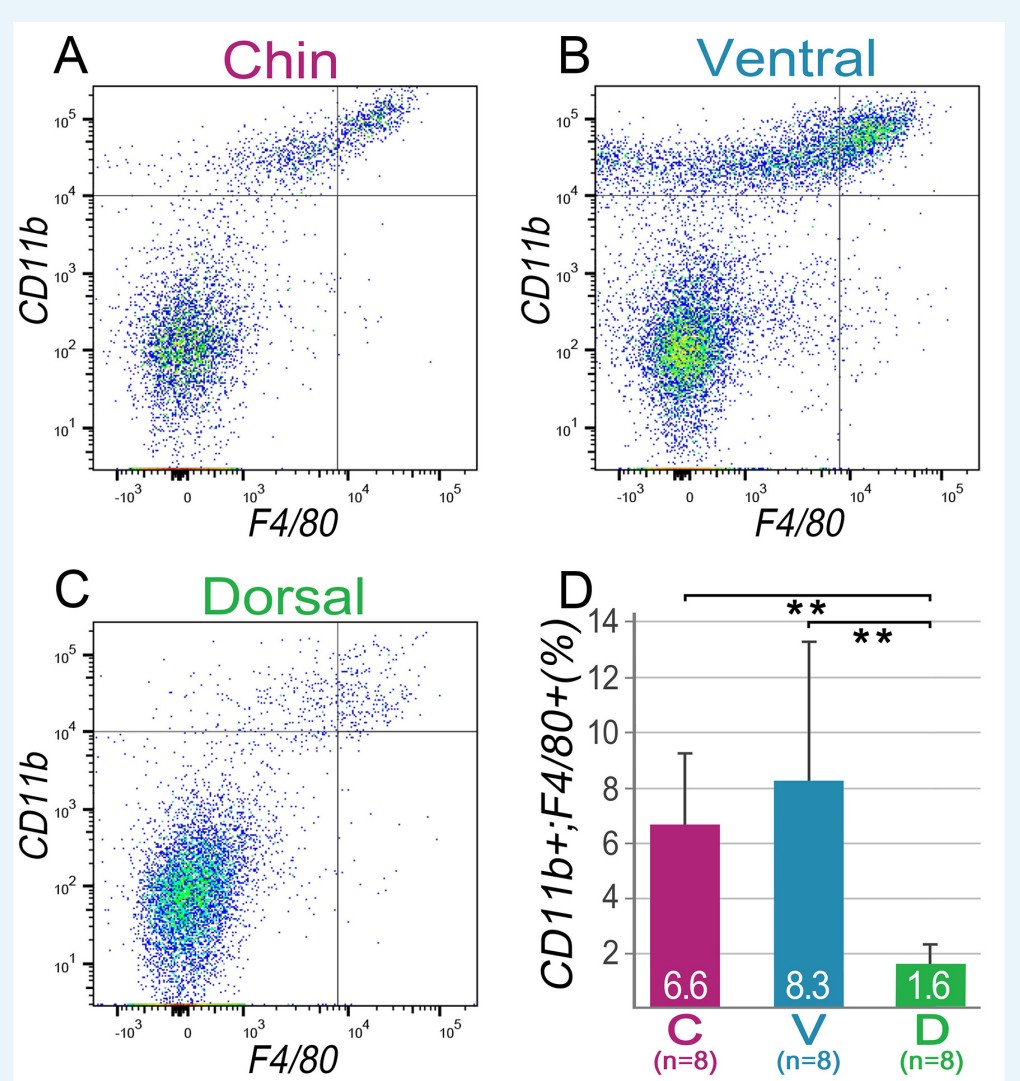

**Appendix 1—figure 21.** Domain-specific differences in macrophage abundance. FACS analysis on early telogen skin reveals that chin and ventral domains contain significantly more CD11b$^+$;F4/80$^+$ macrophages as compared to dorsal domain. Example FACS plots are shown on **A–C**. Statistical analysis is shown on **D** and domain-specific data is color-coded as follows: chin (purple), ventral (blue) and domains (green). Average percentage of macrophages is indicated within each bar (** p<0.01).

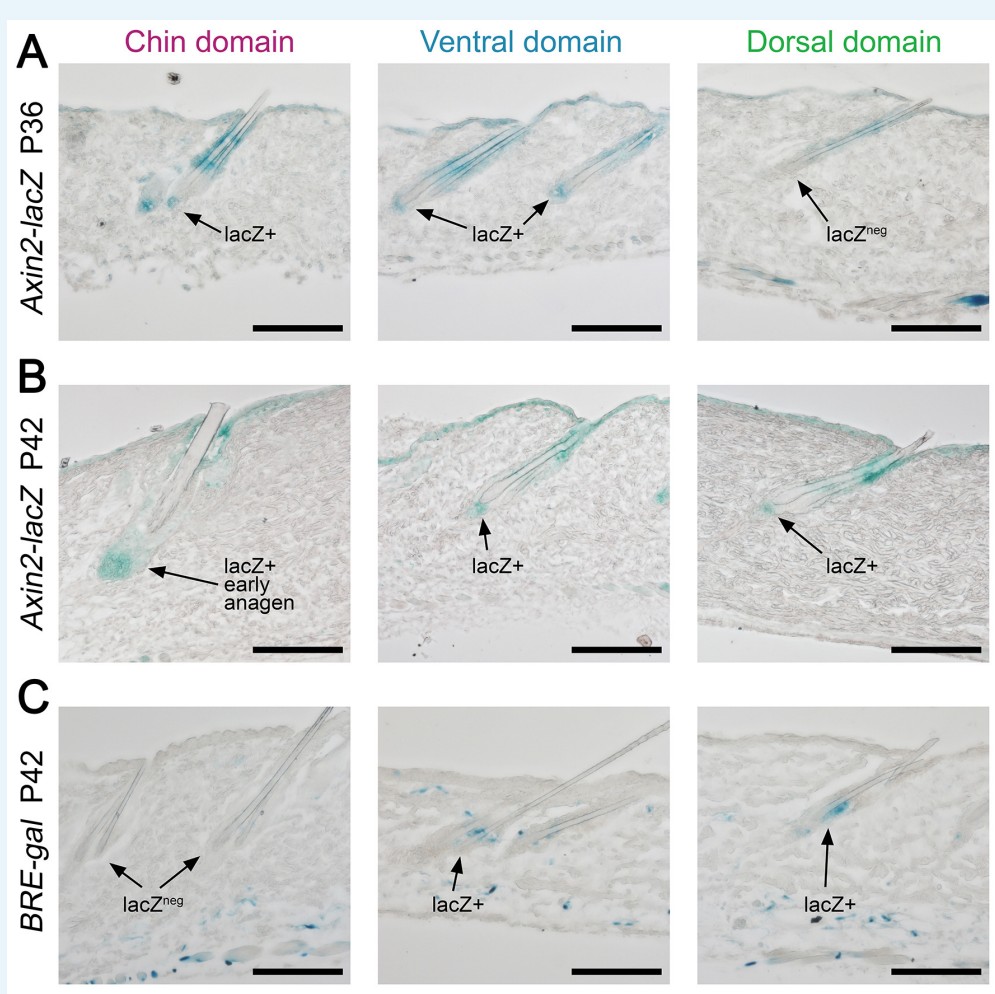

**Appendix 1—figure 22.** Domain-specific expression of WNT and BMP reporters. (**A**) In P36 *Axin2-lacZ* WNT reporter mice, the majority of chin and ventral telogen HFs contain WNT-active DPs, while many dorsal telogen HFs have WNT-negative DPs. (**B**) By P42, the number of telogen HFs with WNT-active DPs in dorsal skin raises to the levels compatible to ventral skin. By P42, chin HFs in some animals having already entered early anagen. (**C**) In P42 *BRE-gal* BMP reporter mice, dorsal skin contains many more HFs with BMP-active bulges as compared to chin skin. Ventral skin shows intermediate percentage of BMP-active HFs. Also see main *Figure 4*. Scale bars: A-D – 100 um.

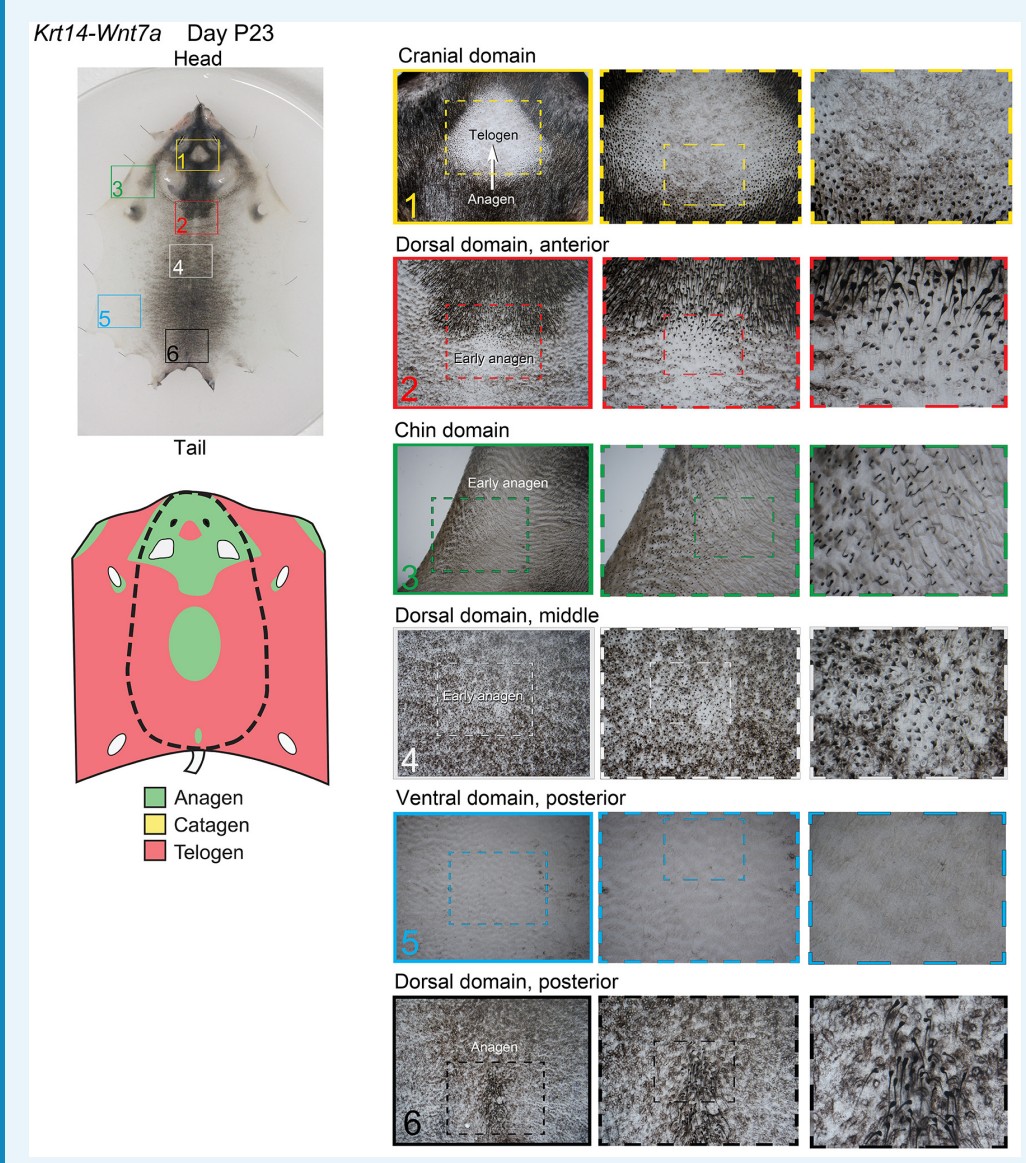

**Appendix 1—figure 23.** Distribution of the hair cycle stages in P23 *Krt14-Wnt7a* mouse skin. On day P23, *Krt14-Wnt7a* mouse shows disruption of ventral-to-dorsal dominance and spontaneous anagen initiation occurs in several locations: chin domain (insert 3) and several sites along the midline in the dorsal domain (inserts 2, 4 and 6). In addition, anagen is prominent in the cranial domain (insert 1).

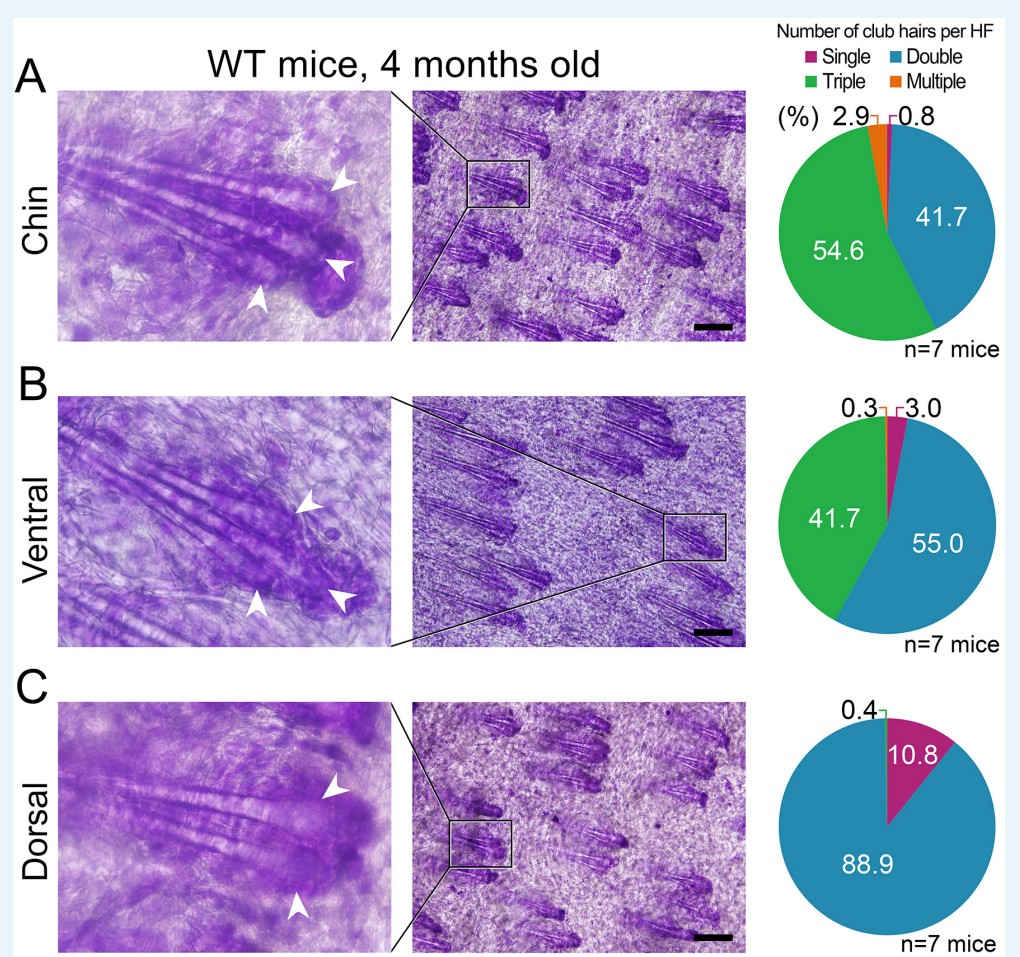

**Appendix 1—figure 24.** Differences in club hair density between the domains. (**A, B**) In four-month-old mice, a large portion of chin (54.6%) and ventral telogen HFs (41.7%) contain three club hairs. However, in the same mice, only 0.4% of dorsal HFs contain three club hairs and the majority of HFs only have two clubs (**C**). Quantification of club hair density is shown on the right. Pie charts are color-coded according to the top label: single club HFs (purple), double club HFs (blue), triple clubs HFs (green) and multiple club HFs (more than 3, orange). Scale bars: A-C – 200 um.

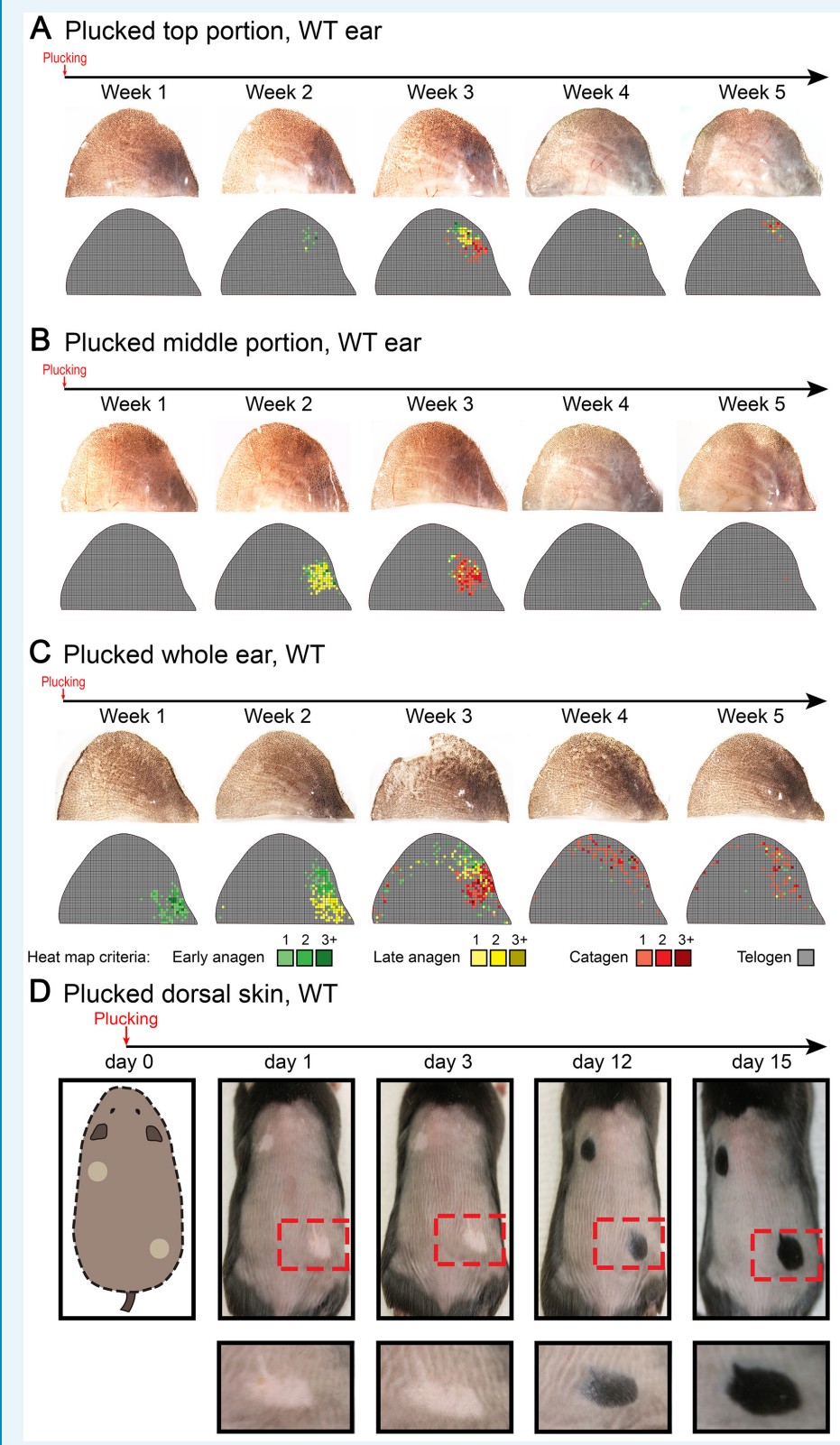

**Appendix 1—figure 25.** Anagen induction in response to hair plucking. Club hair plucking activates anagen in dorsal skin (**D**) as well as in the ear skin along the medial side of the caudal pinna (**A, B**). Also see main *Figure 5D*. Hair plucking across the whole ear (**C**) reveals that HFs along the lateral side and in the center of the pinna either do not re-enter anagen

(center) or re-enter it with a significant delay (lateral side). Hair cycle activation on **D**) was accessed based on the re-appearance of pigmentation. Ear hair plucking experiments on **A–C**) are based on five mice for each time point analyzed. Back hair plucking experiment on **D**) is based on five mice. Representative images and accompanying heatmaps are shown.

**Appendix 1—figure 26.** Lack of hair growth propagation across the ear pinna. Anagen activation in response to plucking in the ear skin remains restricted to the plucked site (caudal pinna skin medially and rostral pinna skin laterally) and does not propagate across the ear onto the opposite side. Caudal pinna skin is the same skin sample as on main *Figure 5D*.

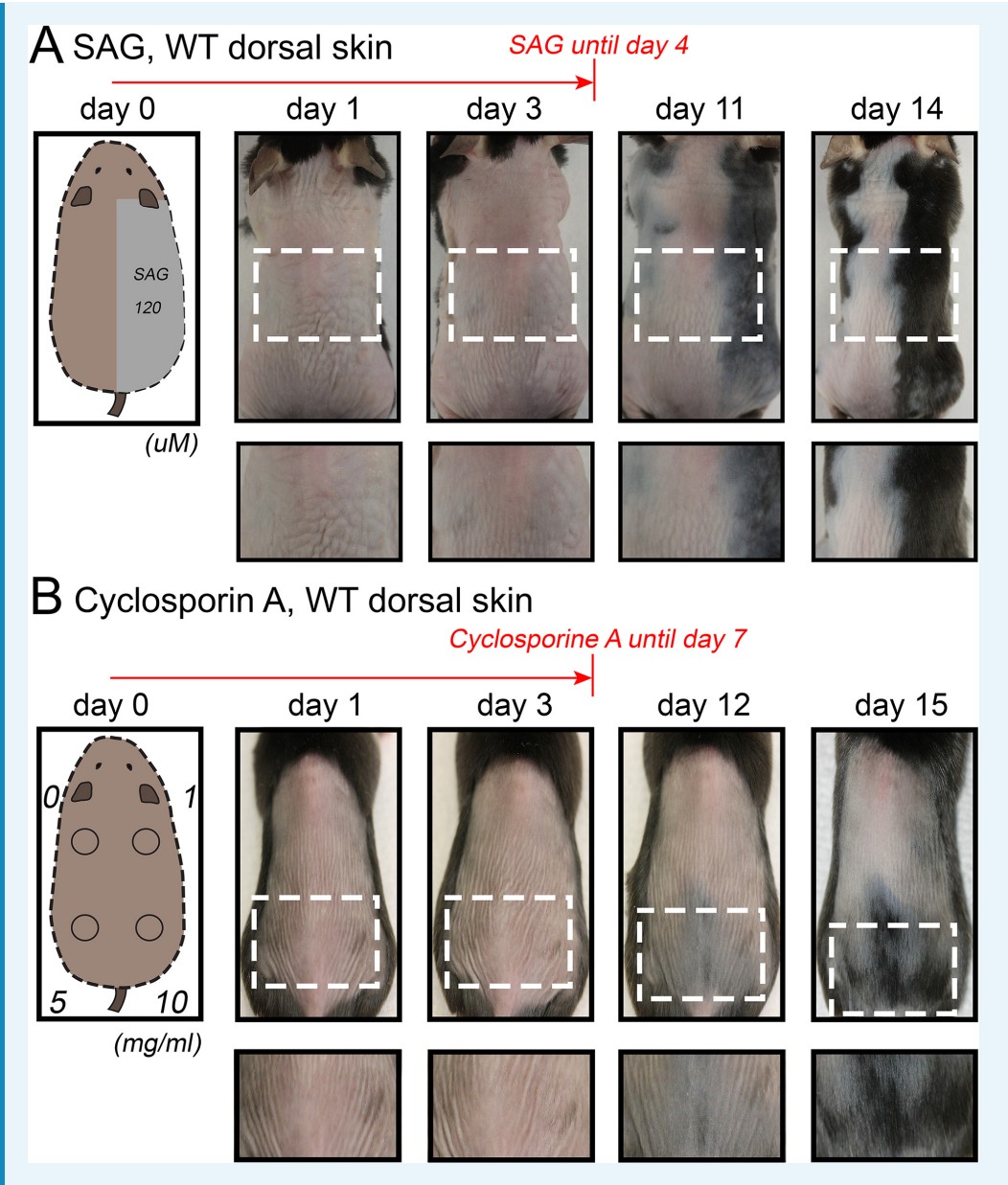

**Appendix 1—figure 27.** Hair growth activation in response to SAG and cyclosporin A in dorsal skin. Topical SAG (**A**) and Cyclosporin A treatment (**B**) induce rapid hair cycle activation in dorsal telogen skin. Hair cycle activation was assessed based on the re-appearance of pigmentation, a feature of early anagen. Each experiment shown is based on three mice. Representative mice are shown.

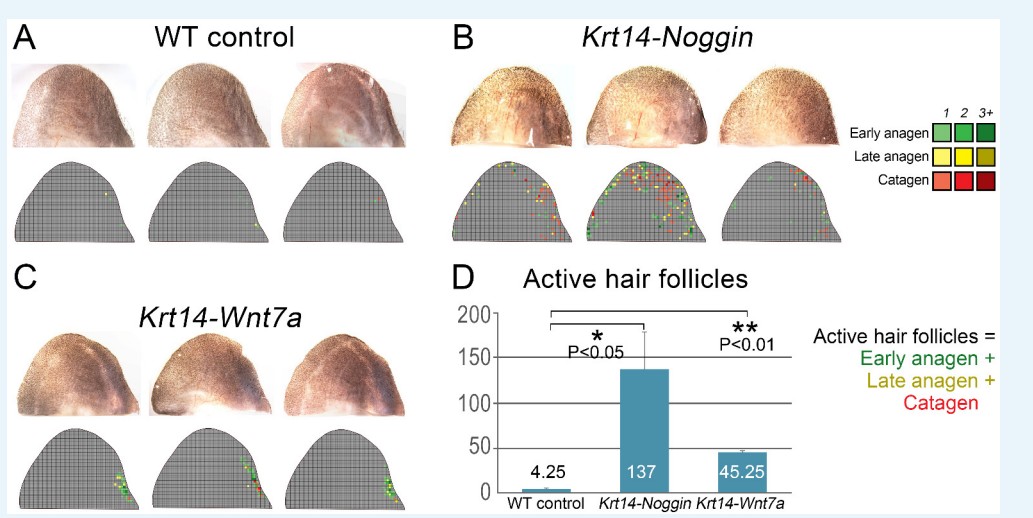

**Appendix 1—figure 28.** Partial rescue of the hair cycle in the ear skin of mutant mice. Individual ear skin samples from WT control (**A**), *Krt14-Noggin* (**B**) and *Krt14-Wnt7a* mice (**C**) are shown. Each image is accompanied by a heatmap color-coded according to the label in the top-right corner. (**D**) Statistical analysis is shown on the number of active HFs (these with early and late anagen and catagen morphologies) per ear skin sample. Average number of active HFs is indicated within each bar.

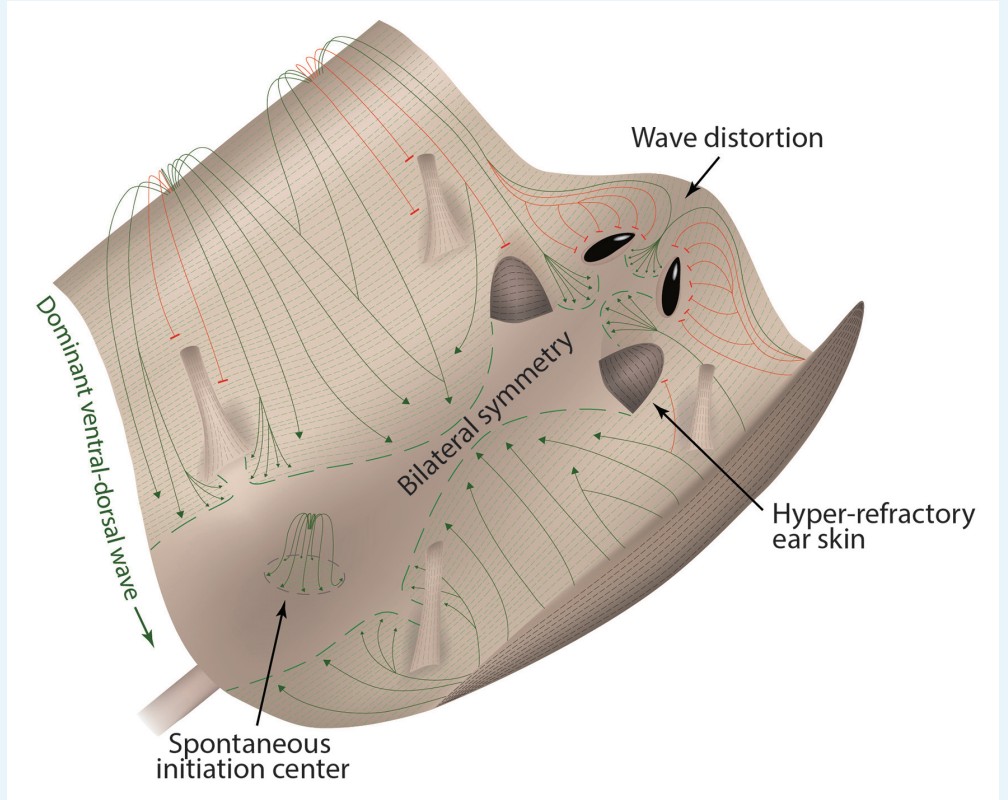

**Appendix 1—figure 29.** Landscape model of hair growth pattern formation. Mouse skin is represented as a landscape with peaks (ventral and chin skin), valleys (dorsal and cranial skin) and obstacles (eye openings, ear pinnae and limbs). In analogy to water streaming downward, hair growth waves (green) preferentially spread from ventral to dorsal skin,

producing bilateral symmetry. In analogy to water waves diffracting around physical obstacles, hair growth waves (red) diffract around hyper-refractory ear pinnae, limb skin, and eye openings.

**Appendix 1—table 1.** Numbers of club hairs used for analysis.

| Hair type | Control | Krt14-Noggin | Krt14-Bmp4 | Control | Krt14-Cre; Wnt7b[-/-] | Krt5-rtTA; tetO-Dkk1 control | Krt5-rtTA; tetO-Dkk1 induced |
|---|---|---|---|---|---|---|---|
| Auchene | 52 | 43 | 23 | 50 | 25 | 30 | 43 |
| Awl | 70 | 55 | 40 | 50 | 25 | 30 | 45 |
| Guard | 73 | 48 | 40 | 52 | 25 | 30 | 45 |
| Zigzag | 54 | 64 | 40 | 18 | 28 | 19 | 33 |

## Appendix 2

# Modeling: A multi-scale model for hair follicles reveals heterogeneous domains driving rapid spatiotemporal hair growth patterning

## 1-dimensional (1D) HF model

### Geometry of a hair follicle

The 1D HF model is on a computation domain along the $z$-axis: $z \in [Z, 0]$ (**Appendix 2—figure 1**). HF growth, characterized by elongation of the bottom part of the HF, ranges in a region $[h_{\min}, h_{\max}] \subset [Z, 0]$. Region I is the bulge region, which does not move associatively with HF growth, and is located at $[h_{\mathrm{I}}, h_{\mathrm{I}} + d_{\mathrm{I}}]$. Region II includes HG and DP during telogen, or matrix and DP during anagen. Region II is the bottom region of the HF, hence it constantly moves in association with HF growth. Computationally, Region II is located at $[h(t), h(t) + d_{\mathrm{II}}]$, where $h(t)$ marks the bottom tip of the HP. We have the following relation: $h_{\min} = h_{\mathrm{I}} - d_{\mathrm{II}}$.

### Equations for activators and inhibitors

The dynamical system consists of the diffusion of activator ($\mathrm{Act}_L$) and inhibitor ligands ($\mathrm{Inh}_L$), the reactions between ligands and receptors of the same species, where $\mathrm{Act}_R/\mathrm{Inh}_R$ denotes the free activator/inhibitor receptors; and $\mathrm{Act}_{LR}/\mathrm{Inh}_{LR}$, which denotes the activator/inhibitor ligand-bound receptors. We assume that the total amount of free and bound receptors for either activator or inhibitor are preserved at any $z$-level, i.e.,

$$[\mathrm{Act}_R](z;t) + [\mathrm{Act}_{LR}](z;t) \equiv R_{\mathrm{tot}}^A, \quad [\mathrm{Inh}_R](z;t) + [\mathrm{Inh}_{LR}](z;t) \equiv R_{\mathrm{tot}}^I$$

The dynamical system includes the following equations:

1. Two stochastic PDEs on the whole computational domain $[Z, 0]$, depicting the production (in region II) and the diffusion of the ligands, and their binding to corresponding receptors (in region I):

$$\frac{\partial}{\partial t}[\mathrm{Act}_L](z,t) = \underbrace{D_A \frac{\partial^2}{\partial z^2}[\mathrm{Act}_L]}_{\text{diffusion}} + \underbrace{I_A(h(t))\delta_{\mathrm{II}}(z)}_{\text{production}}$$
$$+ \underbrace{\left[-k_{\mathrm{on}}^A[\mathrm{Act}_L]\left(R_{\mathrm{tot}}^A - [\mathrm{Act}_{LR}]\right) + k_{\mathrm{off}}^A[\mathrm{Act}_{LR}]\right]\delta_{\mathrm{I}}(z)}_{\text{reaction}} \qquad (1)$$

$$\frac{\partial}{\partial t}[\mathrm{Inh}_L](z,t) = D_I \frac{\partial^2}{\partial z^2}[\mathrm{Inh}_L] + I_I(h(t))\delta_{\mathrm{II}}(z)$$
$$+ \left[-k_{\mathrm{on}}^I[\mathrm{Inh}_L]\left(R_{\mathrm{tot}}^I - [\mathrm{Inh}_{LR}]\right) + k_{\mathrm{off}}^I[\mathrm{Inh}_{LR}]\right]\delta_{\mathrm{I}}(z) \qquad (2)$$

where $\delta_{\mathrm{I}}$ and $\delta_{\mathrm{II}}$ indicates region I and II:

$$\delta_{\mathrm{I}}(z) = \begin{cases} 1 & \text{if } z \in [h_{\mathrm{I}}, h_{\mathrm{I}} + d_{\mathrm{I}}] \\ 0 & \text{otherwise} \end{cases}$$
$$\delta_{\mathrm{II}}(z;h(t)) = \begin{cases} 1 & \text{if } z \in [h(t), h(t) + d_{\mathrm{II}}] \\ 0 & \text{otherwise} \end{cases}$$

Functions $I_A(h)$ and $I_I(h)$ give the ligand production rates, which will be discussed in Appendix 2-Modeling the production of ligands.

2. Two stochastic ODEs in region I, evaluated at each $z \in [h_I, h_I + d_I] \subseteq [h_{\max}, 0]$, depicting the binding reaction between ligands and corresponding receptors, and the degradation of (ligand-bound) receptors:

$$\frac{d}{dt}[\text{Act}_{LR}](z;t) = k_{\text{on}}^A[\text{Act}_L]\left(R_{\text{tot}}^A - [\text{Act}_{LR}]\right)$$
$$- \left(k_{\text{off}}^A + k_{\text{deg}}^A\right)[\text{Act}_{LR}] + \beta_A + \sigma_A \qquad (3)$$

$$\frac{d}{dt}[\text{Inh}_{LR}](z;t) = k_{\text{on}}^I[\text{Inh}_L]\left(R_{\text{tot}}^I - [\text{Inh}_{LR}]\right)$$
$$- \left(k_{\text{off}}^I + k_{\text{deg}}^I\right)[\text{Inh}_{LR}] + \beta_I + \sigma_I \qquad (4)$$

where $\beta_A$ and $\beta_I$ represent the constant contributions due to the reactions between the ligands produced from and receptors located at region I, or any extra sources that may be added to the system; $\sigma_A, \sigma_I$ are noise terms. In our simulations, unless specified, we use multiplicative noise, that is

$$\sigma_A = \lambda_A[\text{Act}_{LR}]\frac{d\omega}{dt}, \quad \sigma_I = \lambda_I[\text{Inh}_{LR}]\frac{d\omega}{dt},$$

where $\lambda_A, \lambda_I$ are the noise strength, and $dt$ is the time step.

3. Boundary conditions for PDEs:

$$\text{At} z = Z: [\text{Act}_L] \equiv 0, \quad [\text{Inh}_L] \equiv 0 \qquad (5)$$
$$\text{At} z = 0: \frac{\partial}{\partial z}[\text{Act}_L] = 0, \quad \frac{\partial}{\partial z}[\text{Inh}_L] = 0 \qquad (6)$$

The parameter values related to the above equations are given in **Appendix 2—table 1**, temporal and geometric parameter values used in simulations are given in **Appendix 2— table 3**.

## Modeling HF phases by concentration difference

Let $\Delta$ be the difference of the average of $[\text{Act}_{LR}]$ and $[\text{Inh}_{LR}]$ in region I:

$$\Delta(t) = \fint [\text{Act}_{LR}]dz - \fint [\text{Ihn}_{LR}]dz$$
$$= \frac{1}{d_I}\int_{h_I}^{h_I+d_I}[\text{Act}_{LR}](z;t)\,dz - \frac{1}{d_I}\int_{h_I}^{h_I+d_I}[\text{Inh}_{LR}](z;t)\,dz \qquad (7)$$

Modeled HF growth relies on $\Delta(t)$, subject to the following rules: during telogen, HF rests at its minimum length, meanwhile $\Delta$ progressively increases; once $\Delta$ exceeds a certain threshold $\Delta_+$ at very late telogen, the activator senses it and amplifies its own production (i.e. positive feedback). The latter results in a quick increase in $\Delta$, which in turn induces anagen initiation and HF growth toward its maximal length. At that point in time, inhibitor starts to accumulate while activator degrades, leading to a decrease of $\Delta$. Once $\Delta$ deceases back to $\Delta_- = 0$ (i.e., activator and inhibitor levels balance out), HF starts to involute toward its original minimal length. In other words, HF completes its cycle and returns to telogen (**Appendix 2—figure 2**). There are two critical time points associated with these dynamics: $t^+$, the moment when $\Delta$ reaches $\Delta_+$ and activator production amplifies; and $t^-$, the moment of anagen termination, when $\Delta$ decreases back to $\Delta_-$. On these bases, we divided modeled hair growth cycle into two phases: the period from a $t^+$ to its following $t^-$ and referred to as

$\sim$ *anagen*, which includes very late competent telogen (C) and the full of anagen — both propagating (P) and autonomous anagen (A); and the period from a $t^-$ to the next $t^+$ is referred to as $\sim$ *telogen*, which includes catagen (Cat), refractory (R) telogen and most of competent telogen (**Appendix 2—figure 2**).

## Modeling the production of ligands

The production rates of $\text{Act}_L$ and $\text{Inh}_L$ are modeled in relative to the follicle growth $h(t)$ based on experimental observation presented in **Appendix 2—table 4**, where we qualitatively evaluate the activity strengths of activator/inhibitor ligands (L), antagonists (A), receptors (R), and ligand-bound receptors (LR), in region I and region II during different phases within a full HF cycle.

We simplify our model by eliminating the antagonist (A in **Appendix 2—table 4**), and estimate the *net* ligand production rate using the following formula:

$$\text{NetProductionRate} = L - \frac{A + LR}{2} \tag{8}$$

where the values of L, A and LR are from **Appendix 2—table 4**, depending on the phase of hair growth. **Appendix 2—figure 3A** shows the temporal pattern estimated from the above equation, produced by data from **Appendix 2—table 4**.

The production rates of activator ($I_A$) and inhibitor ($I_I$) are given by the following equations:

$$I_A(h) = \begin{cases} \alpha_A^{+,1} \frac{-h_{\max}+h}{-h_{\max}+h_{\min}} + \alpha_A^{+,0} & t^+ \leq t < t^- \\ \alpha_A^- & t^- \leq t < \text{next} t^+ \end{cases} \tag{9}$$

$$I_I(h) = \alpha_I^0 + \alpha_I^1 |h| \tag{10}$$

**Appendix 2—figure 3B** is produced from Appendix 2-**Equations (9, 10)**. Comparing **Appendix 2—figure 3A** with **Appendix 2—figure 3B**, we see that they follow a similar pattern, suggesting that our modeling design of Appendix 2-**Equations (9, 10)** is reasonable.

The parameter values are given in **Appendix 2—table 5**.

## Modeling of HF growth

$h(t)$ gives the location of the bottom tip of the follicle, thus $-h(t) = |h(t)|$ gives the length of the follicle. $h(t)$ is modeled as follows, with parameter values given in **Appendix 2—table 5**:

1. A follicle has a minimum length $-h_{\min} = -h_{\mathrm{I}} + d_{\mathrm{II}}$ (i.e., the length of the follicle during telogen): $h(t) \leq h_{\min}$.

2. Upon $t^+$, the follicle starts growing:

$$\frac{d}{dt}h(t) = -H^+ \left(t - t^+\right)^{m_+} \tag{11}$$

3. When the follicle grows to the maximum length $-h_{\max}$, the follicle stops growing.

4. Upon $t^-$, the follicle dies:

$$\frac{d}{dt}h(t) = H^- \left(t - t^-\right)^{m_-} \tag{12}$$

until it returns to the minimum length $-h_{\min}$.

For illustration of modeling of $h$, please see *Appendix 2—figure 1* and *2*.

## 2-dimensional (2D) and 3-dimensional (3D) HF models

The dynamics of 2D and 3D HF models are the same as the 1D HF model; however, in the 1D model we are considering a single HF in a computation domain of $[Z, 0]$, whereas in the 2D and 3D models we consider an array of HFs in a computation domain of $[0, X] \times [Z, 0]$ (*Appendix 2—figure 4A*) and of $[0, X] \times [0, Y] \times [Z, 0]$ (*Appendix 2—figure 4B*), respectively.

The diffusion terms in Appendix 2-*Equations (1, 2)* will switch to:

$$\frac{\partial^2}{\partial z^2} \to \Delta = \begin{cases} \frac{\partial^2}{\partial x^2} + \frac{\partial^2}{\partial z^2} & \text{in 2D} \\ \frac{\partial^2}{\partial x^2} + \frac{\partial^2}{\partial y^2} + \frac{\partial^2}{\partial z^2} & \text{in 3D} \end{cases}$$

The boundary conditions at $z = 0$ and $z = Z$ are the same as Appendix 2—*Equations (5,6)*. In 2D model, we apply *no leak* boundary conditions on $x = 0, X$:

$$\text{At } x = 0, X: \quad \frac{\partial}{\partial x}[\text{Act}_L] = \frac{\partial}{\partial x}[\text{Inh}_L] = 0$$

In 3D model, we apply *periodic* or *no leak* boundary conditions on $x = 0, X$, and *no leak* boundary conditions on $y = 0, Y$, depending on the detailed model. For example, when we apply periodic and no leak boundary conditions on $x = 0, X$ and $y = 0, Y$, respectively, we have:

$$\begin{aligned} [\text{Act}_L](x = 0, y, z; t) &= [\text{Act}_L](x = X, y, z; t) \\ [\text{Inh}_L](x = 0, y, z; t) &= [\text{Inh}_L](x = X, y, z; t) \\ \text{At } y = 0, Y: \quad \frac{\partial}{\partial y}[\text{Act}_L] &= \frac{\partial}{\partial y}[\text{Inh}_L] = 0 \end{aligned}$$

Parameter values used in 2D and 3D simulations are given in *Appendix 2—table 1 - 3*, *5*.

## Numerical methods

In 1D and 2D model, we use implicit finite difference to discretize the diffusion term and forward Euler methods to discretize the production and reaction terms. For example, the discretized equation of Appendix 2-*Equation (1)* is in the form of

$$\frac{u_n^{m+1} - u_n^m}{\Delta t} - D_A \frac{u_{n+1}^{m+1} - 2u_n^{m+1} + u_{n-1}^{m+1}}{\Delta z^2} = f_n^m \tag{13}$$

where $u_n^m$ stands for $[\text{Act}_L]_n^m$, and $f$ represents the production and reaction terms in Appendix 2-*Equation (1)*. It is unconditionally stable, and the truncation error is $O(\Delta t) + O(\Delta x^2)$ in 1D, or $\delta \sim O(\Delta t) + O(\Delta x^2) + O(\Delta y^2)$ in 2D.

In 3D model, implicit finite difference method is not practical, since it requires solution of a system of $N_x \times N_y \times N_z$ equations. Instead we use explicit finite difference method. The truncation error is $\delta \sim O(\Delta t) + O(\Delta x^2) + O(\Delta y^2) + O(\Delta z^2)$. To make this method stable, we require that

$$\nu = D\Delta t \left( \frac{1}{\Delta x^2} + \frac{1}{\Delta y^2} + \frac{1}{\Delta z^2} \right) \le \frac{1}{2}$$

For the discretization parameter values we used in simulations (**Appendix 2—table 3**), we have $\nu \sim 0.13$, which satisfies the stability requirement.

## Effects of noise on HF dynamics

As we discussed in the main text, cycling occurs only within a range of signal strengths (**Figure 1C**, white region), and when activator/inhibitor signals are either too strong or too weak (**Figure 1C**, grey regions), HF reaches a stable equilibrium state and fails to cycle. When inhibitor is very strong or activator is very weak, the signal difference threshold required for spontaneous excitation cannot be reached (**Appendix 2—figure 5A**), and HF equilibrates in a prolonged telogen-like state. On the other hand, when inhibitor is very low, it fails to catch up with the activator, and HF equilibrates in a prolonged anagen-like state instead (**Appendix 2—figure 5B**).

Within the excitable region on which the HF can cycle, we used noise-free model to show the effects of signal levels on $\sim$ anagen and $\sim$ telogen lengths: higher inhibitor level results in shorter $\sim$ anagen and longer $\sim$ telogen (**Figure 1C**, yellow and magenta lines). Qualitatively similar results were also obtained in the stochastic regime with multiplicative noise in both activator and inhibitor signaling (**Appendix 2—figure 6A**).

We investigate how noise in signaling might affect the HF growth by adding multiplicative noise with different strengths in both activator and inhibitor signaling. Simulations show that strong signaling noise shortens the average length of both $\sim$ anagen and $\sim$ telogen, and at the same time, increases their variability (**Appendix 2—figure 6B**). This is interesting, although not surprising, because both $\sim$ anagen and $\sim$ telogen checkpoints are largely determined by the differential level between the activator and inhibitor signals. Stronger noise is likely to increase the probability of the signal reaching the critical differential level, advancing HFs to the next phase - an irreversible process.

Next we repeat the above simulations with additive noise instead. With additive noise added to both activator and inhibitor LR equations, $\sim$ anagen is shortened while $\sim$ telogen is extended, the same as **Appendix 2—figure 6A Appendix 2—figure 6B**. However, the change of Std of additive noise is not as much as when it is from multiplicative noise (**Appendix 2—figure 6C,D**). On the other hand, for medium inhibitor level but different noise strength, stronger additive noise also leads to shorter $\sim$ anagen and $\sim$ telogen, but the change is not as large as it is for multiplicative noise - mostly due to the reason that we choose same noise strength $\lambda_A$ and $\lambda_I$ for both multiplicative and additive noise to make a comparison, in which case multiplicative noise would cause bigger effects on the activator and inhibitor levels.

Finally we would like to point out that we use multiplicative noise in all 2D and 3D simulations. However, if additive noise is adopted instead, the results will not change essentially. For example, we repeat the simulation of interaction between dorsal and ventral HF waves with additive noise having strength $\lambda_A = \lambda_I = 0.05$. Similar to that with multiplicative noise (**Figure 2F**), interaction between dorsal and ventral HF waves last for two cycles, then breaks into more stochastic waves with bilateral symmetry (**Appendix 2—figure 7**).

## Changes in the total amount of activator and inhibitor receptors results in different sensitivity of $\sim$ anagen and $\sim$ telogen lengths to signaling changes

While higher inhibitor levels or lower activator levels always result in shorter $\sim$ anagen and $\sim$ telogen, we find that by altering certain background parameter values, HF may achieve different sensitivity to such signal changes. One such pair of parameters we find is the total amount of activator and inhibitor receptors ($R_{\text{tot}}^A$ and $R_{\text{tot}}^I$).

We investigate $\sim$ anagen and $\sim$ telogen lengths with either both low amounts of activator and inhibitor receptors ($R^A_{\text{tot}} = R^I_{\text{tot}} = 6$), or both high amounts ($R^A_{\text{tot}} = R^I_{\text{tot}} = 60$). The result is shown in **Appendix 2—figure 8**. First, under the same signaling levels, high $R^A_{\text{tot}}$ and $R^I_{\text{tot}}$ always result in both short $\sim$ anagen and $\sim$ telogen. This difference between receptor levels in ventral and dorsal domains was used in the 3D modeling of HF wave interactions across dorsal and ventral domains (see Appendix 2-Dorsal and ventral HF patterns); as shown by experimental observations, ventral HFs have both shorter anagen and telogen than dorsal HFs. Next, these two modeling HF types also show different sensitivity to inhibitor signaling changes. When the receptor levels are both low, both $\sim$ anagen and $\sim$ telogen phases react more dramatically to changes in signaling (**Appendix 2—figure 8A,B**, solid bars). For instance, after the same net increase in the inhibitor ligand value, simulated $\sim$ anagen phase decreased by $\sim$ 16 days and $\sim$ telogen phase increased by $\sim$ 23.8 days under low $R^A_{\text{tot}}$ and $R^I_{\text{tot}}$ regime. However, under high $R^A_{\text{tot}}$ and $R^I_{\text{tot}}$ regime, $\sim$ anagen phase decreased by $\sim$ 6.4 days and $\sim$ telogen phase increased by $\sim$ 14.8 days. **Appendix 2—figure 8C,D** exemplifies changes in activator/inhibitor LR profiles and accompanying changes in HF growth dynamics under low $R^A_{\text{tot}}$ and $R^I_{\text{tot}}$ regime, while **Appendix 2—figure 8E,F** under high $R^A_{\text{tot}}$ and $R^I_{\text{tot}}$ regime. Based on this, we speculate that higher $R^A_{\text{tot}}$ and $R^I_{\text{tot}}$ abundance confers HFs with better ability to read out signal changes, and increases the robustness of $\sim$ anagen and $\sim$ telogen durations.

We also explore all four possible combinations of low vs. high activator and inhibitor profiles. From the results in **Appendix 2—figure 9** we see that each combination gives different $\sim$ anagen and $\sim$ telogen lengths, and that each combination results in different sensitivity to inhibitor level changes.

## Possible interactions between the activator and inhibitor pathways do not qualitatively alter the HF dynamics

In our model the temporal patterns of activator and/or inhibitor ligand activities are dependent on HF growth, which is based on experimental observation. This is a phenomenological description, part of which may come out from any interaction between the pathways.

Recently, *Kandyba et al. (2013)* discovered the inhibition from BMP (inhibitor) to WNT (activator); other than this, so far there is no solid support that other interactions between activator and inhibitor exist. We incorporate this inhibitory interaction into our model in two ways:

1. the inhibitor LR inhibits the ligand-receptor binding of activator (**Appendix 2—figure 10A**):

$$\frac{d}{dt}[\text{Act}_{LR}](t) = \frac{k^A_{\text{on}}[\text{Act}_L]\left(R^A_{\text{tot}} - [\text{Act}_{LR}]\right)}{1 + \left(a[\text{Inh}_{LR}]\right)^m} - \left(k^A_{\text{off}} + k^A_{\text{deg}}\right)[\text{Act}_{LR}] + \beta_A + \sigma_A \tag{14}$$

or

2. the inhibitor ligands inhibits the production of activator ligands (**Appendix 2—figure 10B**):

$$I_A \rightarrow I_A 1 + \left(a[\text{Inh}_L]\right)^m \tag{15}$$

However, in either way, please be aware that the original model which gives the phenomenological description already involves possibly existing interactions, and adding such an inhibitory feedback only enhances the effect.

In either Appendix 2-**Equation (14) or (15)**, we run simulations with $m = 2$, $\beta_I = -4$ for a wide range of $a$, where small/large $a$ corresponds to weak/strong inhibition of the inhibitor

(BMP) to the activator (WNT), with $a = 0$ indicating that the inhibition does not exist, and the dynamics return to Appendix 2-*Equation (1-4)*. Simulation results show that in either mode of inhibition, $\sim$ *anagen* length will decrease and $\sim$ *telogen* length will increase. (*Appendix 2—figure 10C,D*). Moreover, as the inhibition becomes stronger, $\sim$ *anagen* length quickly reaches a minimum (*Appendix 2—figure 10C,D*, green lines); however, $\sim$ *telogen* length increases gradually (*Appendix 2—figure 10C,D*, red lines). Observing the dynamical patterns of HF growth and activator and inhibitor LR, we find that when the inhibition is extremely strong, the inhibition of the inhibitor to the activator push the activator LR almost flat (*Appendix 2—figure 10E*, green line), while the inhibitor LR still shows the hill-valley pattern (*Appendix 2—figure 10E*, red line).

When the activator is inhibited, a single HF is still able to present three dynamic regimes upon changes of signal levels: an excitable one and two equilibrium ones, the same as in *Figure 1C*. *Appendix 2—figure 10F* gives the profile of the excitable regime, from which we see that the dynamics react to inhibitor levels in a same way as before, with the only difference that the excitable regime requires a weaker inhibitor level now: $\beta_I \in [-13, -4]$ with the inhibitory feedback in L-R binding with a strength $a = 10$, compared to $\beta_I \in [-9, 0.5]$ without any feedback (*Figure 1C*).

## 2D and 3D modeling details

### Dorsal and ventral HF patterns

It is observed on mouse dorsal skin that during the first 2-to-3 cycles, synchronized anagen waves propagate from head to tail, followed by more spontaneous, asynchronous waves from the fourth cycle onwards, while still showing some bilateral symmetry. To study the cause of such patterns, we propose four scenarios. The first three of them consider a uniform HF domain simulating the dorsal domain only, and they are simulated by both the 2D and 3D models. The fourth scenario involves interactions between different HF domains: that is simulating the interactions between dorsal and ventral domains; hence, it is only simulated in the 3D model. Below we list certain modeling details of the 3D simulations, with the 2D simulations designed in similar ways:

1. Consider only the dorsal domain, where all HFs have equal probability of anagen initiation. To simulate this scenario, we consider on a $60 \times 100$ domain, we add multiplicative noise terms $\sigma_A$ and $\sigma_I$ ($\lambda_A = \lambda_I = 0.035$) to the activator and inhibitor LR Appendix 2-*Equations (3, 4)* (*Appendix 2—figure 11* for 2D, *Appendix 2—figure 12* and *Appendix 2—video 1* for 3D simulation). In this case, we see spontaneous HF waves from the very first cycle without any bilateral symmetry.

2. Consider only the dorsal domain, where HFs near the head have higher built-in levels of activator signaling. Again we simulate this scenario on a $60 \times 100$ domain, adding multiplicative noise terms $\sigma_A$ and $\sigma_I$ ($\lambda_A = \lambda_I = 0.02$) to the activator and inhibitor LR Appendix 2-*Equations (3, 4)*. To simulate the built-in high level of activator near head, for all HFs on the top row ($y = 100$), we give them constant increase in their activator production ($I_A \rightarrow I_A + 50$). In addition, we set a small threshold $\Delta_+$ ($\Delta_+ = 1.8$) for anagen activation (*Appendix 2—figure 13*, *Appendix 2—video 2*). In this case, wee see highly successive wave patterns steadily propagate from head to tail, which may last for many cycles. This scenario is referred to as 'target pattern' (*Murray et al., 2012*).

3. Consider only the dorsal domain, where HFs near the head have built-in temporal phase advancement. To simulate this scenario, for all HFs on the top row ($y = 100$), we give a one time increase (at $t = 0.1$) in the activator production ($I_A \rightarrow I_A + 3000$). The noise terms are the same as in the first scenarios ($\lambda_A = \lambda_I = 0.035$). In both 2D and 3D simulations, the first few waves steadily propagate from head to tail (*Appendix 2—figure 14* for 2D, *Appendix 2—figure 15* and *Appendix 2—video 3* for 3D simulations). Spontaneous initiations gradually

show up in later waves. However, the initial head-to-tail wave pattern persists for at least 10 consecutive cycles despite constant disruption from spontaneous initiations, which violates biological observation where the steady head-to-tail pattern breaks down as early as the third wave. Moreover, 3D simulation shows no bilateral symmetry.

4. Finally we consider the scenario where dorsal HFs interact with ventral HFs, where dorsal HFs near the head have built-in temporal phase advancement than other dorsal HFs. Moreover, the first anagen entrance of ventral HFs occurs later than in dorsal HFs, and ventral HFs have shorter anagen and telogen phases. We consider a $100 \times 100$ domain, where the $60 \times 100$ area in the middle is to simulate the dorsal domain, with its HFs having less available receptors ($R^A_{tot} = R^I_{tot} = 15$) resulting in longer anagen and telogen; the two side-domains, each with a size of $20 \times 100$ simulate the ventral domain, which flank the dorsal domain and are connected via periodic boundary condition, and HFs in them have more available receptors ($R^A_{tot} = R^I_{tot} = 50$). In the beginning, for all HFs on the top row in the dorsal domain ($20 \leq x \leq 80, y = 100$), we give a one time increase in the activator production ($I_A \rightarrow I_A + 1000$), and we block the growth of ventral HFs by setting all HF length $h = h_{min}$ in these areas during $t \leq 14$. Multiplicative noise terms $\sigma_A$ and $\sigma_I$ ($\lambda_A = \lambda_I = 0.02$) are added to the activator and inhibitor LR Appendix 2-**Equations (3, 4)**. In this case, we see wave patterns similar to those in our biological observations, shown in **Figure 2F** and **Appendix 2—video 4** and discussed in the main text.

## Hyper-refractory domain

In studying the ear pinna skin behaving as hyper-refractory HF domain, we model two phenomena: 1) the high level of inhibitor in ear skin results in hair cycling termination, and HFs equilibrate in an extended telogen; 2) anagen waves spreading from dorsal skin could not propagate into the ear skin due to its hyper-refractory state.

We model the first phenomenon on a $100 \times 100$ domain, with parameter values given in **Appendix 2—table 1** (3D dorsal column), 1 and 5, except $\Delta^+ = 3$, $\lambda_A = \lambda_I = 0.05$, $\alpha^1_I = 22$ and

$$\beta_A(t) = 2 + \frac{1.8}{t + 0.2}$$ (16)

Noise is able to randomly initiate the first HF growth cycle, but the high inhibitor levels prevent HF wave propagating, and no other HF cycles appear, HFs stay in an extended telogen (**Appendix 2—figure 16**).

For the second phenomenon, that is no HF wave propagating from dorsal skin into the ear skin, we model on a $100 \times 100$ domain. Parameter values are given in **Appendix 2—table 1** 3D Dorsal column, with $\lambda_A = \lambda_I = 0.02$, and the region $[30, 70] \times [30, 60]$ and with elevated inhibitor level ($\alpha^0_I = 2$ in this region) to simulate the ear skin, while the other part the dorsal domain. We stimulate a top-to-bottom HF wave by giving a one-time increase (at $t = 0.1$) in the activator ligand production in the top row ($I_A \rightarrow I_A + 1000$ at $y = 100$). We see steady HF wave propagate through the dorsal region for several cycles along with stochastic initiations; however, no HF wave is able to propagate into the ear skin (**Appendix 2—figure 17**).

## Wave breaker

Hairless skin domains and physical breaks in the skin can break homogenous wave spreading. To model this wave-breaker phenomenon, we consider a $100 \times 100$ domain to simulate the Cranial domain, with parameter values given in **Appendix 2—table 1** (the 3D dorsal column), 3 and 5, with $\lambda_A = \lambda_I = 0.02$. The $[30, 72] \times [56, 60]$ region is 'cut-off' (**Figure 7A**, black region), that is no HF growth permitted and no molecules are allowed to diffuse into this region. Additionally, we apply no-slip boundary conditions to the four boundaries of this

strip. We stimulate the top row of HF growth at the beginning of the simulation by giving it a short-time increase in activator production (for $0 \leq t \leq 0.1$, set $I_A \rightarrow I_A + 400$ for $y = 100$), which creates a steady HF wave propagating from top to bottom. The 'eyelid' region breaks the wave propagation, creating the distortion-like effect which is particularly clear during the first few cycles.

In comparison, we also model another wave-breaker scenario: while still no HF growth is allowed in the wave-breaker region, molecules are allowed to diffuse into this region. This might be a similar scenario to a skin where a part of it has the HF growth impaired. We block the HF growth in this region by setting $h(t) \equiv h_{\min}$ at all times. Other parameter values are the same as in the previously described scenario. Simulation results show that such a growth-impaired region also creates a distortion-like effect which is quite similar to the 'eyelid' simulation results (*Appendix 2—figure 18*).

## Sensitivity test of parameters
In this part, we test the sensitivity of several crucial parameters in our model.

### Threshold $\Delta_+$
At single HF level, $\Delta_+$ has great effect on $\sim$ *telogen* length, but little effect on $\sim$ *anagen* length. For instance, whereas increasing $\Delta_+$ greatly increases $\sim$ *telogen* length, it slightly increases $\sim$ *anagen* length (*Appendix 2—figure 19*). For different values of $\Delta_+$, increasing inhibitor level will always result in shorter $\sim$ *anagen* and longer $\sim$ *telogen*, although to different extents (*Appendix 2—figure 20*).

### Threshold $\Delta_-$
On the other hand, at the single HF level, increasing $\Delta_-$ greatly decreases $\sim$ *anagen* length while slightly decreasing $\sim$ *telogen* length (*Appendix 2—figure 21*). For different values of $\Delta_-$, increasing inhibitor level will always result in shorter $\sim$ *anagen* and longer $\sim$ *telogen*, although to different extents (*Appendix 2—figure 22*).

### Growth parameters $H^+, H^-, m_+, m_-$
Parameters $H^+, m_+$ determine the growth rate of a HF during phase P, while $H^-, m_-$ determine its degeneration rate during catagen. The HF dynamics are not sensitive to these parameters. With fixed activator and inhibitor levels, changing either of these four parameters have little effect on $\sim$ *anagen*/$\sim$ *telogen* length (*Appendix 2—figure 23*). Increasing inhibitor level will always result in shorter $\sim$ *anagen* and longer $\sim$ *telogen*, and the pattern does not change much under different values of these parameters (*Appendix 2—figure 24*).

### Maximum HF growth length $h_{\max}$
At the single HF level, increasing $h_{\max}$ will decrease $\sim$ *telogen* length, with a relatively slight effect on $\sim$ *anagen* length (*Appendix 2—figure 25*). For different values of $h_{\max}$, increasing inhibitor level will always result in shorter $\sim$ *anagen* and longer $\sim$ *telogen*, although to different extents (*Appendix 2—figure 26*).

At HF wave level, we test the effect of $h_{\max}$ in noise-free 3D model on a $60 \times 100$ domain, with parameter values given in *Appendix 2—table 1* (3D dorsal column), 3 and 5, except $\lambda_A = \lambda_I = 0$, and $\Delta_+ = 2.6$ or $2.7$ as shown above. At $t = 0.1$, we give a one time increase in the activator production ($I_A \rightarrow I_A + 3000$) for all HFs on the top row ($y = 100$) to generate steady head-to-tail HF waves. We tracked the time when the second HF wave reaches $y = 60$ and $y = 40$ (denoted as $T_{60}^2$ and $T_{40}^2$); when it leaves $y = 40$ (denoted as $\overline{T}_{40}^2$); and when the

third HF wave reaches $y = 40$ (denoted as $T_{40}^3$). We measured the wave speed, wave interval and wave length in the following way:

$$\text{WaveSpeed} = \frac{20}{T_{40}^2 - T_{60}^2}$$
$$\text{WaveInterval} = T_{40}^3 - T_{40}^2$$
$$\text{WaveLength} = \text{WaveSpeed} \cdot (\overline{T}_{40}^2 - \underline{T}_{40}^2)$$

for different values of $h_{\max}$. Simulation results show that longer HF (i.e., larger $|h_{\max}|$) results in a longer wave interval and a wider wave, but the wave speed is hardly affected (**Appendix 2—figure 27**).

## Total available receptors $R_{\text{tot}}^A$ and $R_{\text{tot}}^I$

As we discussed in section Section 5, the total amount of activator and inhibitor receptors results in different ~anagen and ~telogen lengths, and we use this property to model the difference between dorsal and ventral HFs. In **Figure 2F**, the dorsal-ventral interaction is simulated with $R_{\text{tot}}^A = R_{\text{tot}}^I = 15$ for the dorsal and $R_{\text{tot}}^A = R_{\text{tot}}^I = 50$ for the ventral domains, which leads to an interaction lasting for 2–3 cycles. To test the effects of $R_{\text{tot}}^A$ and $R_{\text{tot}}^I$ on the dorsal-ventral interaction behavior, we keep $R_{\text{tot}}^A = R_{\text{tot}}^I = 15$ for the dorsal while using $R_{\text{tot}}^A = R_{\text{tot}}^I = 20, 30, 40$ for the ventral domain.

Simulations show that with $R_{\text{tot}}^A = R_{\text{tot}}^I = 20, 30, 40$ for the ventral domain, the HF wave interactions during the first 3 cycles are similar to **Figure 2F** where $R_{\text{tot}}^A = R_{\text{tot}}^I = 50$ for the same domain. However, during later cycles, with large ventral $R_{\text{tot}}^A$ and $R_{\text{tot}}^I$, there is clearly a delay between ventral and dorsal HF wave development (**Appendix 2—figure 28**, **29**, **30**).

## Comparison with FitzHugh-Nagumo (FHN) model

The FitzHugh-Nagumo model is a 2D model that describes an excitable medium, and it has also been used in the study of HF behavior (**Murray et al., 2012**). Below we provide a detailed comparison between the FHN model and our model. While there are many similarities between the two models in predicting experimental observations, we do find there are several important differences between them at both single and population HF levels. The codes and parameters of the FHN model are based on the original ones from **Murray et al. (2012)**.

### Single HF

Main differences between our model and the FHN model at the single HF level:

1. FHN model cannot predict reasonable anagen and telogen times as observed in experiment. **Plikus et al. (2008b)** reported 14 and 28 days for anagen and refractory telogen for WT mice, respectively, while competent anagen may vary in the range of 0–60 days. It is reported in **Murray et al. (2012)** that such a time scale cannot be reached: '...in order for the proposed model to yield stochastic excitations at a rate in agreement with observations we find that Tc (competent telogen time) in the model must be of the order of $10^5$ days...' 'A notable feature of our simulations is that competent telogen times must be of the order of $\sim 10^5$ days such that the frequency of stochastic excitations across a population of follicles is comparable to the population scale patterns measured by Plikus et al. However, at the single follicle scale Plikus et al. have measured competent telogen times in the range of 0–60 days. In fact, when we used these much shorter competent telogen times the simulations are dominated by stochastic excitations in a manner inconsistent with population scale measurements from wild-type mice (data not shown).'

However, we would like to point out that *Chen et al. (2014)* use the FHN model and reach biologically observed time scales, as shown in *Chen et al. (2014)*, with a specific choice of parameters, the authors reach $T_{PA}$ (propagating anagen) $\sim 15$ days, $T_R$ (refractory telogen) $\sim 30$ days and $T_C$ (competent telogen) $\sim 10$ days. However, we fail to reproduce their results without further information about parameter values. Moreover, to reach such a state it needs to sacrifice the sensitivity of telogen length - in particular, that of competent telogen length - with respect to the change of activator level.

2. The dynamics of the FHN model are either periodic or excitable. In the periodic regime there is no competent telogen phase, while in the excitable regime sustained excitability relies on noise or exogenous activating factors, which makes the competent telogen too sensitive to certain core parameters. Our model, however, in its periodic dynamics provides competent telogen with variable length, and noise and/or external factors affect the length of different phases instead of determining the system moves on to the next cycle. In the FHN model, when the nullcline $\dot{w} = 0$ intersects with the other nullcline $\dot{v} = 0$ in its first piece, that is $v_{ss} < v_0$, which is the scenario shown in *Appendix 2—figure 31A*, the system becomes an excitable medium. This is the scenario discussed in *Murray et al. (2012)*, and to end the competent telogen, the noise has to be strong enough to pass over the threshold (*Appendix 2—figure 31B*). For such an excitable scenario, the mean time spent in competent telogen is estimated by *Murray et al. (2012)*. When the noise is not strong enough, the system will stay in competent telogen for a long time. In contrast, when the inhibitor nullcline $\dot{w} = 0$ intersects the activator nullcline $\dot{v} = 0$ in its second piece (*Appendix 2—figure 31C*), that is $v_0 < v_{ss} < v_1$, the system will deterministically undergo cycles, since there is no more threshold to pass over that relies on the help of noise. *Appendix 2—figure 31D* shows the deterministic trajectory (black line) which clearly goes in cycles, unlike the stochastic trajectory in *Appendix 2—figure 31B* which has to wait in competent telogen for a long time waiting for excitation.

To study a HF system with all four phases (PARC) using the FHN model, it has to be in the excitable regime and depend on noise and/or external factors to get the system excited. In contrast, most simulations of our model are done in the periodic regime, which also provide the competent regime that is missing from a periodic FHN model. In addition, the length of the competent regime in the periodic regime of our model can vary a lot, allowing noise greatly to adjust the length of different phases (*Figure 1C*).

3. Subjected to changes in the activator/inhibitor production rate ($I/J$), in FHN model refractory telogen reacts differently to experimental observations, while competent telogen acts too sensitively. *Plikus (2012)* showed that anagen, refractory and competent telogen lengths change associated to activator and/or inhibitor level changes. Such an activator/inhibitor level change can be simulated via changes of activator/inhibitor production rate $I/J$, or the positive feedback in the activator pathway $\alpha_1$ — the latter will be discussed later.

Regarding change of $I$, we did not find published data. For change of $J$, as is pointed out in *Murray et al. (2012)* or shown in *Chen et al. (2014)*, 'a notable feature of the (FHN) model is that a decrease in the parameter $J$ does not yield the reduced refractory telogen times observed by Plikus et al.'

We reproduced the phase time to $I/J$ relation using FHN model based on parameter values provided in *Murray et al. (2012)*, shown in *Appendix 2—figure 32* for different values of $\alpha_1$.

When the positive feedback in the activator pathway $\alpha_1$ is not too low, refractory telogen length (dashed line, $T_R$) barely changes with activator/inhibitor production rate change $I/J$ (*Appendix 2—figure 32B,C,E,F*), which is stated in *Murray et al. (2012)* and against experimental observations. For small $\alpha_1$, refractory telogen length increases as activator production rate increases (*Appendix 2—figure 32A*) and decreases as inhibitor production rate increases (*Appendix 2—figure 32D*), also against biological observations.

Competent telogen length (dash-dot line, $T_C$) reacts dramatically to activator/inhibitor production rate changes: it decreases from the order of $10^{4-5}$ down to 0 quickly as $I$ increases (*Appendix 2—figure 32A,B,C*) or as $J$ decreases (*Appendix 2—figure 32D,E,F*). The change rate of $T_C$ is affected by the value of $\alpha_1$: with bigger $\alpha_1$ (*Appendix 2—figure 32C,F*) the change rate is slightly milder comparing to extremely abrupt change when $\alpha_1$ is small (*Appendix 2—figure 31A,D*). In fact, indicating by *Appendix 2—figure 32*, only within a small window of $(I, J)$ could $T_C$ stay in a reasonable scale. Although the reaction trends coincide with experimental observations – $T_C$ decreases as $I$ increases or $J$ deceases; however, it might be too sensitive to changes of $I$ and/or $J$, comparing to a reasonable change of $T_{PA}$. If we restrict to the small window of $(I, J)$ where $T_C$ behaves reasonable, then $T_{PA}$ reacts too insensitive to changes in $I$ and/or $J$.

4. The FHN model is unable to reach an extended anagen scenario. Our model gives two equilibrium regimes besides the excitable regime (*Figure 1C*): one relates to extended telogen achieved with high inhibitor or low activator level (*Appendix 2—figure 5A*), which can also be achieved in FHN model; however, the other equilibrium regime relates to extended anagen achieved with low inhibitor or high activator level (*Appendix 2—figure 5B*), this scenario may explain phenomena such as human scalp hair anagen, which can last for years, or the long hair observed in angora rabbits. The FHN model is unable to simulate such an extended anagen scenario.

5. In order to reach both short anagen and telogen in the FHN model, it needs many parameter values changed, while in our model we can easily achieve this state by changing one or two parameters. Thus, our model is a convenient tool to model the interactions between dorsal and ventral domains. In vivo experiments support fast hair cycle dynamics in the ventral domain, more specifically, ventral HFs are short in both anagen and telogen lengths. While in both the FHN and our model we discover parameters that result in short anagen and long telogen (or vice versa), in our model we also have parameters that can be modified to alter anagen and telogen length, $R^A_{\mathrm{tot}}$ and $R^I_{\mathrm{tot}}$ (*Appendix 2—figure 8*, *9*), or $h_{\mathrm{max}}$ (*Appendix 2—figure 25*). In contrast, in the FHN model we do not know if by adjusting one or two parameters we could obtain such a fast cycling HF type. This directly affects the simulation of the interaction between ventral and dorsal domains.

Below are some similarities between the FHN model and our model at a single HF level:

1. Subjected to changes in the activator/inhibitor production rate ($I/J$), anagen length reacts similar to what is predicted by our model. Despite the noncoincidence of refractory and competent telogen to experimental observations and our model predictions, in FHN model, anagen length increases as activator production rate increases (*Appendix 2—figure 32A-C*) and decreases as inhibitor production rate increases (*Appendix 2—figure 32D-F*), which qualitatively coincides with our model predictions. Moreover, the change rate of anagen length to $I$ or $J$ is not greatly affected by the value of $\alpha_1$.

2. Anagen/telogen length change subjected to the positive feedback in the activator pathway $\alpha_1$ coincides with experimental observations. In the FHN model, if the anagen/telogen length change is adjusted via $\alpha_1$ instead of the production rates $I/J$, the simulation results coincide with experimental observations. However, since in our model there is no direct cross-talk between the activator and inhibitor pathways, there is no way to directly compare the two models regarding such a positive feedback effect.

As summarized in *Murray et al. (2012)*, 'increased positive feedback in the activator dynamics results in the observed phenomena of faster activation wavefronts, shorter refractory and competent telogen times, unchanged anagen time, increased spontaneous initiation rates and the emergence of target patterns at the population scale.'

We reproduce the simulation results of phase length to positive feedback ($\alpha_1$) relation under different values of $I$ and $J$, shown in *Appendix 2—figure 33*.

As $\alpha_1$ increases, anagen length (solid line, $T_{PA}$) increases slightly while refractory telogen length (dashed line, $T_R$) clearly decreases; both coincide with experimental observations.

Depending on the base values of $I$ and $J$, competent telogen length shows totally different reactions to changes in $\alpha_1$. When $J$ is not high compared to $I$ (**Appendix 2—figure 33A-C**), the competent telogen length mostly stays at 0, that is, the system dynamics are periodic, absent of competent telogen. On the other hand, when J is relatively high to $I$, that is, the inhibitor level is high in the system (**Appendix 2—figure 33D**), competent telogen increases dramatically as $\alpha_1$ decreases. That is, in a system where inhibitor level ($J$) is high, as the positive feedback in the activator pathway ($\alpha_1$) gets weaker, the mean time spent for the system waiting to get excited increases extremely fast. However, there are certain values of $I$ and $J$ that allow the system - in particular, competent telogen reacts in a similar way to what is observed experimentally or predicted in our model (**Appendix 2—figure 33E,F**). In these scenarios, competent telogen length decreases at a rate comparable to that of refractory telogen as $\alpha_1$ increases.

In **Appendix 2—figure 34**, we present the nullclines and trajectories in $v - w$ phase corresponding to values of $I$, $J$ and $\alpha_1$ in **Appendix 2—figure 33**.

## HF wave

Main differences between our model and the FHN model at the HF wave level:

1. In the context of the heterogeneous two-domain model (i.e. dorsal and ventral domains), the dorsal head-to-tail asynchrony resulted from dorsal-ventral interactions breaks down within approximately 2–3 cycles in our model, which is consistent with the biological observations. Similar breakdowns take many more cycles to achieve in the FHN model. Experimental observations show ventral-dorsal interactions leading to dorsal head-to-tail asynchrony in the first 2–3 cycles, which quickly degrade into spontaneous initiations yet possessing bi-lateral symmetry.

Our model simulated similar patterns (**Figure 2F**) - the dorsal head-to-tail asynchrony resulted from dorsal-ventral interactions lasts for approximately 2–3 cycles, after which it turns into spreading waves with centers located on the ventral-dorsal borders showing bilateral , together with spontaneous initiations. Using the FHN model, we see the similar results; however, the dorsal-ventral interaction always lasts for more cycles (**Appendix 2—figure 35**, dorsal-ventral interactions persists for five cycles). We tried different parameter values but the dorsal-ventral interaction always lasts for more than three cycles, as is observed experimentally or predicted by our model.

A possible reason behind this might be that while our model allows us to modulate $\sim anagen$ and $\sim telogen$ lengths by adjusting $R_{\text{tot}}^A$ and $R_{\text{tot}}^I$ to approximate ventral HF behavior, that is shorter $\sim anagen$ and $\sim telogen$, the FHN model has no easy way to modulate them. Therefore, we have to set HFs in ventral domain to have shorter telogen in sacrifice of slightly longer anagen compared to dorsal HFs.

2. In the wave-breaker simulations, we are able to investigate two scenarios using our model: a physical region cut-off from the HF wave domain (like eyes, ears), or a region permeable to signal molecules with disabled HF growth (by wound, for example). The FHN model is unable to distinguish these two scenarios.

Please refer to Appendix 2-Wave breaker, **Figure 7A** and **Appendix 2—figure 18**.

3. Experiments also imply ventral HFs are shorter than dorsal HFs, combined with our single HF analysis that within a region, short HF have shorter anagen and telogen (**Appendix 2—figure 25**), in the future extension of our study, we will investigate the interaction between dorsal and ventral domains characterized by different HF length. The FHN model does not permit for this.

4. In a homogeneous domain with HFs near the head having temporal phase advancement built-in, both models reproduce the head-to-tail hair cycle asynchrony during early cycles, which will degenerate into random initiations with no bilateral symmetry (*Appendix 2— figure 14*, *15*, *Appendix 2—video 3*, *Appendix 2—figure 36*). However, the early head-to-tail steady pattern fades quickly in the FHN model while in ours it degenerates gradually. Simulations show that in our model simulations, even after 10 cycles, the wave may still resemble the early head-to-tail pattern. This is because we are sitting in a periodic state with non-zero competent telogen length, and noise is still able to initiate waves allowing the system show excitable properties, and the memory of an early wave pattern can persist through later waves.

On the other hand, in the FHN model, the early head-to-tail pattern may be lost as early as the 2nd cycle. *Appendix 2—figure 36* shows the time course of a typical simulation by the FHN model, in this simulation, the head-to-tail pattern persists for the first three cycles, while stochastic initiation takes over starting in the fourth cycle. The reason for such a quick loss of the early memory in the FHN model is because for the early head-to-tail pattern to survive, the system should possess a periodic property; however, the system is in an excitable regime where stochastic effects dominate, hence the stochastic initiations quickly wash away the head-to-tail pattern.

Below are some similarities between the FHN model and our model at HF wave level:

1. In a homogeneous domain, if all HFs are synchronized in telogen, they have equal probability of anagen initiation, in both models we see stochastic initiations dominate and there is no bilateral symmetry (*Appendix 2—figure 11, 12*, *Murray et al., 2012*).

2. In a homogeneous domain, when HFs near the head have higher built-in levels of activator signaling, both models show highly successive waves, that is target-like pattern (*Appendix 2—figure 13*, Figure S4 in *Murray et al., 2012*).

## Supplementary figures

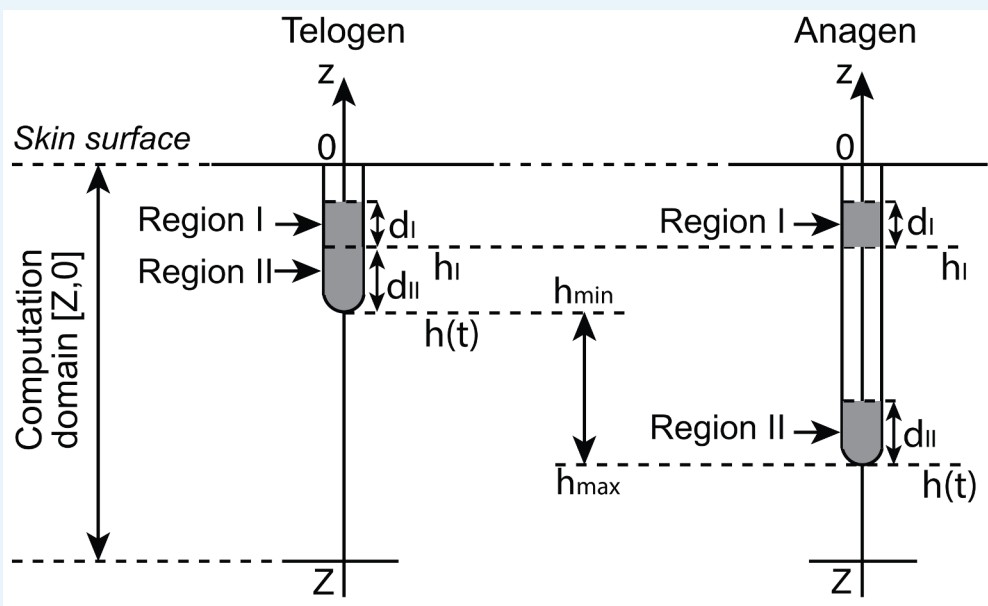

**Appendix 2—figure 1.** Geometry of a single hair follicle.

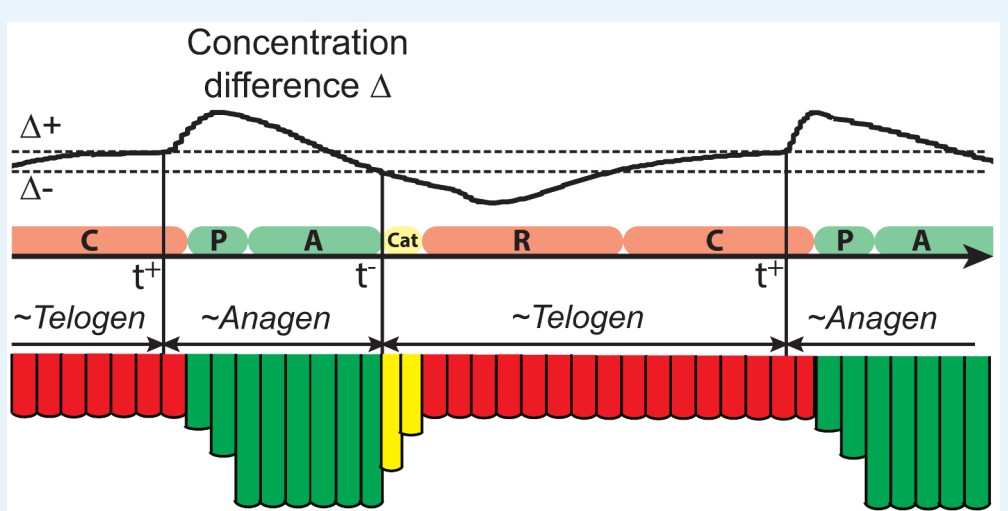

**Appendix 2—figure 2.** Illustration of the hair growth cycle phases.

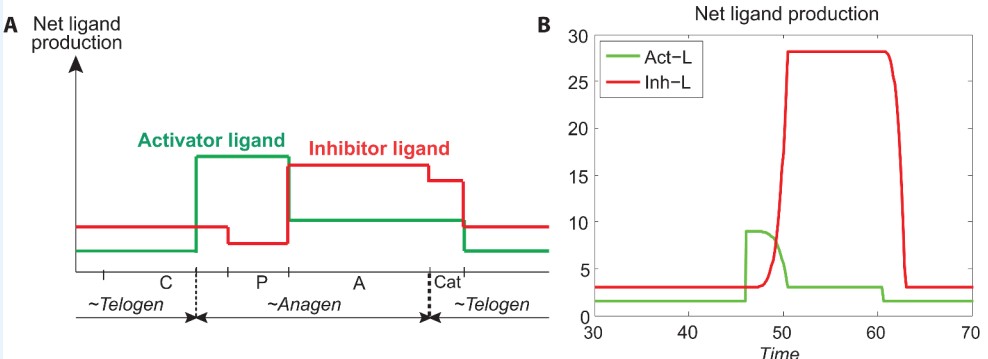

**Appendix 2—figure 3.** Temporal patterns of activator and ligand production. (**A**) The temporal patterns of activator (green line) and inhibitor ligands (red line) produced from *Appendix 2—table 4* and calculated by Appendix 2-*Equations (8)*. (**B**) The temporal patterns of activator and inhibitor ligands produced from Appendix 2-*Equations (9, 10)*. Comparing (**A**) to (**B**), we see they follow the similar patterns, indicating that our modeling of activator and ligand productions (Appendix 2-*Equations (9, 10)*) is reasonable.

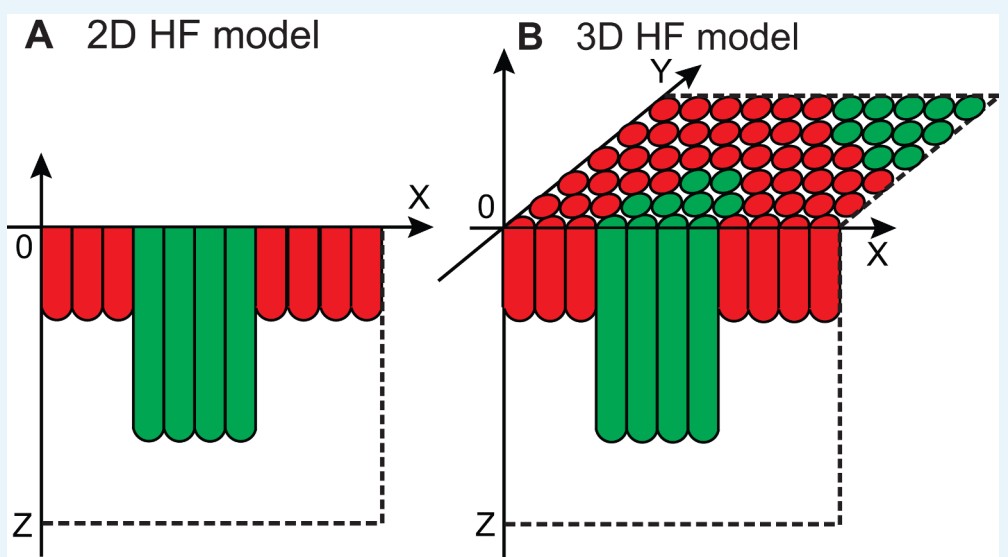

**Appendix 2—figure 4.** Illustration of 2D and 3D HF model.

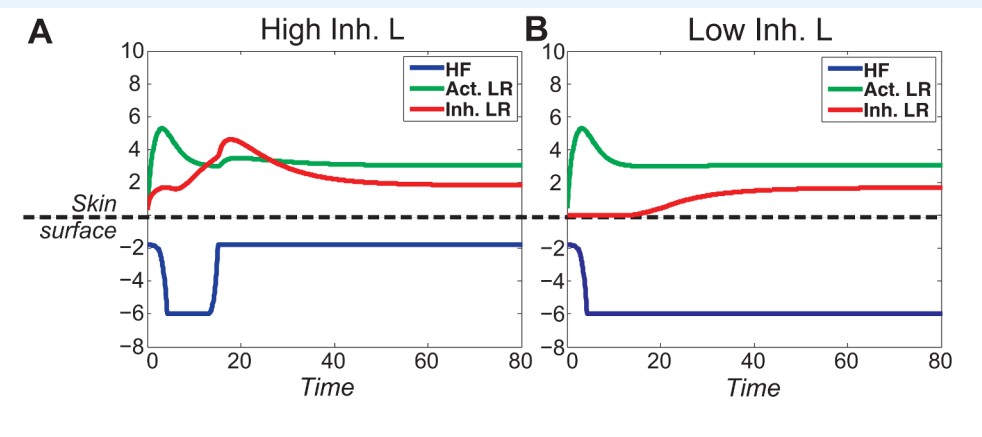

**Appendix 2—figure 5.** When activator/inhibitor signals are either too strong or too weak (**Figure 1C**, grey regions), the excitability breaks down and instead of cycling, the HF reaches a stable equilibrium state. (**A**) When inhibitor is very strong, signal difference per se cannot reach the threshold $\Delta_+$ required for spontaneous excitation, and the HF equilibrates in a prolonged telogen-like state (**Figure 1C**, Equilibrium II). (**B**) When inhibitor is very low, it fails to catch up with the activator, and the HF equilibrates in a prolonged anagen-like state instead (**Figure 1C**, Equilibrium I).

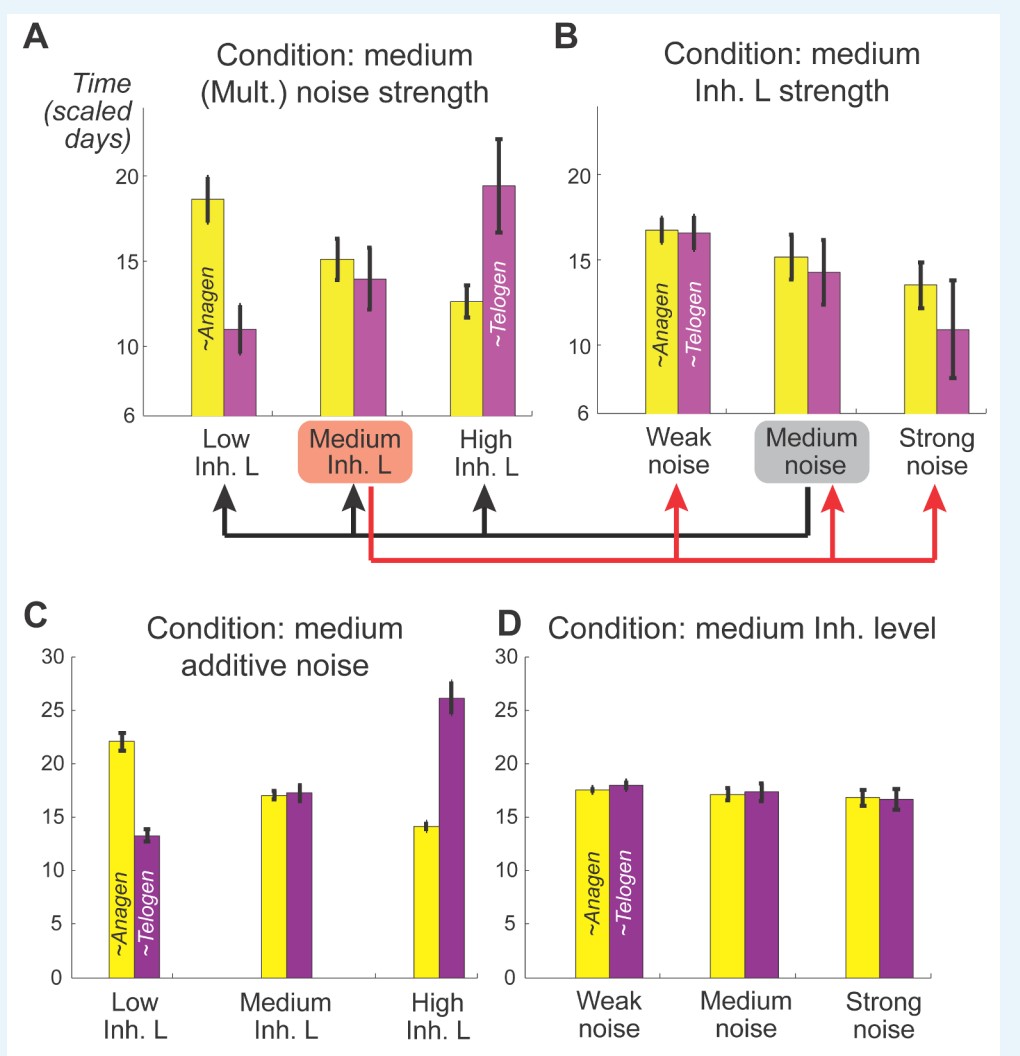

**Appendix 2—figure 6.** Effects of inhibitor ligand level and noise on ~ *anagen* and ~ *telogen* lengths. (**A, B**) Mean and variances of ~ *anagen* and ~ *telogen* durations as functions of different inhibitor ligand levels and different levels of multiplicative noise. (**A**) Here, multiplicative noise of medium strength was added to both activator and inhibitor equations (Appendix 2-*Equations (3, 4)*). Similar changes in the mean of ~ *anagen* and ~ *telogen* durations as those in the noise-free simulation on *Figure 1C*. (**B**) With a medium level of inhibitory signaling, increasing the multiplicative noise strength shortens both ~ *anagen* and ~ *telogen* durations and increases their variances. (**C, D**) Mean and variances of ~ *anagen* and ~ *telogen* durations as functions of different inhibitor ligand levels and different levels of additive noise, the results are qualitatively the same with (**A, B**) obtained from mutiplicative noise. All simulations are based on 50 samples.

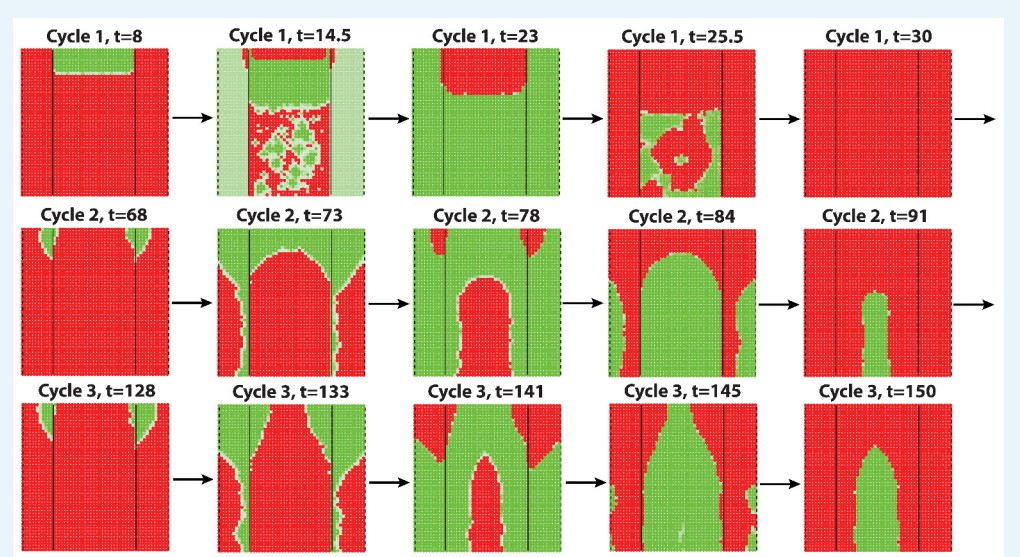

**Appendix 2—figure 7.** Dorsal-ventral interaction simulated with additive noise with noise strength $\lambda_A = \lambda_I = 0.05$. Similar to the results obtained with multiplicative noise (*Figure 2F*) and biological observations, dorsal-ventral interaction lasts for mostly 2 or 3 cycles, after which it breaks into spontaneous waves showing some bilateral symmetries.

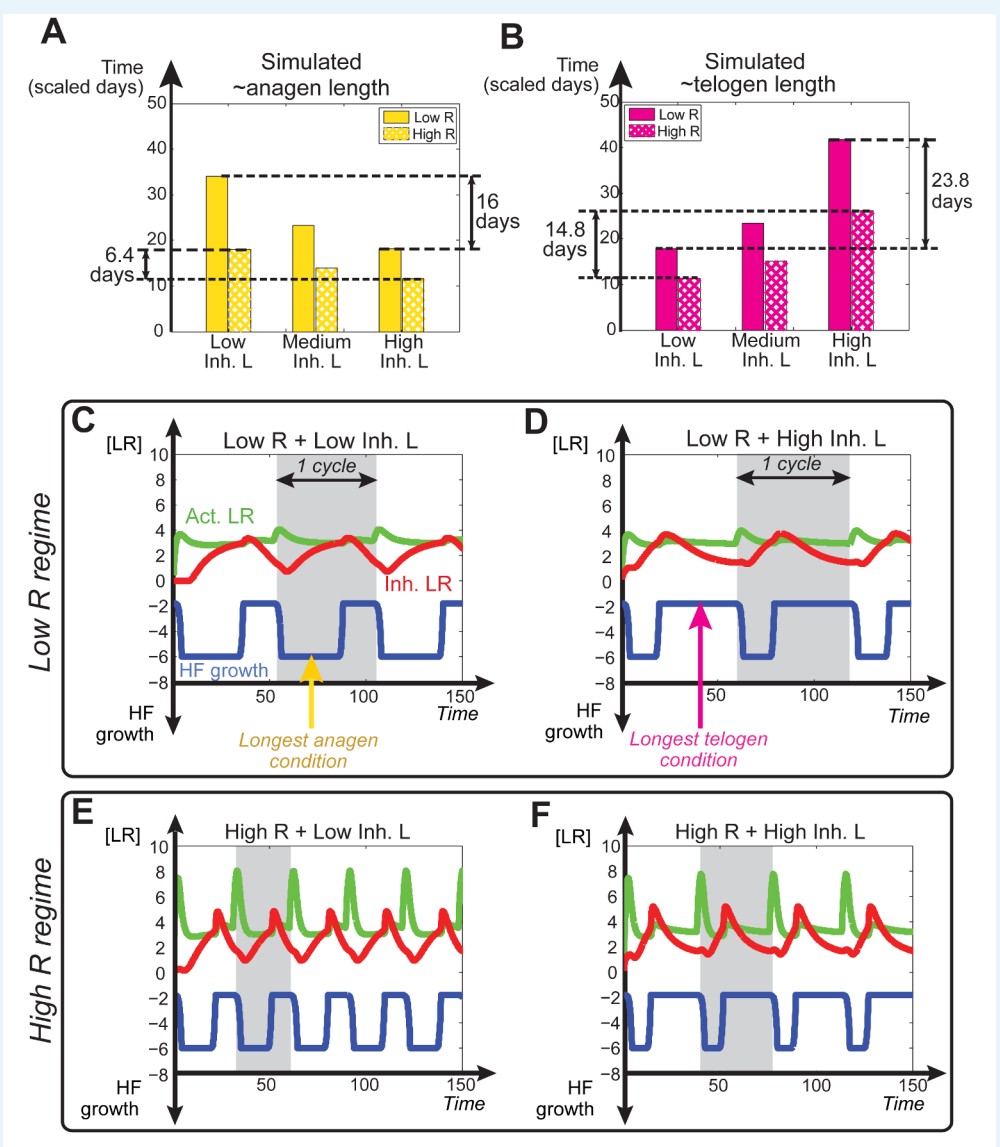

**Appendix 2—figure 8.** Dependence of ~ *anagen* and ~ *telogen* length on receptor levels. In noise-free simulations we adjusted the levels of total available activator and inhibitor receptors ($R_{tot}^A$ and $R_{tot}^I$). This allowed differential HF sensitivity to the same ligand level changes. (**A**, **B**) Comparative changes in ~ *anagen* and ~ *telogen* duration under the conditions of low (solid bars) and high levels (checkered bars) of $R_{tot}^A = R_{tot}^A = R$ under three different levels of inhibitor signaling. (**C-F**) Temporal dynamics of noise-free simulations performed using varying levels of inhibitor signaling under low $R$ (**C, D**) vs. high $R$ regime (**E, F**).

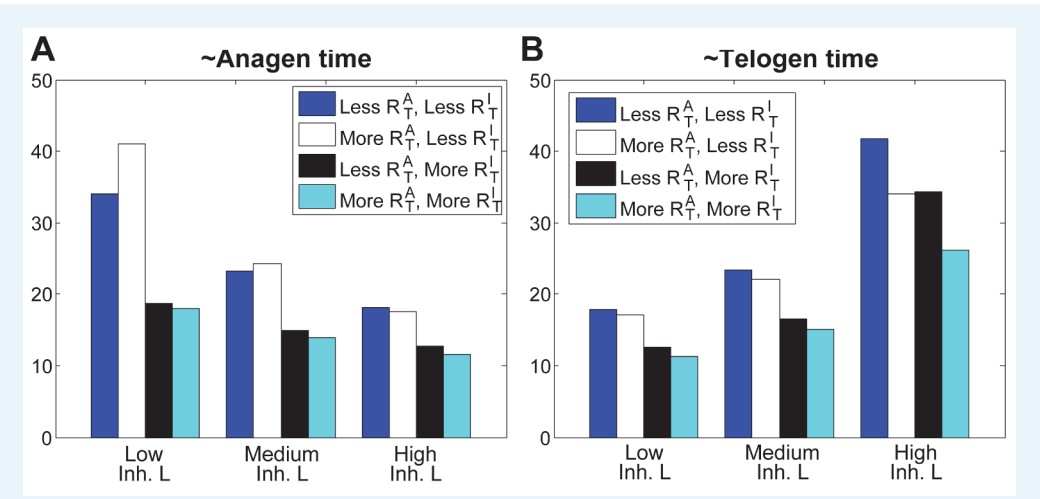

**Appendix 2—figure 9.** Separate roles of $R_{tot}^A$ and $R_{tot}^I$ on $\sim$ anagen and $\sim$ telogen lengths.

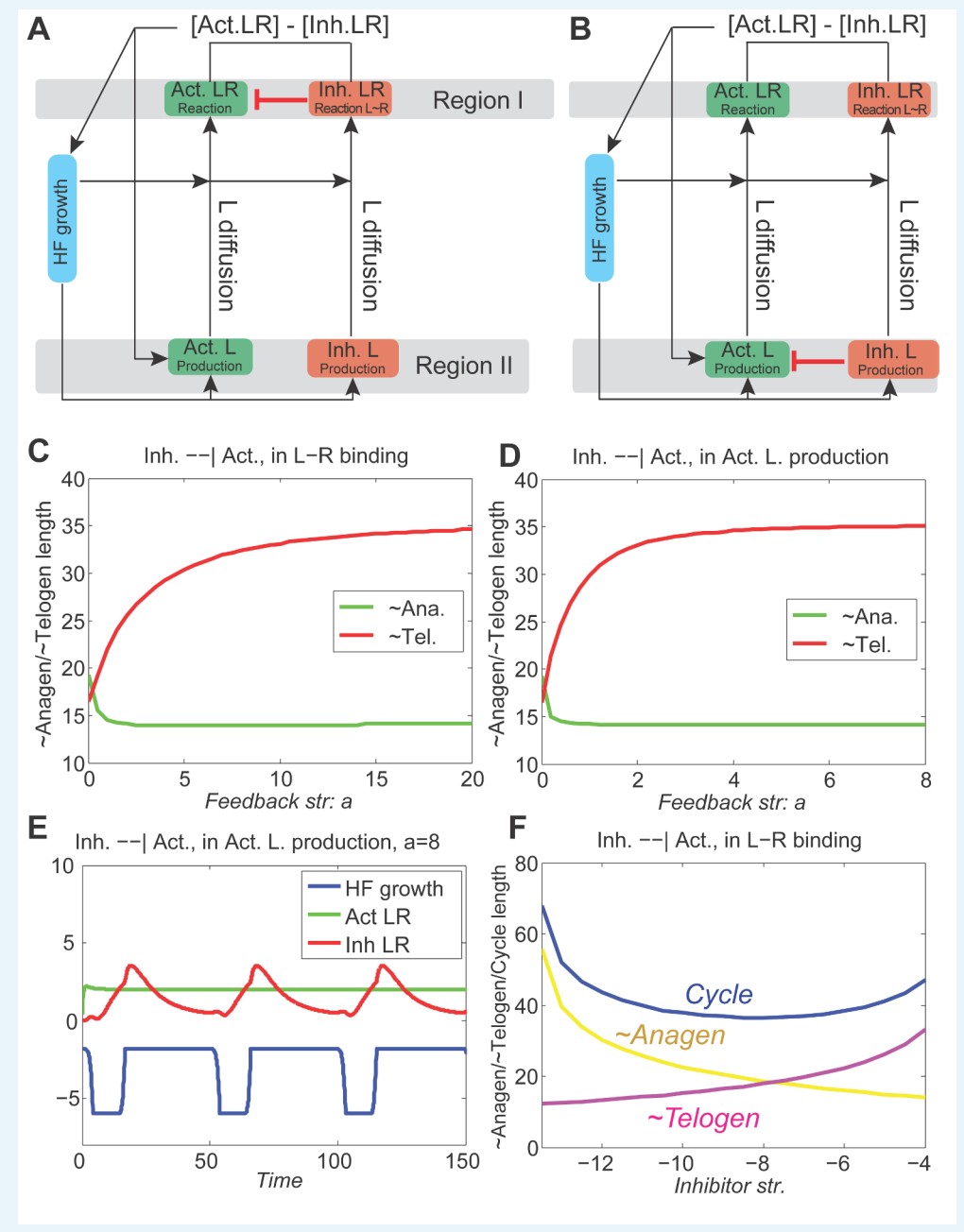

Appendix 2—figure 10. With inhibitory pathway from inhibitor to activator incorporated to the model, the dynamics are still stable. (A, B) The inhibitor pathway is incorporated either as inhibitor LR inhibiting the activator ligand-receptor (A, Appendix 2-*Equation (14)*) or inhibitor ligands inhibiting the production of activator ligands (B, Appendix 2-*Equation (15)*). (C, D) In either type of inhibitory pathway, ~anagen (green line) is shortened and ~telogen (red line) is lengthened; moreover, as the inhibition becomes stronger (larger $a$), ~anagen length quickly drops to a minimum while ~telogen gradually reaches a maximum. (E) With very strong inhibition, the activator LR is pushed almost flat while the inhibitor LR still shows the hill-valley pattern, however, the HF growth (blue line) still cycles. (F) With the inhibitory pathway incorporated into our model (in the activator ligand-receptor binding with strength $a = 10$), changing the inhibitor ligand levels affects the ~anagen, ~telogen and cycle lengths in a qualitative similar way as no pathway incorporated (*Figure 1C*), although the excitable regime is being switched from $\beta_I \in [-9, 0.5]$ down to $\beta_I \in [-13, -4]$.

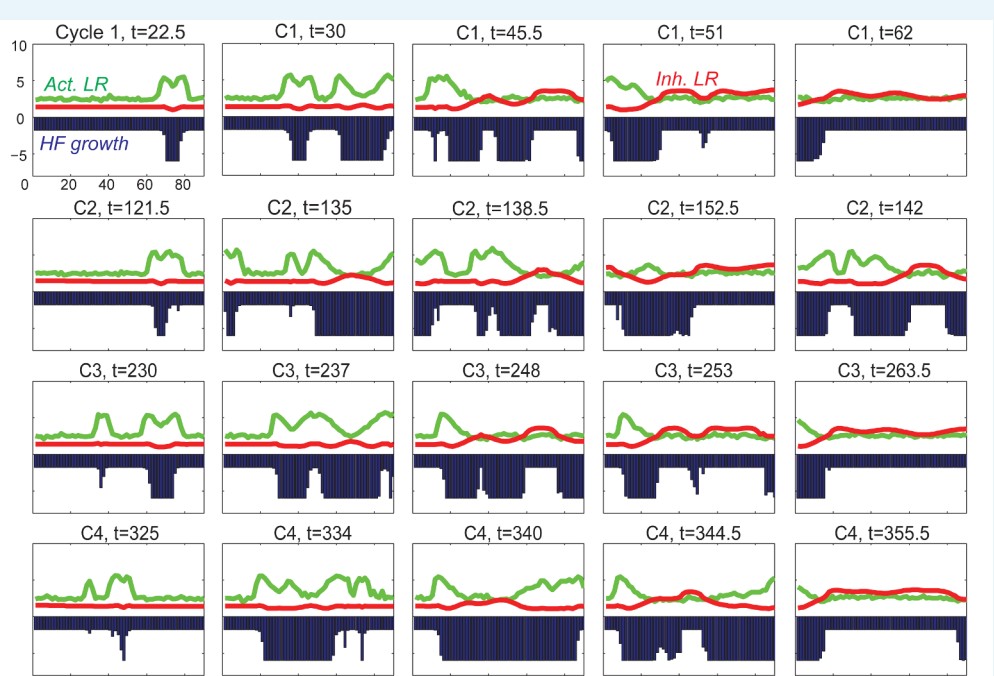

**Appendix 2—figure 11.** 2D simulation of dorsal HF waves, when all HFs have equal probability of anagen initiation, allowing spontaneous anagen initiations.

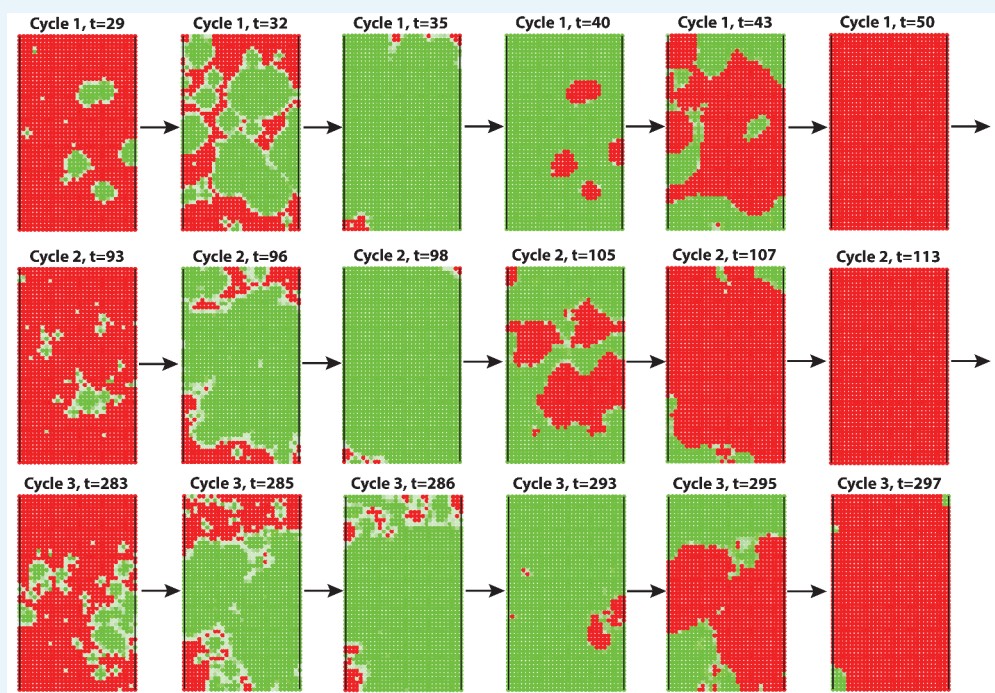

**Appendix 2—figure 12.** 3D simulation of dorsal HF waves, when all HFs have equal probability of anagen initiation, allowing spontaneous anagen initiations. Here we see spontaneous anagen initiations over the dorsal domain from the very first cycle, against the experimental observations of early head-to-tail wave degenerating into later spontaneous initiations

showing some bilateral symmetries. The full time course is shown in *Appendix 2—video 1*.
See *Appendix 2—tables 1 - 3* and *5* for parameter values, except $\lambda_A = \lambda_I = 0.035$.

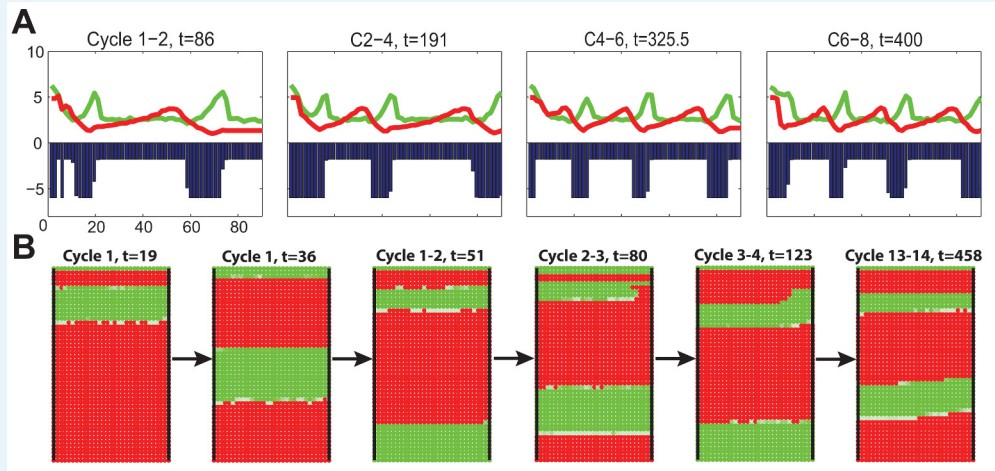

**Appendix 2—figure 13.** 2D (**A**) and 3D (**B**) simulations of dorsal HF waves, when HFs near the head have higher build-in levels of activator signaling. Here we see highly successive steady waves characterized as the 'target-like wave' pattern, also against experimental observations. The full time course of the 3D simulation is shown in *Appendix 2—video 2*. See *Appendix 2—tables 1 - 3* and *5* for parameter values, except $\lambda_A = \lambda_I = 0.02$ and $\Delta_+ = 1.8$ in the 3D simulation 13B.

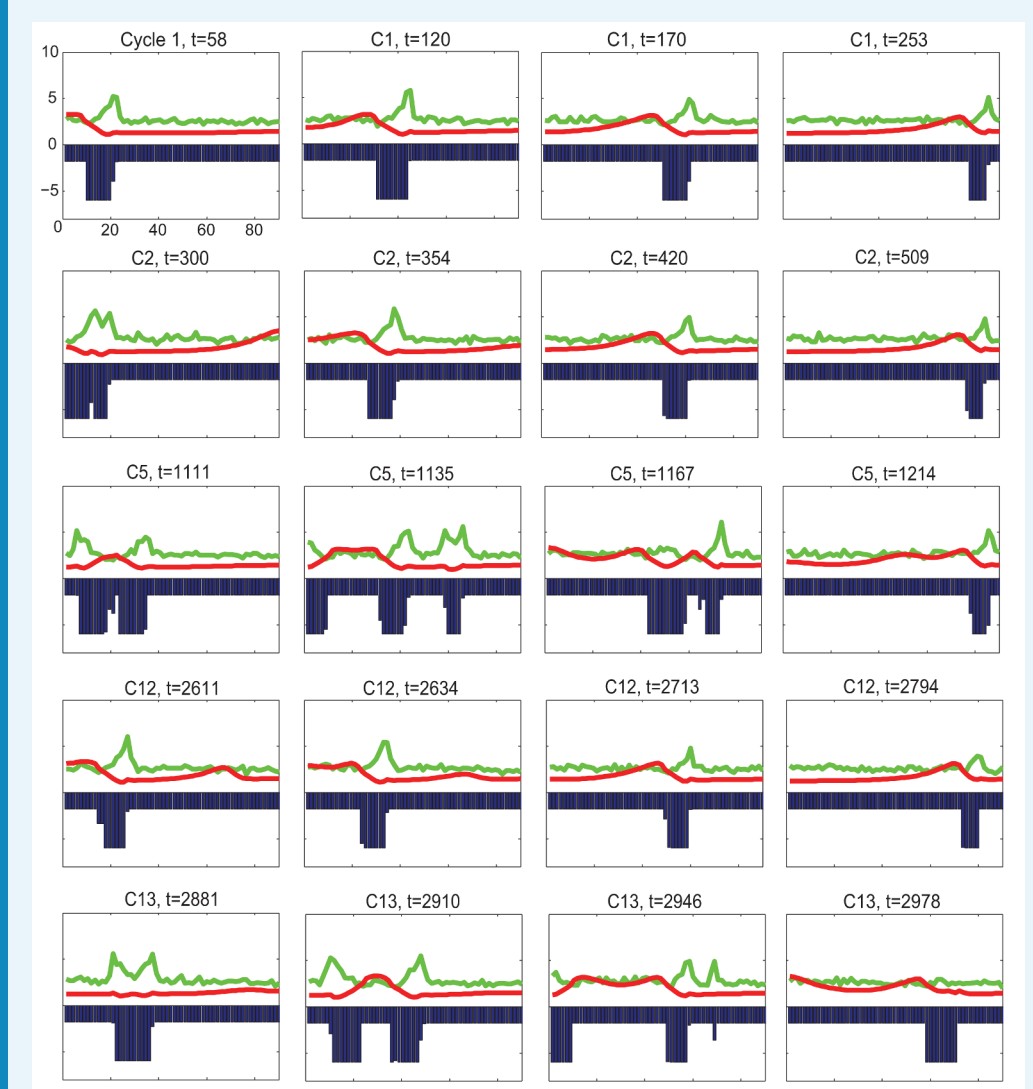

**Appendix 2—figure 14.** 2D simulation of dorsal HF waves, when HFs near the head have build-in temporal phase advancement.

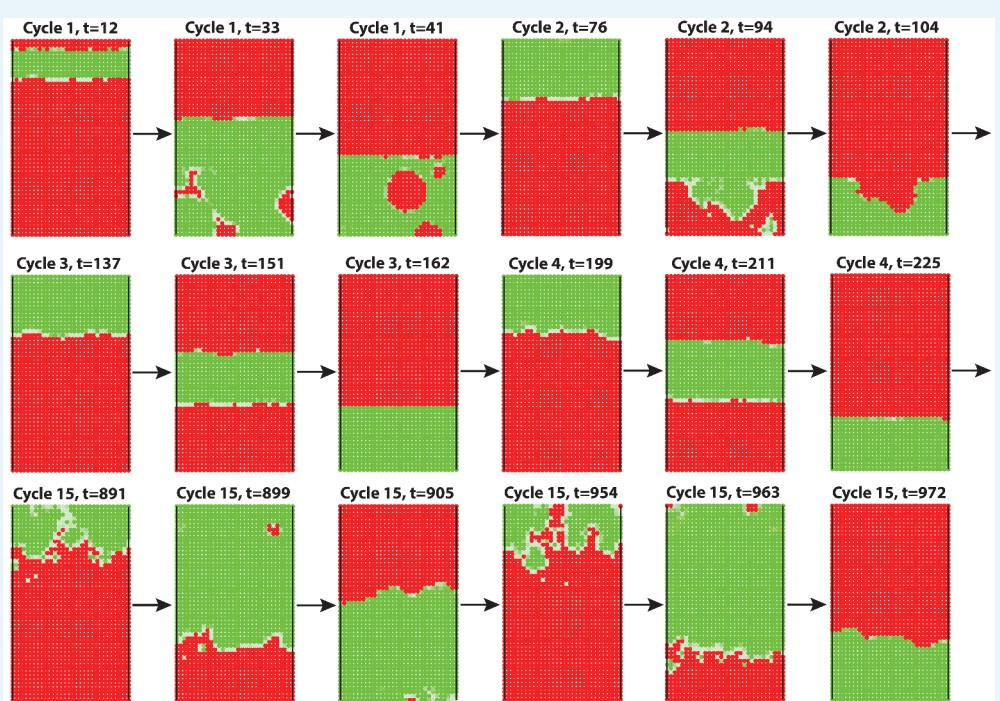

**Appendix 2—figure 15.** 3D modeling of HF growth pattern evolution with head-to-tail hair cycle asynchrony. Initial asynchrony was introduced into the field by assigning a group of HFs on one end with a one-time increase in activator ligands. Simulations faithfully reproduce head-to-tail asynchrony of the first few cycles. However, this asynchrony fails to degrade quickly, and persists during multiple consecutive cycles. In addition, no clear bilateral symmetry is observed. This differs from rapid HF growth pattern evolution observed in mice. Also see *Appendix 2—video 3* and see *Appendix 2—tables 1 - 3* and *5* for parameter values, except $\lambda_A = \lambda_I = 0.035$.

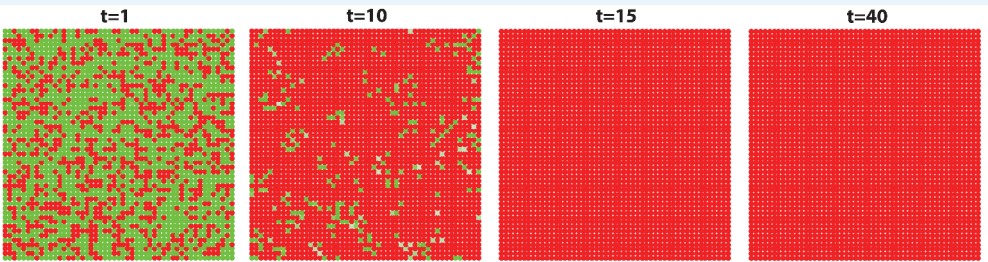

**Appendix 2—figure 16.** The high level of inhibitor in ear skin results in hair cycling stops and HFs equilibrate in an extended telogen. Inhibitor level is high in this whole hyper-refractory domain. Since initially the activator level is slightly higher than later, noise is able to randomly initiate the first HF growth cycle. After that, HFs stay in an extended telogen. See *Appendix 2—tables 1 - 3* and *5* for parameter values, except $\Delta_+ = 3$, $\lambda_A = \lambda_I = 0.05$, $\alpha_I^1 = 22$ and $\beta_A(t)$ given by Appendix 2-*Equation (16)*.

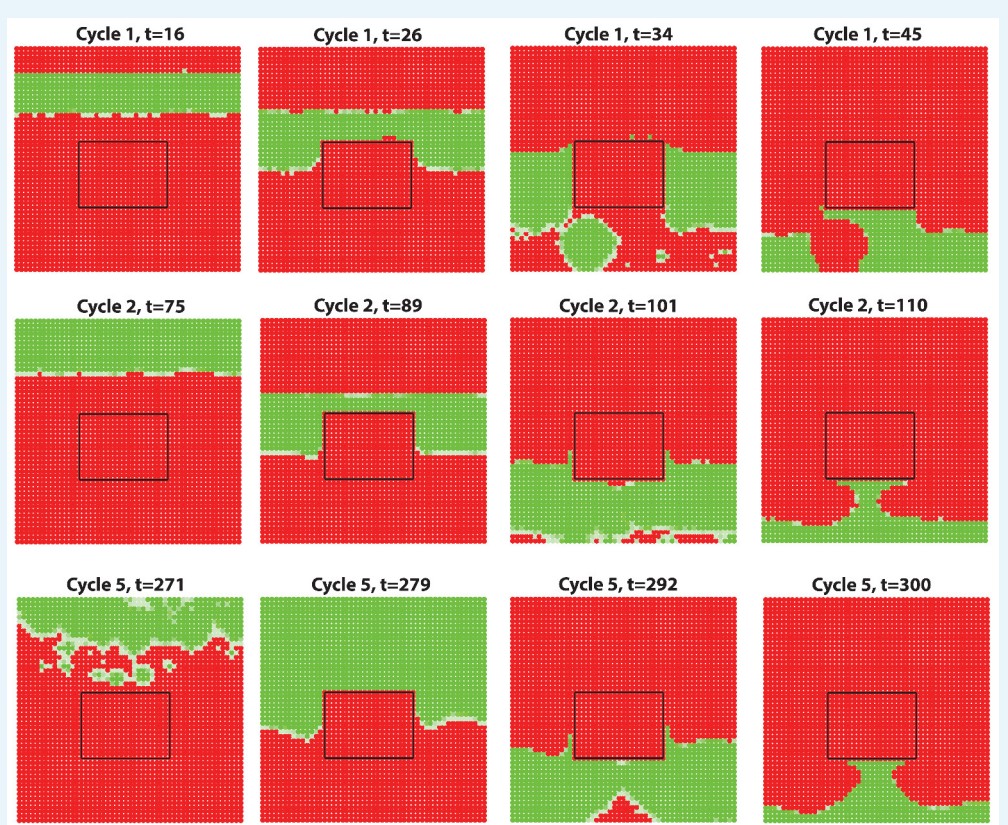

**Appendix 2—figure 17.** Anagen waves spreading from dorsal skin could not propagate into the ear skin due to the hyper-refractory state. We simulate a domain of dorsal skin, with a small patch (the region inside the box) having high inhibitor level in simulating the hyper-refractory ear skin. Steady HF waves propagate through the dorsal region for several cycles together with a few stochastic initiations, however, no HF wave is able to propagate into the ear skin due to the hyper-refractory state.

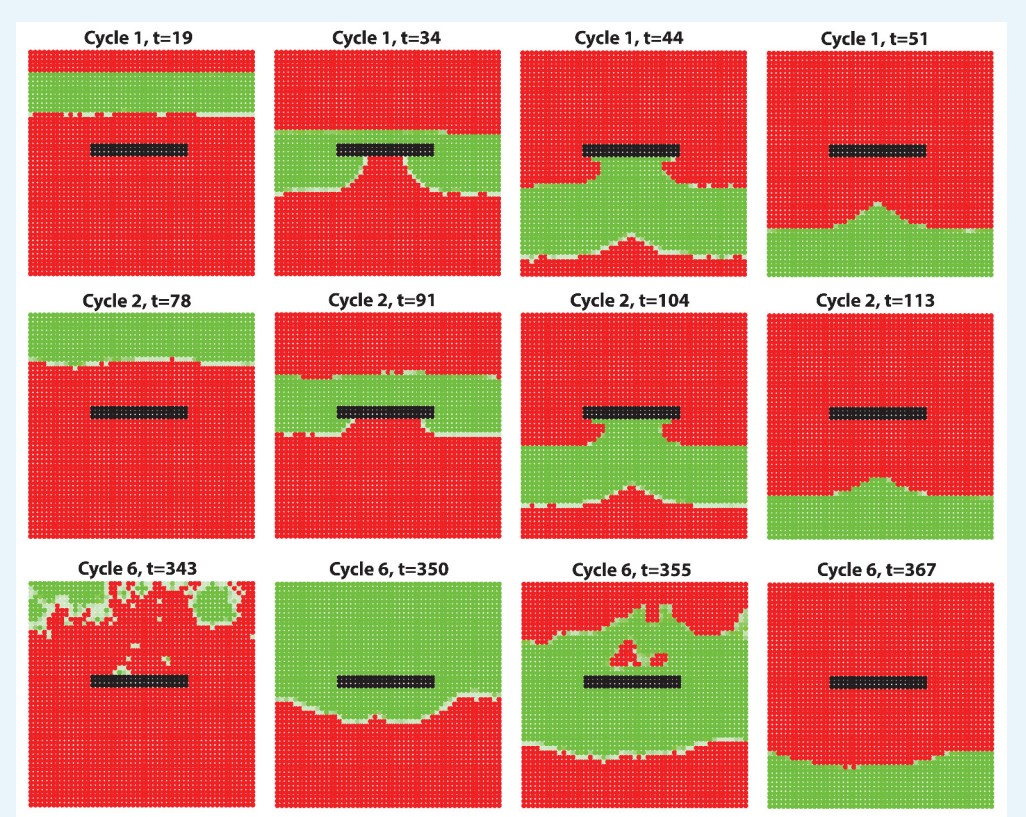

**Appendix 2—figure 18.** Simulation results of a wave-breaker region where HF growth is disabled while molecules are still allowed to diffuse in this region. The growth-impaired region creates a distortion-like effect, which breaks the HF growth wave propagating.

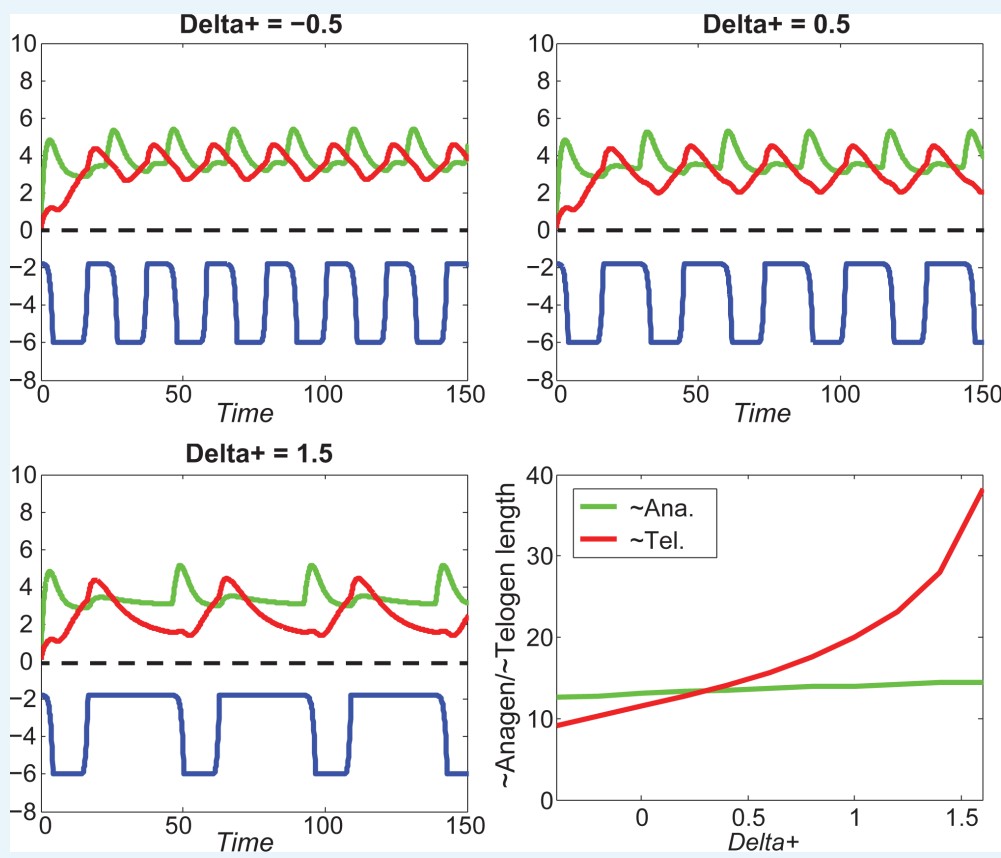

**Appendix 2—figure 19.** Increasing $\Delta_+$ greatly increases $\sim telogen$ length, slightly increases $\sim anagen$ length.

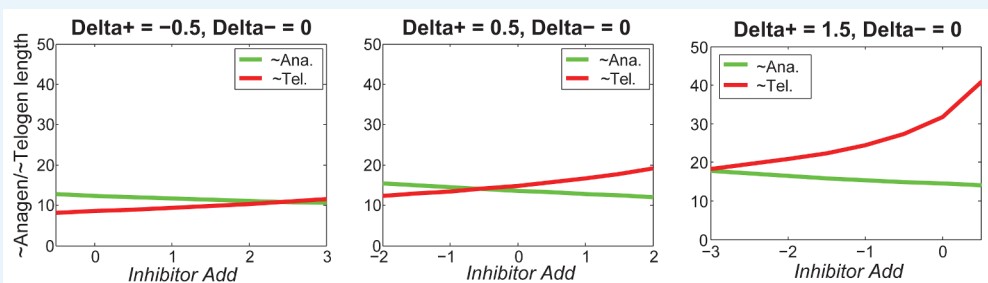

**Appendix 2—figure 20.** For different values of $\Delta_+$, increasing inhibitor level will always result in shorter $\sim anagen$ and longer $\sim telogen$, although to different extents.

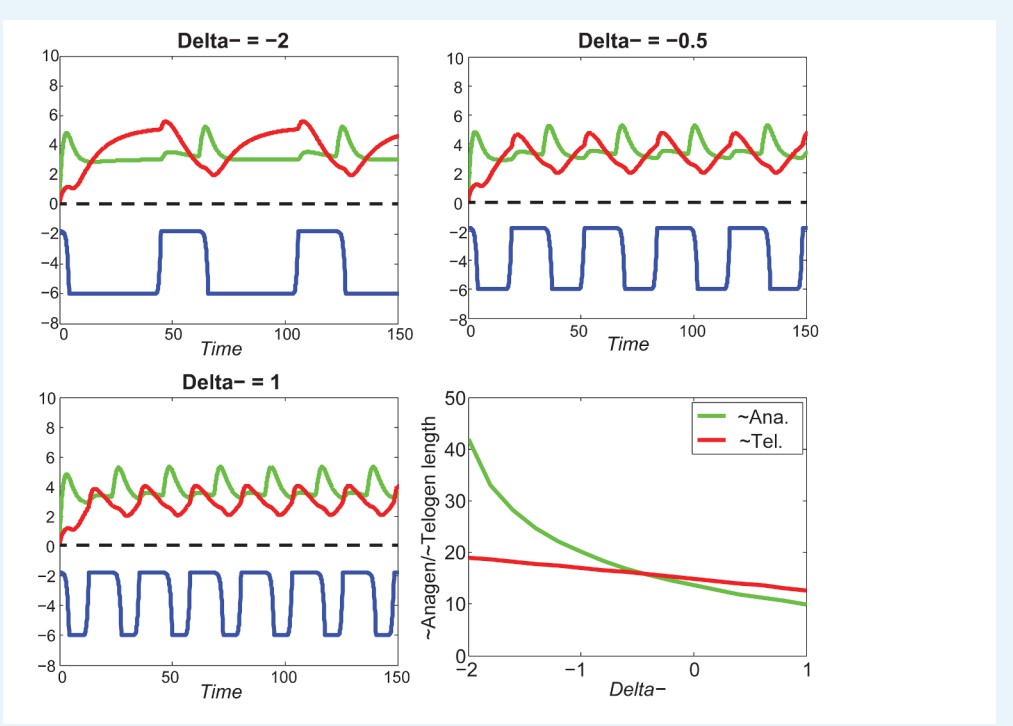

**Appendix 2—figure 21.** At the single HF level, increasing $\Delta_-$ greatly decreases $\sim$ *anagen* length while slightly decreasing $\sim$ *telogen* length.

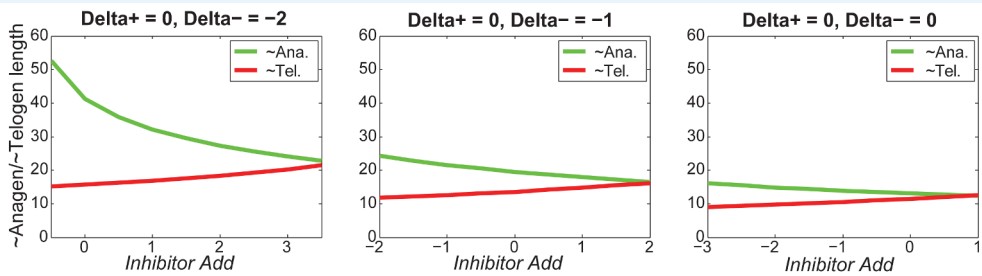

**Appendix 2—figure 22.** For different values of $\Delta_-$, increasing inhibitor level will always result in shorter $\sim$ *anagen* and longer $\sim$ *telogen*, although to different extents.

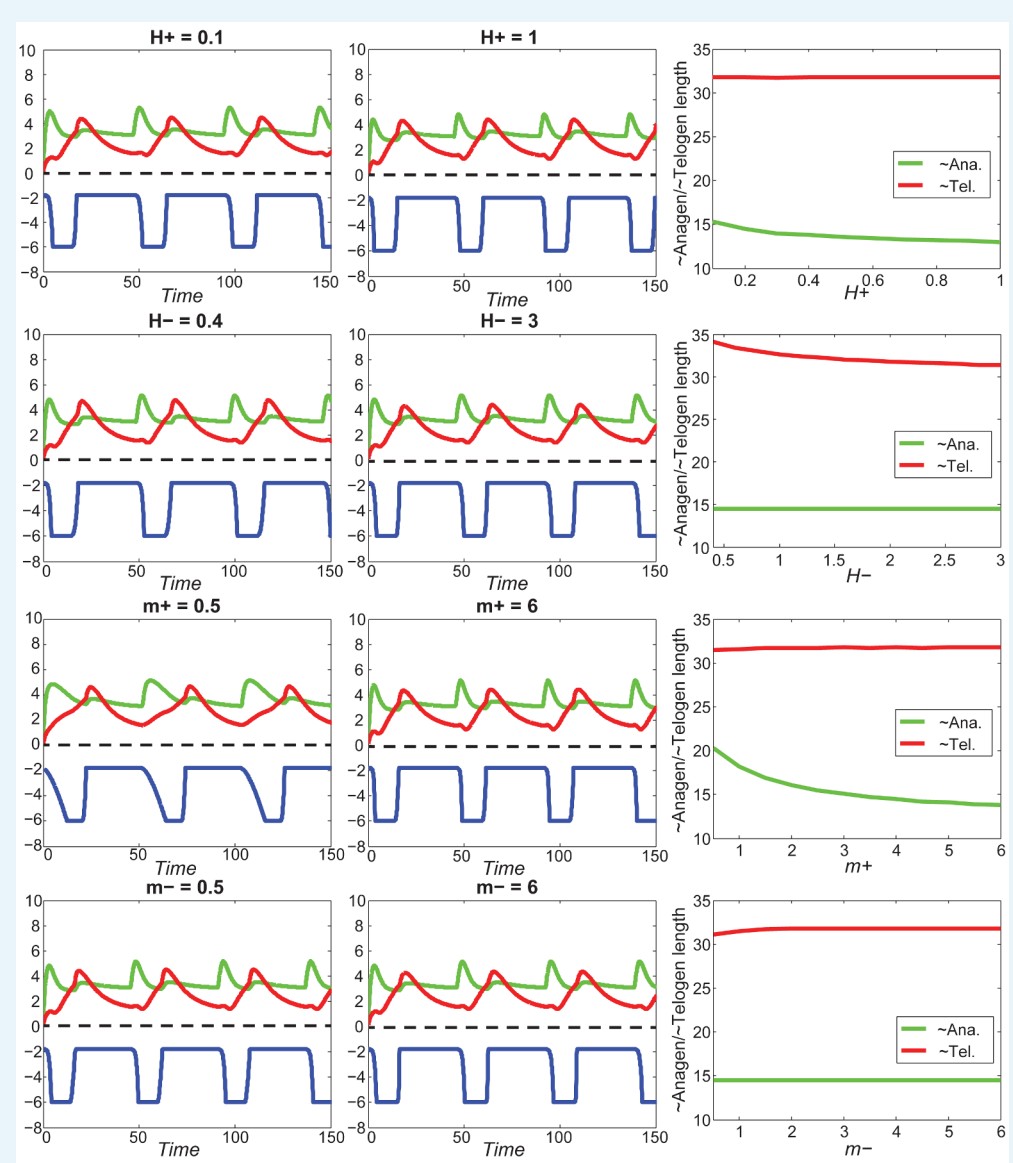

**Appendix 2—figure 23.** With fixed activator and inhibitor levels, changing either of the four parameters ($H^+, H^-, m_+, m_-$) has little effect on $\sim anagen/\sim telogen$ length.

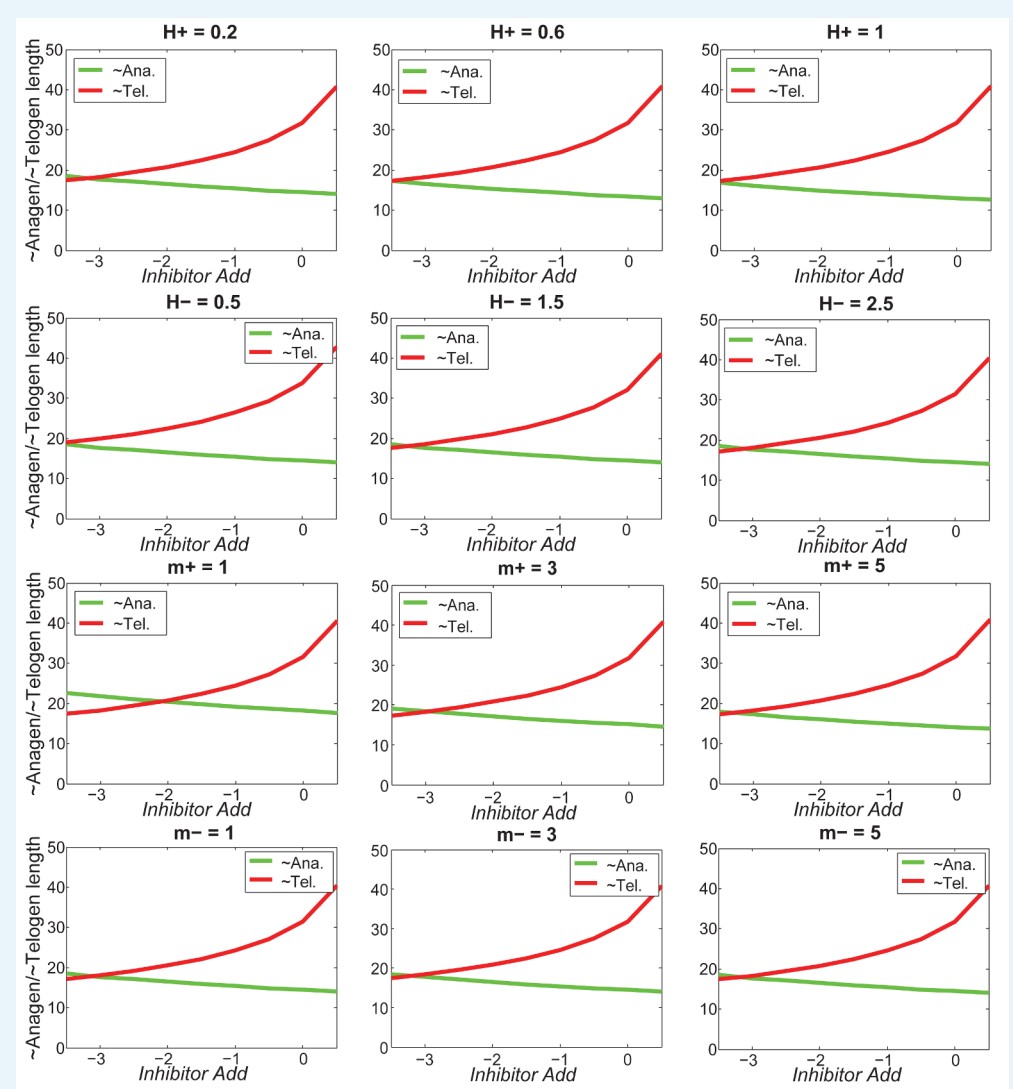

**Appendix 2—figure 24.** Increasing inhibitor level will always result in shorter ~ *anagen* and longer ~ *telogen*, and the pattern does not change much under different values of $H^+, H^-, m_+, m_-$.

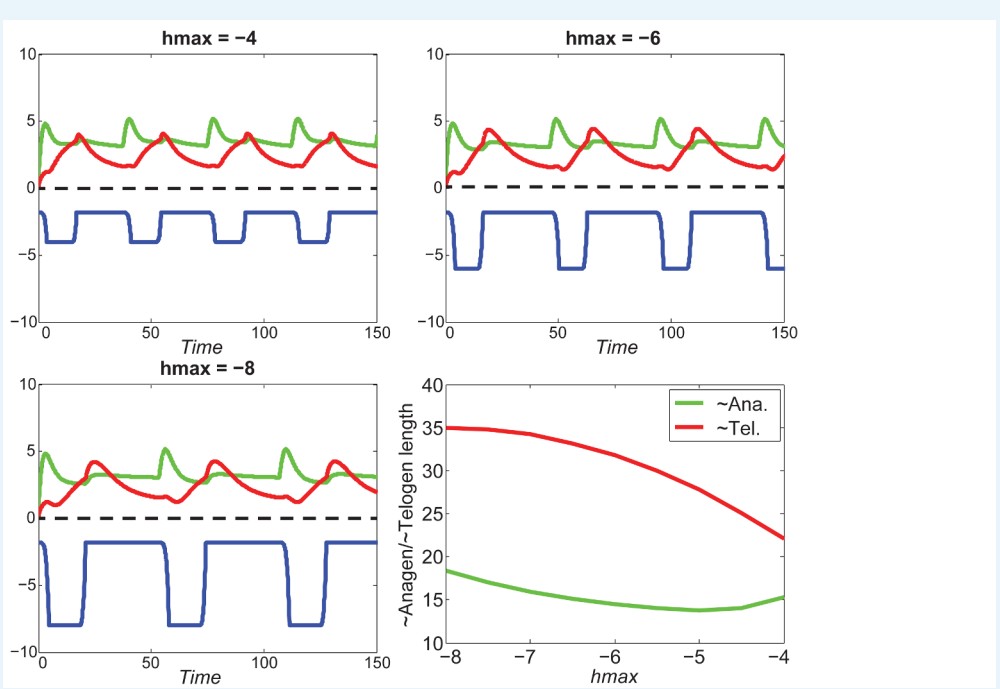

**Appendix 2—figure 25.** At the single HF level, increasing $h_{max}$ will shorten ~ *telogen* length, and will also shorten ~ *anagen* length within a certain region.

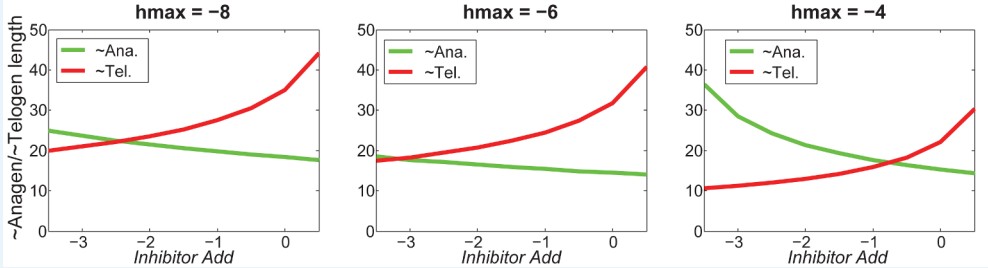

**Appendix 2—figure 26.** For different values of $h_{max}$, increasing inhibitor level will always result in shorter ~ *anagen* and longer ~ *telogen*, although to different extents.

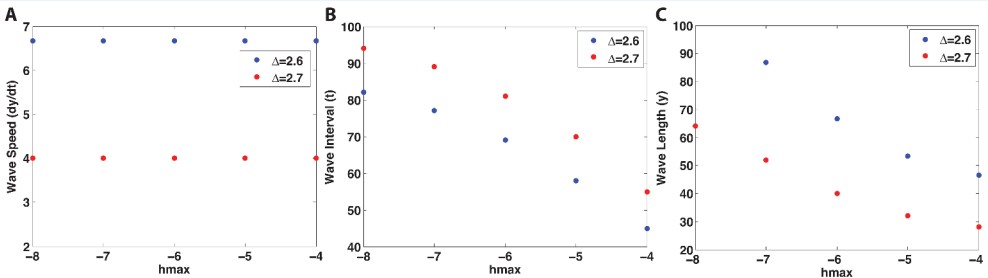

**Appendix 2—figure 27.** Larger $|h_{max}|$ (i.e., longer HF) results in a longer wave interval (**B**) and a wider wave (**C**), but the wave speed is hardly affected (**A**). Simulations are noise-free, on a $60 \times 100$ uniform domain, with two values of $\Delta_+ = 2.6$ or $2.7$.

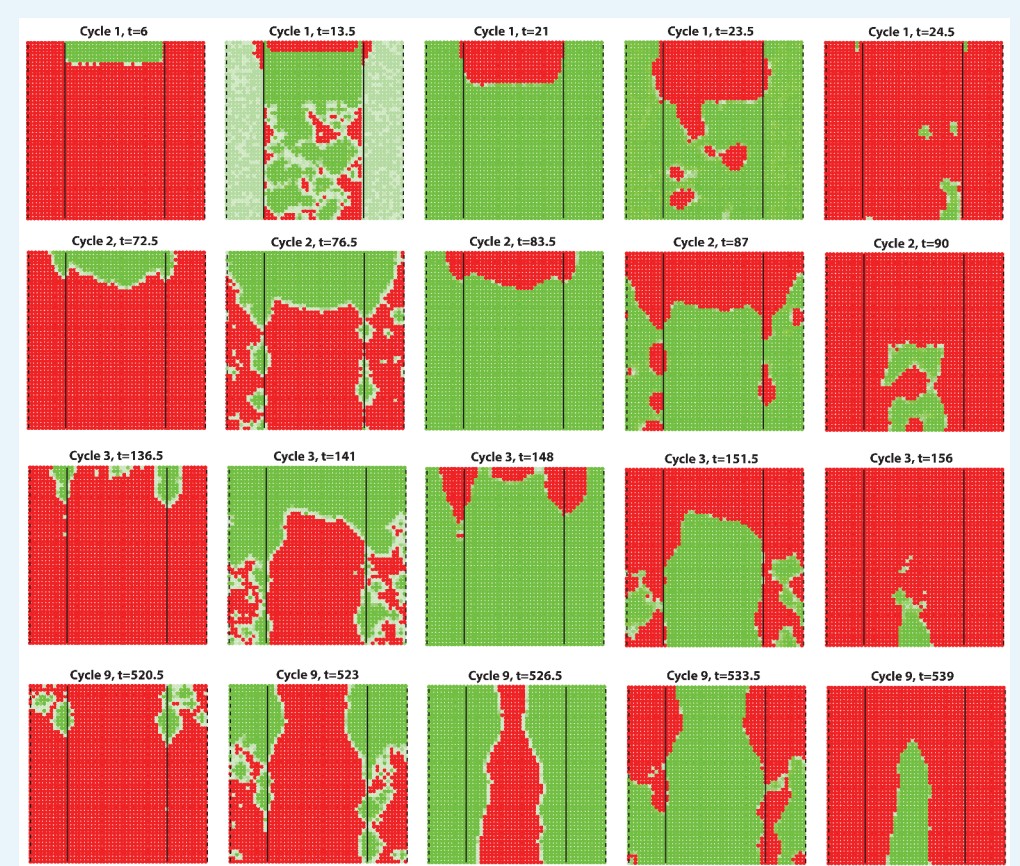

**Appendix 2—figure 28.** Dorsal-ventral interaction with $R^A_{\text{tot}} = R^I_{\text{tot}} = 15$ for the dorsal while $R^A_{\text{tot}} = R^I_{\text{tot}} = 20$ for the ventral. First few HF waves behave similar to *Figure 2F* where $R^A_{\text{tot}} = R^I_{\text{tot}} = 15$ for the dorsal while $R^A_{\text{tot}} = R^I_{\text{tot}} = 50$. However, with the 15/20 profile, there is no clear delay between dorsal and ventral HF wave development during later cycles.

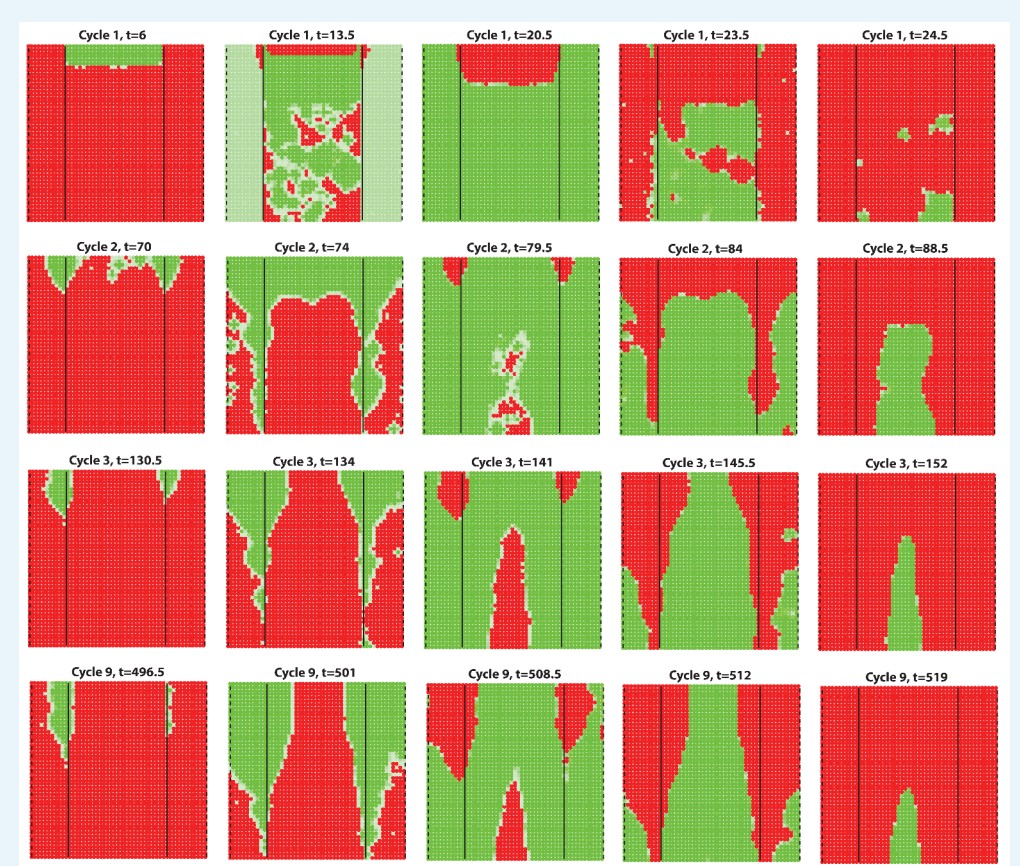

**Appendix 2—figure 29.** Dorsal-ventral interaction with $R^A_{tot} = R^I_{tot} = 15$ for the dorsal while $R^A_{tot} = R^I_{tot} = 30$ for the ventral. First few HF waves behave similar to **Figure 2F** where $R^A_{tot} = R^I_{tot} = 15$ for the dorsal while $R^A_{tot} = R^I_{tot} = 50$. With the 15/30 profile, there is a little delay between dorsal and ventral HF wave development during later cycles.

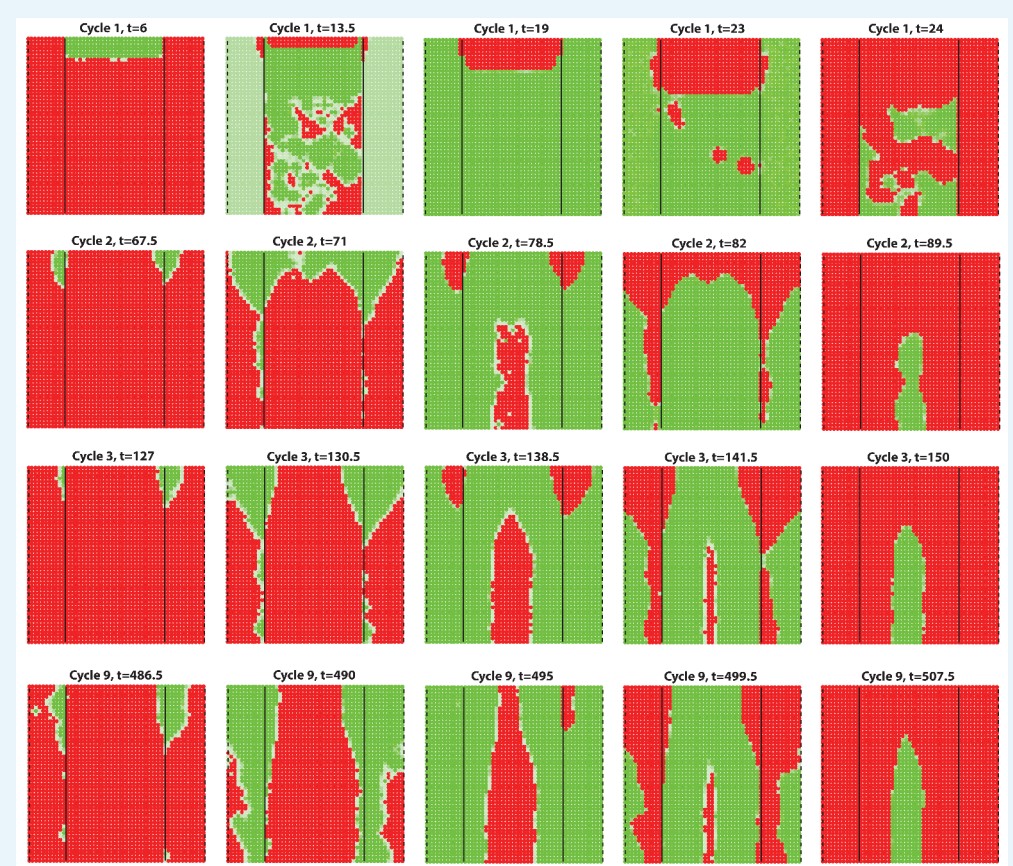

**Appendix 2—figure 30.** Dorsal-ventral interaction with $R_{\text{tot}}^A = R_{\text{tot}}^I = 15$ for the dorsal while $R_{\text{tot}}^A = R_{\text{tot}}^I = 40$ for the ventral. First few HF waves behave similar to **Figure 2F** where $R_{\text{tot}}^A = R_{\text{tot}}^I = 15$ for the dorsal while $R_{\text{tot}}^A = R_{\text{tot}}^I = 50$. With the 15/40 profile, there is clearly a delay between dorsal and ventral HF wave development during later cycles.

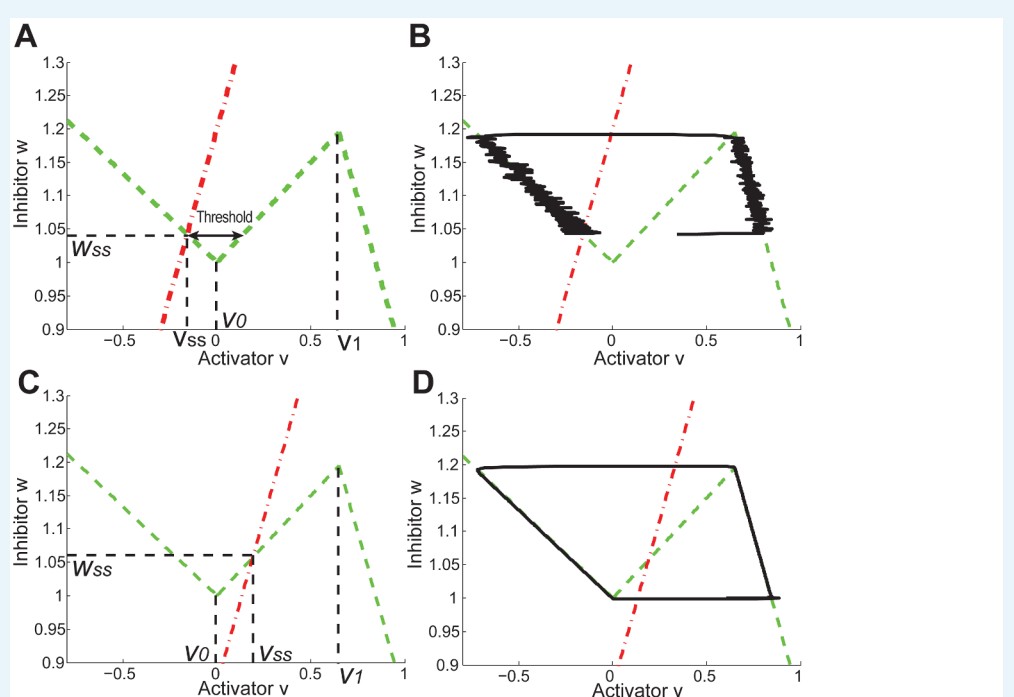

**Appendix 2—figure 31.** Periodic and excitable regimes in the FHN model. (**A**) Nullcline $\dot{w} = 0$ (red dash-dot line) intersects with $\dot{v} = 0$ (green dash line) in its first piece ($v_{ss} < v_0$). The system is excitable in this case. (**B**) With nullclines shown in (**A**), it requires strong enough noise to pass over the threshold so end the competent telogen. When noise is not strong enough, the system stays in competent telogen for a long time. Stochastic trajectory (black line) presented. (**C**) Nullcline $\dot{w} = 0$ (red dash-dot line) intersects with $\dot{v} = 0$ (green dash line) in its second piece ($v_0 < v_{ss} < v_1$). The system is periodic in this case. (**D**) With nullclines shown in (**C**), the system undergoes cyclic dynamics automatically without help from noise, as there is no threshold to pass over. Deterministic trajectory (black line) presented.

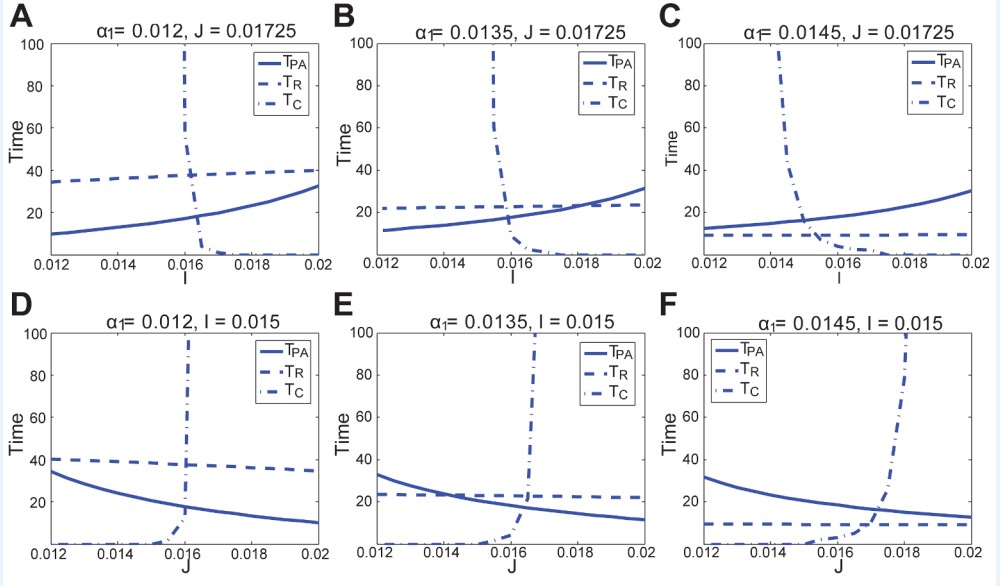

**Appendix 2—figure 32.** In the FHN model, changes in anagen length (solid line, $T_{PA}$), refractory telogen length (dashed line, $T_R$), competent telogen length (dash-dot line, $T_C$) to the activator

(*I*) or inhibitor (*J*) production rate, under different values of positive feedback in the activator pathway ($\alpha_1$).

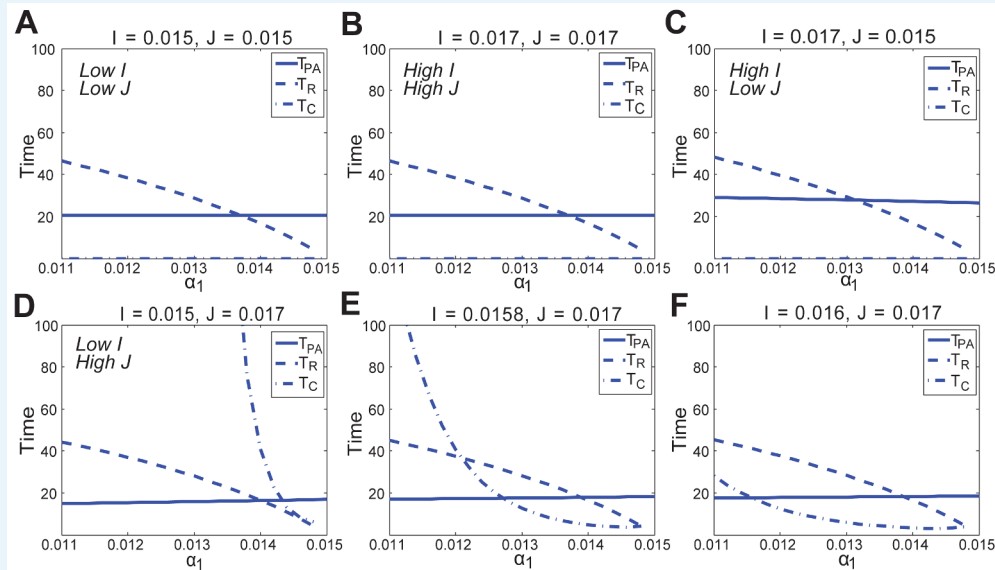

**Appendix 2—figure 33.** In the FHN model, changes in anagen length (solid line, $T_{PA}$), refractory telogen length (dashed line, $T_R$), competent telogen length (dash-dot line, $T_C$) to the positive feedback in the activator pathway ($\alpha_1$) under different values of activator (*I*) and inhibitor (*J*) production rates.

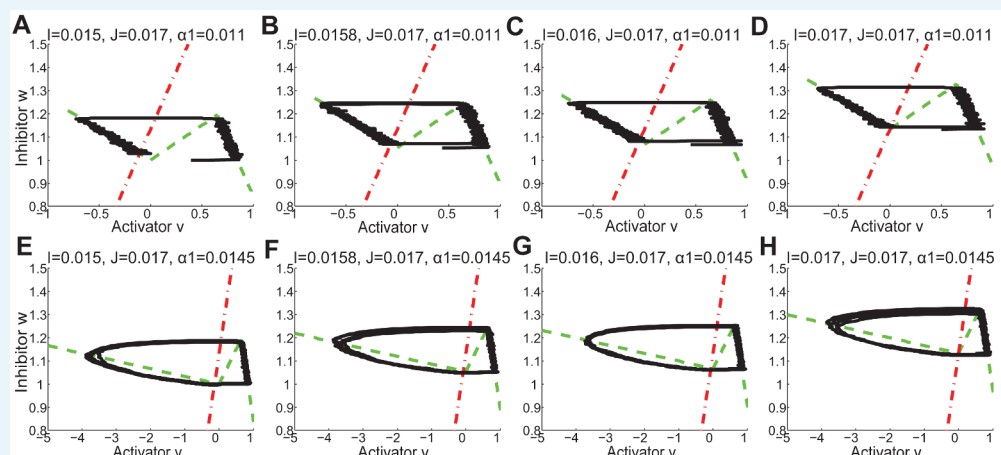

**Appendix 2—figure 34.** The nullclines ($\dot{w} = 0$ red dash-dot lines, and $\dot{v} = 0$ green dash lines) and stochastic trajectories (black lines) under different values of $I$, $J$ and $\alpha_1$ in FHN model. (A, E) $I = 0.015$, $J = 0.017$, as in **Appendix 2—figure 33D**. (B, F) $I = 0.0158$, $J = 0.017$, as in **Appendix 2—figure 33E**. (C, G) $I = 0.06$, $J = 0.017$, as in **Appendix 2—figure 33F**. (D, H) $I = 0.017$, $J = 0.017$, as in **Appendix 2—figure 33B**.

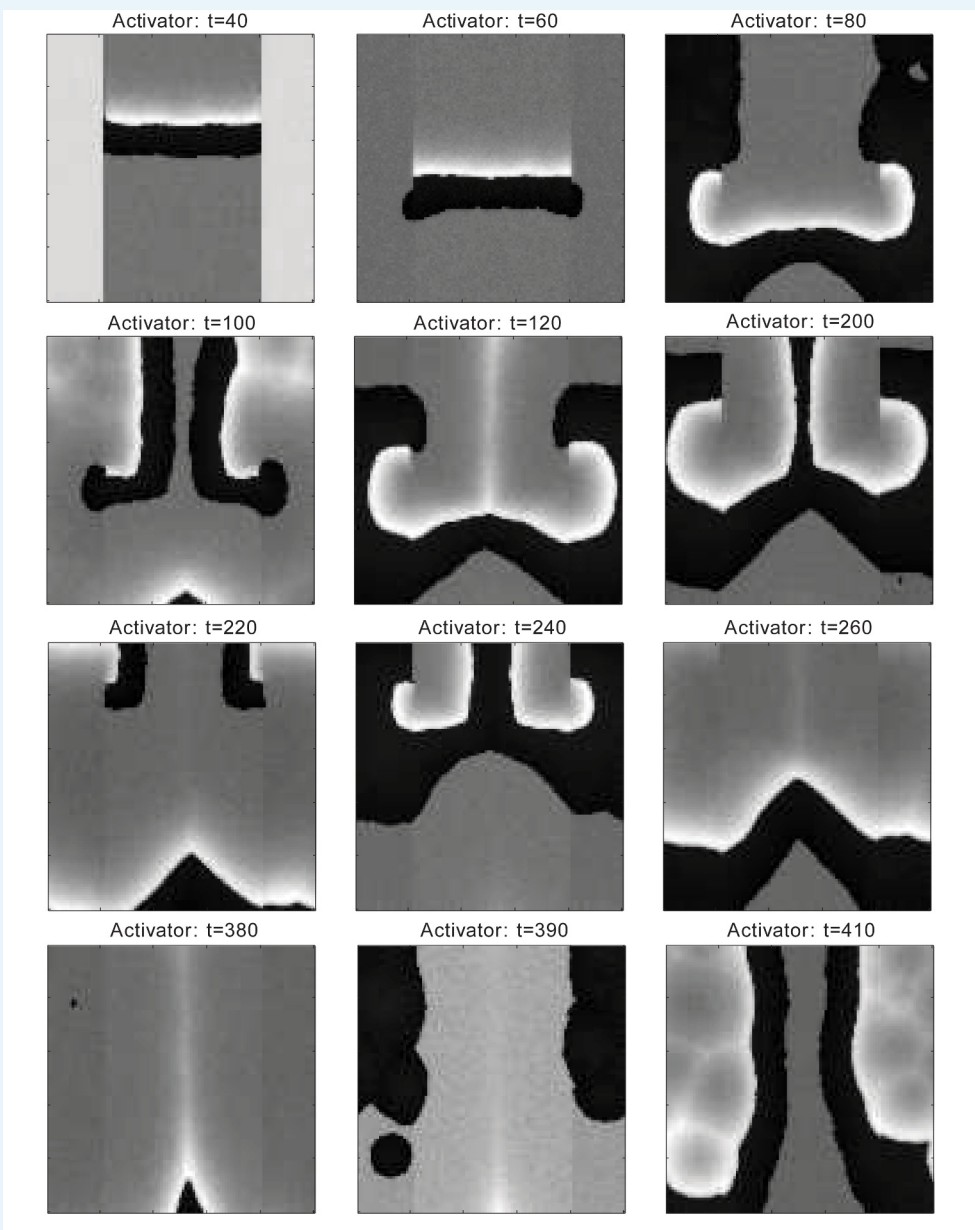

**Appendix 2—figure 35.** Dorsal-ventral interaction simulation by the FHN model. In comparison to *Figure 2F*. Dorsal temporal advancement is built-in, the HFs have shorter telogen in the ventral domain than in the dorsal. Figure shows the activator activities (white-low, black-high). Interaction between the domains with the dorsal domain showing head-to-tail asynchrony caused by the built-in temporal advancement last for several cycles, until it breaks into waves propagating from ventral to dorsal showing bilateral symmetry, which lasts during the following few cycles. Using the FHN model, we cannot obtain the quick degradation of the early dorsal head-to-tail asynchrony as early as the third or fourth cycle, as observed in experiments or simulated by our model.

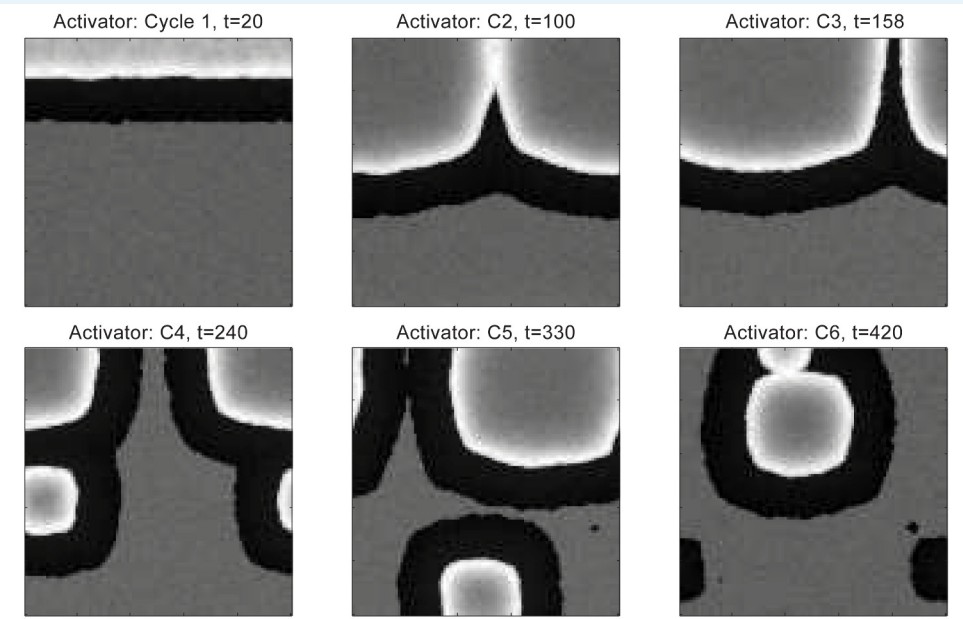

**Appendix 2—figure 36.** Dorsal wave propagation with temporal phase advancement built-in in HFs near the head, simulated by the FHN model. In the homogeneous dorsal domain, the early head-to-tail asynchrony quickly degrades into spontaneous initiations, and no bilateral symmetry is observed.

## Supplementary tables

**Appendix 2—table 1.** Parameter values in the reaction-diffusion system: Appendix 2-*Equations (1–4)*. (1) In *Figure 1C*, $\beta_I$ ranges from $-11$ to 2, and *Figure 1B* is obtained under $\beta_I = 0$; both are noise-free, thus $\lambda_A = \lambda_I = 0$. (2) In *Appendix 2—figure 6*, low, medium and high inhibitor L correspond to $\beta_I = -6, -3, 0$, respectively, and weak, medium and strong noise correspond to $\lambda_A = \lambda_I = 2, 4, 6$ – the same for both multiplicative and additive noise. (3) In *Appendix 2—figure 8* and *9*, low R and high R correspond to $R_{\text{tot}}^A = R_{\text{tot}}^I = R = 6$ or 60, respectively, while low, medium and high inhibitor L correspond to $\beta_I = -6, -3, 0$, respectively, which is the same as in *Appendix 2—figure 6*. *Appendix 2—figure 8* and *9* again show noise-free results, hence $\lambda_A = \lambda_I = 0$. Also see *Appendix 2—table 2* for supplementary parameter values.

**Reaction - Diffusion System**

| Parameters | | 1D | 2D | 3D Dorsal | Ventral |
|---|---|---|---|---|---|
| Preservation of Receptors | $R_{\text{tot}}^A$ / $R_{\text{tot}}^I$ | 10 | 10 | 15 | 50 |
| Diffusion | $D_A$ / $D_I$ | 1 | 0.5 | 0.5 | |
| Reaction | $k_{\text{on}}^A$ | 0.3 | | | |
| | $k_{\text{off}}^A$ | 0 | | | |
| | $k_{\text{on}}^I$ | 0.5 | | | |
| | $k_{\text{off}}^I$ | 0 | | | |

*Appendix 2—table 1 continued on next page*

*Appendix 2—table 1 continued*

**Reaction - Diffusion System**

| Parameters | | 1D | 2D | 3D Dorsal | Ventral |
|---|---|---|---|---|---|
| Degradation | $k_{\text{deg}}^A$ | 2 | | | |
| | $k_{\text{deg}}^I$ | 4 | 6.5 | 7 | |
| Concentration Threshold | $\Delta_+$ | 1.5 | 2.25 | 2.6 | |
| | $\Delta_-$ | 0 | | | |
| Noise | $\lambda_A$ | 2/4/6 | 3.2 | See *Appendix 2—table 2*. | |
| | $\lambda_I$ | | | | |
| Others | $\beta_A$ | 4 | | | |
| | $\beta_I$ | $-11 \rightarrow 2$ | 2.3 | | |

**Appendix 2—table 2.** Parameter values in the reaction-diffusion system in 3D simulations, supplementary to *Appendix 2—table 1*.

| Figures | $\lambda_A = \lambda_I$ | Others |
|---|---|---|
| *Figure 2F* | 0.02 | |
| *Appendix 2—figure 7* | 0.05 | |
| *Appendix 2—figure 12* | 0.035 | |
| *Appendix 2—figure 13B* | 0.02 | $\Delta_+ = 1.8$ |
| *Appendix 2—figure 15* | 0.035 | |
| *Appendix 2—figure 16* | 0.05 | $\Delta_+ = 3, \; \alpha_I^I = 22$ |
| *Appendix 2—figure 17* | 0.02 | |
| *Appendix 2—figure 18* | 0.02 | |
| *Appendix 2—figure 27* | 0 | $\Delta_+ = 2.6/2.7$ |
| *Appendix 2—figure 28* | 0.02 | $R_{\text{tot}}^A = R_{\text{tot}}^I = 15/20$ |
| *Appendix 2—figure 29* | 0.02 | $R_{\text{tot}}^A = R_{\text{tot}}^I = 15/30$ |
| *Appendix 2—figure 30* | 0.02 | $R_{\text{tot}}^A = R_{\text{tot}}^I = 15/40$ |

**Appendix 2—table 3.** Values of temporal and geometric parameters.

**Time and geometry**

| Parameters | 1D | 2D | 3D |
|---|---|---|---|
| $\mathrm{d}t$ | 0.1 | | 0.01 |
| $\mathrm{d}x$ | NA | 1.5 | 2 |
| $\mathrm{d}y$ | NA | | 2 |
| $\mathrm{d}z$ | 0.1 | | 0.2 |
| $h_I$ | -1 | | |
| $d_I$ | 0.5 | | 0.6 |
| $d_{II}$ | 0.8 | | |
| $h_{\max}$ | -6 | | |
| $h_{\min}$ | $-1.8$ | | |
| $X$ | NA | 90 | 60/100 |
| $Y$ | NA | | 100 |
| $Z$ | $-10$ | | |

**Appendix 2—table 4.** Activity strength of activator/inhibitor ligands (L), antagonists (A), receptors (R) and ligand-bond receptors (LR) in different phases within a full HF growth cycle.

| | | Activator | | Inhibitor | |
|---|---|---|---|---|---|
| | | Region I | Region II | Region I | Region II |
| | | (Bulge) | (sHG, Matrix, DP) | (Bulge) | (sHG, Matrix, DP) |
| Late Competent Telogen (late C) | L | 3 | (3,0,4) | 2 | (2,0,3) |
| | A | 2 | (2,0,1) | 0 | (0,0,3) |
| | R | 2 | (2,0,4) | 3 | (3,0,3) |
| | LR | 4 | (4,0,4) | 1 | (1,0,3) |
| Propagating Anagen (P) | L | 2 | (0,4,4) | 2 | (0,3,3) |
| | A | 2 | (0,2,3) | 0 | (0,2,2) |
| | R | 2 | (0,3,4) | 3 | (0,3,3) |
| | LR | 4 | (0,4,4) | 1 | (0,3,3) |
| Autonomous Anagen (A) | L | 2 | (0,4,4) | 2 | (0,4,4) |
| | A | 2 | (0,4,4) | 0 | (0,2,2) |
| | R | 2 | (0,3,4) | 3 | (0,3,3) |
| | LR | 2 | (0,4,4) | 3 | (0,3,3) |
| Catagen (Cat) | L | 2 | (2,0,2) | 2 | (2,0,4) |
| | A | 2 | (2,0,4) | 0 | (0,0,1) |
| | R | 2 | (2,0,4) | 3 | (3,0,3) |
| | LR | 1 | (1,0,1) | 3 | (3,0,3) |
| Refractory Telogen (R) | L | 2 | (2,0,1) | 2 | (2,0,4) |
| | A | 2 | (2,0,4) | 0 | (0,0,1) |
| | R | 2 | (2,0,4) | 3 | (3,0,3) |
| | LR | 1 | (1,0,1) | 4 | (4,0,4) |
| Early Competent Telogen (early C) | L | | | | |
| | A | Similar to refractory telogen | | Similar to late competent telogen | |
| | R | | | | |
| | LR | | | | |

**Appendix 2—table 5.** Parameter values of ligand productions (Appendix 2-**equations (9, 10)**) and HF growth (Appendix 2-**equations (11, 12)**).

| Ligand production | | | | HF growth | | | |
|---|---|---|---|---|---|---|---|
| Parameters | 1D | 2D | 3D | Parameters | 1D | 2D | 3D |
| $\alpha_A^{+,1}$ | 6 | 20 | 40 | $H^+$ | 0.2 | | |
| $\alpha_A^{+,0}$ | 3 | 0 | 0 | $m^+$ | 4 | | |
| $\alpha_A^{-}$ | 1.5 | 1.5 | 3 | $H^-$ | 2 | | |
| $\alpha_I^{0}$ | −7.8 | −13 | −32 | $m^-$ | 2 | | |
| $\alpha_I^{1}$ | 6 | 10 | 20 | | | | |

## Supplementary appendix 2 - video captions

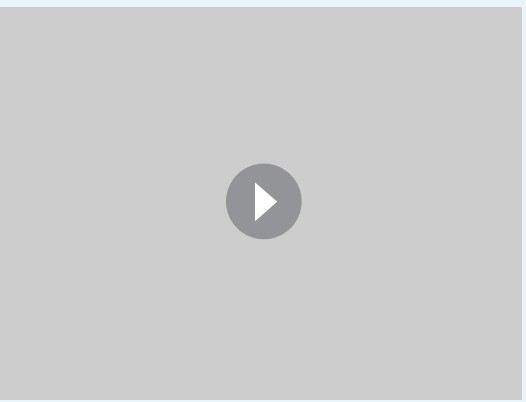

**Appendix 2—video 1.** 3D simulation of dorsal HF waves, where all HFs have equal probability of anagen initiation, allowing spontaneous anagen initiations. Here we see spontaneous anagen initiations over the dorsal domain from the very first cycle, against the experimental observations of early head-to-tail wave degenerating into later spontaneous initiations showing some bilateral symmetry. Also see *Appendix 2—figure 12*.

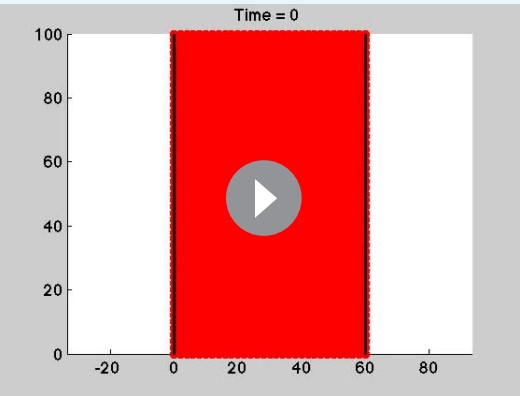

**Appendix 2—video 2.** 3D simulations of dorsal HF waves, where HFs near the head have higher built-in levels of activator signaling. Here we see highly successive steady waves characterized as the 'target-like wave' pattern, also against experimental observations. Also see *Appendix 2—figure 13B.*

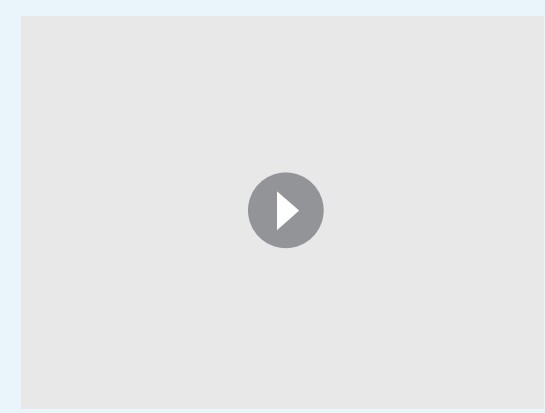

**Appendix 2—video 3.** 3D modeling of HF growth pattern evolution with head-to-tail hair cycle asynchrony. Initial asynchrony was introduced into the field by assigning a group of HFs on one end with a one-time increase in activator ligands. Simulations faithfully reproduce head-to-tail asynchrony of the first cycle. However, this asynchrony fails to degrade quickly, and persists during multiple consecutive cycles. This differs from rapid HF growth pattern evolution observable in mice. Also see *Appendix 2—figure 15*.

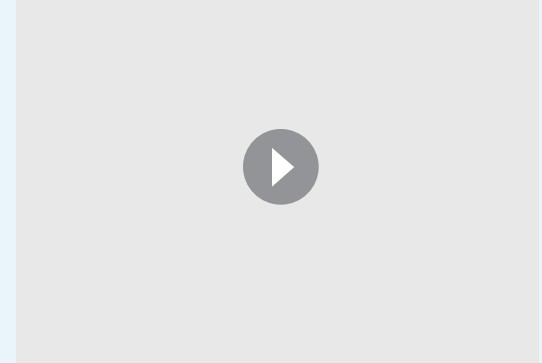

**Appendix 2—video 4.** 3D modeling of dorsal-ventral HF wave interaction. Initial head-to-tail asynchrony in the dorsal domain was introduced by assigning a group of HFs on one end with a one-time increase in activator ligands. We also block ventral HF growth until t = 14 to simulate the temporal-advancement in dorsal domain. Ventral HFs has faster cycle than dorsal HFs. Simulations reproduce head-to-tail asynchrony of first two cycles, which degrades into spontaneous waves possessing bilateral symmetries in later cycles, similar to what is observed from experiments. Also see *Figure 2F*.

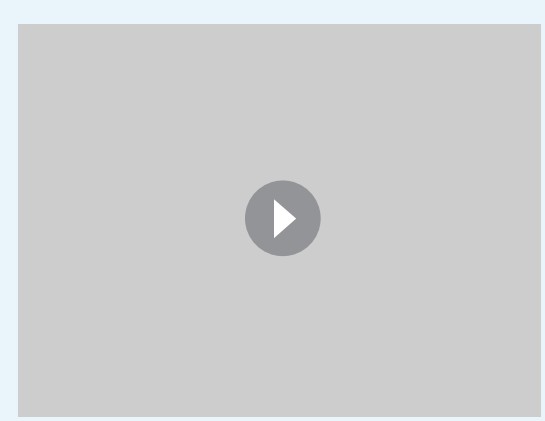

**Appendix 2—video 5.** Regions like eyelids, physical breaks in the skin can break homogenous wave spreading. The black region is 'cut-off' from the homogeneous outside region, i.e., no HF growth is allowed in this region and no molecules are allowed to diffuse into it. As the HF wave propagates, the black region breaks the wave spreading, creating a distortion-like effect. Also see *Figure 7A*.

