## [Decision Letter]

Thank you for submitting your article "‘A multi-scale model for hair follicles reveals heterogeneous domains driving rapid spatiotemporal hair growth patterning"’ for consideration by *eLife*. Your article has been reviewed by three peer reviewers, one of whom is a member of our Board of Reviewing Editors and the evaluation has been overseen by Fiona Watt as the Senior Editor. The reviewers have opted to remain anonymous.

The reviewers have discussed the reviews with one another and the Reviewing Editor has drafted this decision to help you prepare a revised submission.

We found that your manuscript describes an interesting phenomenon that mouse skin is composed of different anatomic regions that interact as heterogeneous excitable domains. The authors used thorough analysis of the whole body hair cycle dynamics coupled with novel mathematical modeling to explain the emergence of certain hair cycle patterns in mouse skin. Extensive mRNA profiling analysis offers some hints as to the molecular mechanisms behind some of the observed features. This work opens up a new angle to analyze and understand the nature of hair growth on the whole body scale, an important perspective not appreciated before.

Major comments to be addressed in a revised manuscript:

1) In general, the reviewers thought that these results should be placed in a broader context with the authors' own work and that of others. First, what is the difference between this manuscript, Plikus et al., 2011 and the interaction between follicles that was recently explored by Chen et al., 2015? The authors make some comparisons in the Results section (subsection “‘Model reveals skin is a heterogeneous excitable medium.”’) but this type of logic needs to come into the experimental design for the introduction and results. Second, the focus on Wnt and Bmp signaling seems redundant based on previous work from several labs. Third, the authors should discuss how a similar approach could benefit the study of other tissues?

2) Often the text is written in a confusing manner. For instance, "‘we scaled the model both into two (Appendix 2—Figure 4) and three dimensions (Appendix 2—Figure 4)"’ does not make sense to me. This lack of clarity makes Figure 1 and Figure 2 almost impossible to interpret.

3) Figure 2 do not clearly show "‘neck-to-tail asynchrony"’ in hair morphogenesis. Do the authors mean that the center is different from the left to right regions?

In Figure 2 the authors demonstrated a scenario of how bilaterally symmetric hair growth pattern could emerge. Even with the assumptive parameters they described in the text, it is not clear why the spread of anagen from dorsal-ventral only occurs in the posterior part of the ventral skin, such as from t=50 to t=62. Is it assumed that the anterior- posterior parts of the ventral skin are also different? Whether or not this is the case here, the authors should expand to explain this phenomenon because it affects the outcome of the entire model. In this manuscript the authors didn't address the anterior- posterior difference of the skin, but given the nature of this study, it would be helpful to clarify why the authors choose to do so.

4) Figure 3 uses a Wnt reporter to examine hair cycling but it is unclear whether the luciferase signal is skin specific.

The result in Appendix 1 is highly interesting. It suggests the faster cycling nature of the chin skin is intrinsic, instead of relying on neighboring tissues in the chin like the cartilage in the ear. To drive the point home, it would help to do the *Flash* analysis on the chin graft and compare the dynamics of Wnt signaling in graft compared to native chin skin. Also this result should be moved to main figure because it serves as a strong justification to do subsequent RNA-seq analysis.

5) The implication that the ventral skin initiates hair cycling is interesting and not fully tested experimentally in this manuscript.

6) The hair follicle is certainly a good model to begin to examine these problems and the combination of experiment and modeling is very powerful. Could it apply to other large spread out tissues such as muscle, blood, bones, etc? Maybe this can be discussed. It is very interesting that the model predicts a sort of self-sufficiency of the regenerative process, in which, like a perpetuum mobile, it can keep going potentially without "‘higher-order"’ control or even trigger from a systemic signal. Maybe this aspect can be emphasized.

7) Given the amount of work the authors put in Figure 4, the message is surprisingly little, other than the description of very complicated and a little confusing dynamic expression patterns. The authors do have multiple genetic models used in this study that can help them sort out the functions of some of the signal pathways in defining the different anatomic domains. For instance in their K14-Noggin, K14-BMP4, K14Cre-Wnt7b fl/fl, K14-Wnt7a, K5trTA-teto-Dkk1 models, do the chin skin cycling frequencies change, do the ventral skin anagen duration change? The authors could choose at least 1 representative model of Wnt and BMP signal pathways to do this.

8) In Figure 6, the percentages of the ear follicles going into anagen in these transgenic models needs to be quantified. It will help address how dominant these signal pathways are in maintaining the quiescence of the ear skin.

---

## [Author Response]

*Major comments to be addressed in a revised manuscript:*

*1) In general, the reviewers thought that these results should be placed in a broader context with the authors' own work and that of others. First, what is the difference between this manuscript, Plikus et al., 2011 and the interaction between follicles that was recently explored by Chen et al., 2015? The authors make some comparisons in the Results section (subsection “‘Model reveals skin is a heterogeneous excitable medium.”’) but this type of logic needs to come into the experimental design for the introduction and results. Second, the focus on Wnt and Bmp signaling seems redundant based on previous work from several labs. Third, the authors should discuss how a similar approach could benefit the study of other tissues?*

We thank the reviewers for their suggestions. In this submission, we made changes in the Introduction and Discussion to clarify that this work is the first to examine hair regeneration across most of the mouse’s body skin, and its findings represent a significant advance over previous knowledge on group-level hair growth behavior in the dorsal skin. From a modeling perspective, this is the first multi-scale hair follicle model that incorporates both molecular dynamics and physical growth into one model consisting of three dimensions in space, and that faithfully predicts cyclic behaviors at the level of both a single hair follicle and of hair populations collectively. Through both experiments and modeling, our work revealed that:

a) The physiological pace of hair regeneration differs among distinct anatomical sites. It is prominently faster in the chin and ventral skin relative to dorsal skin, and very slow in the ear skin. Thus, the skin as a whole appears to function as a complex regenerative landscape with regions of fast, slow, and very slow hair renewal. Our new data in this revision (Appendix 1—Figure 23) shows that this behavior produces fur coat with variable hair density, which likely serves an adaptive role. Prior works on dorsal skin did not consider that hair regeneration dynamics prominently differ between body sites. We now highlight this in the Introduction and in the Discussion.

b) WNT/BMP activator/inhibitor signaling pair modulates hair regeneration in all skin regions. We believe this finding is important and non-redundant with prior works. Our current work shows that WNT/BMP signaling is a universal “‘molecular language”’ for hair growth on the entire body, rather than only on a specific body site, such as the dorsal skin shown in previous works.Indeed, the choice of dorsal skin as a model for studying hair growth in most prior works is primarily due to convenience. By studying hair growth in other, “‘less convenient”’ body sites we observe that the mechanism driven by WNT/BMP signaling pair is general. In this revision we took additional care in further strengthening this important aspect of our work through providing new data on skin-wide WNT and BMP reporter activity (Figure 4; Appendix 1—Figure 21), as well as new data on the changes in hair growth patterns in WNT and BMP mutant mice (Figure 4; Appendix 1—Figure 22).

c) Several additional conclusions stem directly from the findings above. First, since all skin regions share a WNT/BMP “‘language”’, hair-to-hair communications naturally arise across the anatomic boundaries. An alternative to this mechanism would be distinct, site-specific signals and full autonomy of regional hair growth. Because WNT/BMP-based communication is possible across the boundaries, it enables novel hair growth behavior not obvious from prior works – fast cycling skin regions (such as chin skin) function as a kind of pacemaker for skin-wide hair growth. We emphasize this in the Introduction and in the Discussion.

d) Second, we found that non-skin tissues with prominent WNT/BMP expression profiles can act as a signaling macro-environment for hair follicles and strongly influence their regeneration pace. In this study we show that in the mouse ear, WNT-low/BMP-high cartilage/muscle complex slows hair growth in the ear skin to a virtual standstill. This finding expands the repertoire of tissues with a signaling macro-environment function to include closely positioned anatomic structures beyond typical skin cell populations. We highlight this in the Discussion.

e) We posit that some of the newly found hair regeneration features can have analogs in other tissues and organs. For instance, dominant anatomically defined pacemakers are common in the electrically coupled muscle-based tissues. Prominently, cardiac tissue contains pacemaker nodes that generate dominant electric rhythms to drive contractile activity of the entire heart. A similar pacemaker exists in the stomach (gastric pacemaker zone), which controls its coordinated peristalsis. Other actively regenerating organs, such as the gastro-intestinal tract and bone marrow likely contain anatomic regions of faster and slower regeneration and, conceivably, they can be coupled to work in coordination. Knowledge learned from the skin system can guide the search for regenerative landscapes in these and other organs. Because coordination principles observed in the skin may be universal for large-scale patterning systems, the likelihood of them operating in other organs is substantial despite prominent anatomical differences. We highlight this in the Discussion.

*2) Often the text is written in a confusing manner. For instance, "‘we scaled the model both into two (Appendix 2—Figure 4) and three dimensions (Appendix 2—Figure 4)"’ does not make sense to me. This lack of clarity makes Figure 1 and Figure 2 almost impossible to interpret.*

We apologize for the confusion. In this revision we included an expanded description of the modeling, both in the main text (Results section) and in the legends for the main Figure 1 and Figure 2 and Appendix 2—Figure 4. We hope that this helps to improve the clarity of the modeling aspects of the study.

*3) Figure 2 do not clearly show "‘neck-to-tail asynchrony"’ in hair morphogenesis. Do the authors mean that the center is different from the left to right regions?*

Indeed, as noted by the reviewers, the neck-to-tail asynchrony in dorsal hair morphogenesis at P0-P1 is fairly subtle. Additionally, subtle differences are observed between the sides and the center of the dorsal skin. We now noted this explicitly in the revised text (Results section). Our data on main Figure 3 and Figure 4 shows that these initial spatio-temporal asynchronies in dorsal hair morphogenesis become effectively “‘wiped out”’ by the dominant chin/ventral to dorsal hair growth spreading at the onset of the second anagen. Nonetheless, we pointed out these initial asynchronies.

*In Figure 2 the authors demonstrated a scenario of how bilaterally symmetric hair growth pattern could emerge. Even with the assumptive parameters they described in the text, it is not clear why the spread of anagen from dorsal-ventral only occurs in the posterior part of the ventral skin, such as from t=50 to t=62. Is it assumed that the anterior- posterior parts of the ventral skin are also different? Whether or not this is the case here, the authors should expand to explain this phenomenon because it affects the outcome of the entire model. In this manuscript the authors didn't address the anterior- posterior difference of the skin, but given the nature of this study, it would be helpful to clarify why the authors choose to do so.*

We agree with the reviewers’ criticism that the modeling result in the original Figure 2 is not fully consistent with the experimentally observed hair growth patterns. In the resubmission, we critically scrutinized our model through comprehensive exploration of parameters, and found that this feature was the result of a particular set of parameter values chosen for our original simulations. In this revision, we tested a wider range of parameter values and found that dorsal/ventral spreading is not an exclusive feature of the posterior skin, but that it can occur along the entire length of the dorsal/ventral boundary. The updated set of parameter values includes: (a) earlier onset of ventral hair follicle morphogenesis, (b) lower signaling threshold for anagen initiation (Δ^+^=2.6), (c) lower activator/inhibitor signaling noise (λ_A_=λ_I_=0.02), and (d) smaller time step length (dt=0.01). Updated results are now shown in the main Figure 2 and the parameters used are shown in the Appendix 2(subsection “‘2D and 3D modeling details”’.1 and Appendix 2—tables 1 to 3. Furthermore, for consistency, we re-run other simulations and updated Appendix 2—Figure 7, Figure 12, Figure 13, Figure 15 to 18 and 28 to 30. In all cases we obtained qualitatively similar results that do not change our original conclusions. Lastly, we also made three additional changes in the presentation of the modeling results in the Appendix 2 file:

1) The effects of the maximum hair follicle growth length h_max_ on wave propagation behaviors were presented in a supplemental table in the original submission. They are now shown in Appendix 2—Figure 27.

2) The effects of total available receptors R^A^and R^I^ on the ventral-dorsal wave interactions were shown in the movies in the original submission. They are now shown in Appendix 2—Figure 28 to 30.

3) New Appendix 2—table 2 is added to accompany Appendix 2—table 1. It details the parameter values used in in each simulation.

*4) Figure 3 uses a Wnt reporter to examine hair cycling but it is unclear whether the luciferase signal is skin specific.*

The original study on *Flash* WNT reporter mice by Hodgson et al., 2014 carefully examined the origin of the luminescence signal and concluded that it is skin specific. In their work, authors state that *'*[…]Upon dissection, no signal beyond WT level was detected in organs other than the skin at both short and long exposure times, although luciferase expression could be detected at RNA levels, thus indicating that transgene activity at high and detectable level is exclusive to the skin without internal organ signal contamination […]*'.* We thus interpret *Flash* reporter signal in the current study as skin-specific. In the revision, we point out the skin-specific nature of *Flash* signal and cite the above reference.

*The result in Appendix 1 is highly interesting. It suggests the faster cycling nature of the chin skin is intrinsic, instead of relying on neighboring tissues in the chin like the cartilage in the ear. To drive the point home, it would help to do the Flash analysis on the chin graft and compare the dynamics of Wnt signaling in graft compared to native chin skin. Also this result should be moved to main figure because it serves as a strong justification to do subsequent RNA-seq analysis.*

We agree that grafting results support intrinsically faster underlying dynamics of WNT and/or BMP signaling in the chin skin. As suggested, we did consider *Flash* skin grafting experiments; however, our *Flash* mice are on a mixed genetic background; this precludes chin-to-dorsal grafting between individual *Flash* animals. In addition, due to the highly traumatic nature of an autologous transplantation procedure from chin region, we were unable to obtain IACUC approval to conduct chin-to-dorsal grafting experiments in the same animal.

Instead, we performed the following additional experiments to strengthen the link between domain-specific hair cycle dynamics and BMP/WNT signaling:

a) We performed additional chin and dorsal WT skin grafting experiments to SCID donors to strengthen the findings that chin skin has faster intrinsic hair cycle dynamics. New grafting results are now shown in main Figure 3.

b) We performed in-depth analysis of the domain specific RNA-seq data, with a focus on early third telogen. This new analysis is now shown on main Figure 3 to 3F. Compared to dorsal skin, chin and ventral skin shows prominent expression differences for the BMP pathway (less BMP ligands and more BMP antagonists) and WNT pathway (less WNT antagonists). Because the RNA-seq was performed on carefully dissected skin, our analysis further confirms that the observed BMP and WNT signaling differences are intrinsic to these skin regions.

c) We also examined lacZ expression patterns in *Axin2-lacZ* WNT reporter and *BRE-gal* BMP reporter mice during second telogen (main Figure 4 to 4I and Appendix 1—Figure 21). Consistent with *Flash* WNT reporter data, we observe faster increase in *Axin2-lacZ* reporter activity in the chin and ventral skin as compared to dorsal skin. Dynamic lacZ signal localizes to dermal papillae of telogen hair follicles, where it functions as a marker of telogen competence (Plikus et al., 2011). We see many more lacZ+ dermal papillae in the chin and ventral skin during early second telogen, on day P36. For *BRE-gal* reporter mice, we see fewer lacZ+ hair follicles (bulge region expression) in the chin and ventral skin as compared to dorsal skin (Figure 4). These new results strengthen our findings that telogen hair follicles acquire low-BMP and high-WNT signaling status faster in the chin and ventral skin compared to dorsal skin.

d) We also compared second telogen skin of chin, ventral and dorsal sites by histology, immuno-histochemistry and FACS. Consistent with the Gene Ontology analysis on the RNA-seq data (main Figure 3), we observe site-specific differences in the distribution of dermal adipose tissue and *panniculus carnosus* skeletal muscle (Appendix 1—Figure 19), as well as macrophages (Appendix 1—Figure 20). We speculate that these regional differences in skin composition may contribute to regional differences in WNT and BMP activities, which underlie faster chin and ventral hair cycling. Admittedly, an in-depth follow-up study will be necessary to identify and verify the major site-specific cellular sources for WNT and BMP ligands and antagonists. At the same time, our current study establishes clear site-specific hair cycle dynamics and promising regional differences in skin microanatomy, warranting follow-up experiments.

*5) The implication that the ventral skin initiates hair cycling is interesting and not fully tested experimentally in this manuscript.*

We speculate that generally faster hair cycle in the chin and ventral skin helps to achieve higher fur density, which can aid with increased insulation and protection against mechanical abrasion of the ventrum in low gait animals, such as mice. In this revision, we examined club hair density distribution between the three regions in adult, 4-month old mice. Indeed, we found that chin and ventral telogen hair follicles consistently had more club hairs (three or more) as compared to dorsal telogen hair follicles (mostly two or one club hair). This new data is now shown on the Appendix 1—Figure 23, and the relevant speculations are added in the Discussion. We also acknowledged in Discussion that the rate of hair regeneration is only one of the factors determining fur density, and that the latter can also be affected by the rate of hair fallout (*aka* exogen), a parameter not directly tested in this work.

*6) The hair follicle is certainly a good model to begin to examine these problems and the combination of experiment and modeling is very powerful. Could it apply to other large spread out tissues such as muscle, blood, bones, etc? Maybe this can be discussed. It is very interesting that the model predicts a sort of self-sufficiency of the regenerative process, in which, like a perpetuum mobile, it can keep going potentially without "‘higher-order"’ control or even trigger from a systemic signal. Maybe this aspect can be emphasized.*

We thank the reviewers for this excellent suggestion. In the revised Discussion, we added that skin with its many cycling hair follicles is just one, albeit prominent example, of many organs that implements large-scale coordination of its physiological activities. As mentioned above, electrically coupled heart and stomach display large-scale coordination of their contractile activities from anatomically defined pacemaker regions. Other organs, prominently the intestine share many attributes with skin – analogous to skin, intestines contain many repetitive stem cell-rich units along its villi. It is conceivable that higher-order coordination of regeneration, analogous to that seen in haired skin occurs between intestinal villi.

Related to hair regeneration, several levels of higher-order coordination exist:

a) Coordination within hair populations occurs via self-organization between individual hair follicles (Plikus et al., 2008 and Plikus et al., 2011). This can be further modulated by the local signaling macro-environment, including adipose cells, immune cells, etc.

b) Coordination between distinct anatomical regions, such as between chin, ventral and dorsal skin sites described in the current study.

c) Systemic hormonal and neuronal coordination. Indeed, large amount of literature exists on hair growth coordination by humoral factors, including but not limited to prolactin and sex hormones, and sympathetic nerves.

In a given species, fur renewal can be regulated by several of the above mechanism working either in parallel or hierarchically, when one mechanism overrides another. In the revised Discussion, we provide succinct speculation of these ideas.

*7) Given the amount of work the authors put in Figure 4, the message is surprisingly little, other than the description of very complicated and a little confusing dynamic expression patterns. The authors do have multiple genetic models used in this study that can help them sort out the functions of some of the signal pathways in defining the different anatomic domains. For instance in their K14-Noggin, K14-BMP4, K14Cre-Wnt7b fl/fl, K14-Wnt7a, K5trTA-teto-Dkk1 models, do the chin skin cycling frequencies change, do the ventral skin anagen duration change? The authors could choose at least 1 representative model of Wnt and BMP signal pathways to do this.*

We agree with this point. For this revision, we moved most of the correlative RNA-seq analysis into the supplement. Furthermore, we performed more in-depth analysis on the early second telogen RNA-seq (main Figure 4 to 4F). This new analysis shows more clearly that chin and ventral skin express lower levels of WNT antagonists and BMP ligands and higher levels of BMP antagonists, consistent with the possibility that these regions are less refractory. Following on this correlation, we carefully examined lacZ staining patterns in *Axin2-lacZ* WNT reporter and *BRE-gal* BMP reporter mice (main Figure 4 to 4I and Appendix 1—Figure 21). This new data supports the notion that both chin and ventral telogen skin acquires low-BMP and high-WNT signaling status (i.e. less refractory status) faster as compared to dorsal skin. Lastly, following reviewers’ suggestion we examined skin-wide hair growth patterns in *Krt14-Wnt7a* and *Krt14-Bmp4* mutant mice (main Figure 4 and Appendix 1—Figure 22). Consistent with the speculated role for WNT pathway as the skin-wide hair cycle activator we observe loss of ventral-to-dorsal hair cycle initiation dominance and ectopic hair growth initiation sites appear in the dorsal skin of *Krt14-Wnt7a* mice at the start of the second (Appendix 1—Figure 22) and third anagen (Figure 4). In analogy, consistent with the speculated role for BMP pathway as the skin-wide hair cycle inhibitor we observe stalled ventral-to-dorsal hair cycle spreading, formation of premature hair growth boundaries and generally asymmetric, jagged hair growth patterns in *Krt14-Bmp4* mice at the start of third anagen (Figure 4).

*8) In Figure 6, the percentages of the ear follicles going into anagen in these transgenic models needs to be quantified. It will help address how dominant these signal pathways are in maintaining the quiescence of the ear skin.*

Phenotype quantification and additional images of individual ear skin samples from WT, *Krt14-Noggin* and *Krt14-Wnt7a* mice have been added in the new Appendix 1—Figure 28. Both mutants show statistically significant increase in the number of spontaneously cycling ear hair follicles.